# DoF: A Diffusion Factorization Framework for Offline Multi-Agent Reinforcement Learning

**Chao Li**[ab*]**, Ziwei Deng**[ab*]**, Chenxing Lin**[ab]**, Wenqi Chen**[e]**, Yongquan Fu**[c]**,**
**Weiquan Liu**[abd]**, Chenglu Wen**[ab]**, Cheng Wang**[ab]**, Siqi Shen**[ab†]
[a]Fujian Key Laboratory of Sensing and Computing for Smart Cities,
School of Informatics, Xiamen University (XMU), China
[b]Key Laboratory of Multimedia Trusted Perception and Efficient Computing, XMU, China
[c]School of Computer, National University of Defense Technology, China
[d]College of Computer Engineering, Jimei University, China
[e]School of Information and Software Engineering,
The University of Electronic Science and Technology of China
`{chaoli,dengziwei,lincx1123}@stu.xmu.edu.cn`,
`{siqishen,cwang,clwen,wqliu}@xmu.edu.cn, {yongquanf}@nudt.edu.cn`,
`{chenwenqi}@std.uestc.edu.cn`

## Abstract

Diffusion models have been widely adopted in image and language generation and are now being applied to reinforcement learning. However, the application of diffusion models in offline cooperative Multi-Agent Reinforcement Learning (MARL) remains limited. Although existing studies explore this direction, they suffer from scalability or poor cooperation issues due to the lack of design principles for diffusion-based MARL. The Individual-Global-Max (IGM) principle is a popular design principle for cooperative MARL. By satisfying this principle, MARL algorithms achieve remarkable performance with good scalability. In this work, we extend the IGM principle to the Individual-Global-identically-Distributed (IGD) principle. This principle stipulates that the generated outcome of a multi-agent diffusion model should be identically distributed as the collective outcomes from multiple individual-agent diffusion models. We propose DoF, a diffusion factorization framework for Offline MARL. It uses noise factorization function to factorize a centralized diffusion model into multiple diffusion models. We theoretically show that the noise factorization functions satisfy the IGD principle. Furthermore, DoF uses data factorization function to model the complex relationship among data generated by multiple diffusion models. Through extensive experiments, we demonstrate the effectiveness of DoF. The source code is available at https://github.com/xmu-rl-3dv/DoF.

## 1 Introduction

Generative diffusion models (Ho et al., 2020; Song et al., 2021b) have achieved great success in multiple domains such as image generation (Rombach et al., 2022). Due to the powerful modeling ability of generative modeling, researchers have applied diffusion model (Ajay et al., 2023) to generate decisions in the reinforcement learning domain. In this domain, offline reinforcement learning approaches (Fujimoto et al., 2019; Kumar et al., 2020; Shao et al., 2023; Yang et al., 2021) learn policies from offline data logged by the operational system. With access to such data, for reinforcement learning, diffusion models can be used to learn a probabilistic model of trajectories or actions (Ajay et al., 2023; Janner et al., 2022; He et al., 2023; Wang et al., 2023).

The success of diffusion model in offline reinforcement learning domain (Ajay et al., 2023) motivates us to apply it in cooperative multi-agent reinforcement learning (MARL). There are a few diffusion-based MARL approaches that exist. In MADIFF (Zhu et al., 2024), a centralized diffusion process (CDG) is trained to generate joint trajectories. During execution, the same CDG is used to generate

---

[*]Equal contribution
[†]Corresponding author

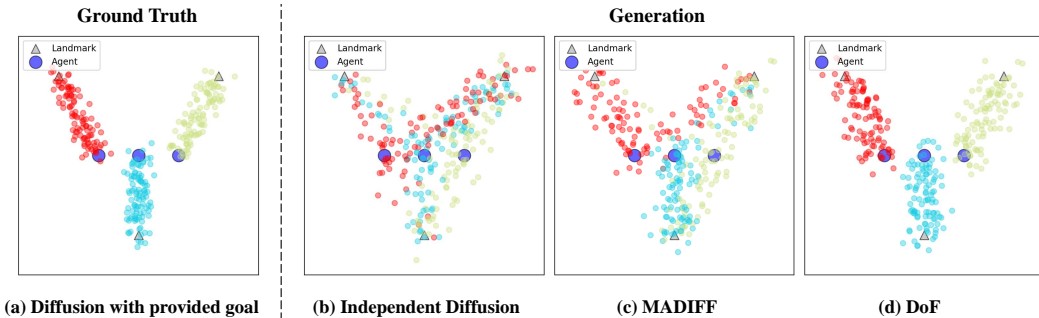

Figure 1: Landmark Covering Game: three agents must reach different landmarks, avoiding collisions. (a) Each agent (Ajay et al., 2023) goes to a distinct goal provided by human. (b) Independent Diffusion Agents are independently trained and tested. (c) MADIFF: multi-agent plan trajectories using diffusion (Zhu et al., 2024). (d) DoF: agents cooperatively plan trajectories that closely match trajectories of (a). A circle is the position of a sampled point of an agent's trajectory. Each color represents a distinct agent.

trajectories for each agent. It suffers from the scalability issue that the state and the action space increase exponentially with the number of agents. Another avenue of applying diffusion (Li et al., 2023) to MARL is to learn an independent diffusion model for each agent. Although this approach is scalable, it suffers from poor-cooperative issues that each independent diffusion model does not fully consider cooperation. It is challenging to address the scalability and poor-cooperative issues of diffusion models in cooperative MARL (Chen et al., 2024).

In MARL, the centralized training with decentralized execution (CTDE) paradigm is widely adopted. Under this paradigm, the individual-global-max (IGM) principle (Rashid et al., 2018) is proposed to address the scalability and poor-cooperative challenges. The IGM principle requires that the collective greedy selection of action of each individual agent is equivalence to the optimal action of a whole multi-agent system. Many excellent algorithms (Rashid et al., 2020a; Son et al., 2019; Qiu et al., 2021) satisfied the IGM principle have been developed. However, the IGM principle is developed for value-based MARL (Hernandez-Leal et al., 2018), not for diffusion-based MARL. Moreover, the IGM principle is only applicable for methods that learn factorized policies (Rashid et al., 2018; Shen et al., 2022; Wang et al., 2021; Rashid et al., 2020a). It is unsuitable for planning-based methods (Ajay et al., 2023) that generate (or predict) future outcomes and plan based on predictions. *A more general principle than the IGM principle is lacking for diffusion-based MARL.*

To address the above limitations, we propose the Individual-Global-identically-Distribute (IGD) principle, which is a generalization of the IGM principle if the diffusion process can generate deterministic actions exactly. It requires that the collectively generated outcome of each individual agent follows the same distribution as the generated outcome of a whole multi-agent system. Given a diffusion method that satisfies the IGD principle, a centralized diffusion model (CDM) can be used to generate high-return data (e.g., trajectories or actions). Once trained, the CDM, parameterized by $\theta_{tot}$, is factored into multiple small decentralized diffusion models (DDM), each parameterized by $\theta_i$. During execution, each agent uses a decentralized diffusion model to generate data. The collection of each agent's generated data follows the same distribution as the high-return data generated by the CDM. The IGD principle is flexible, applying to both factorized policies and planners.

In this work, we propose DoF, a diffusion factorization framework for offline MARL. The same as other diffusion models, the forward process of DoF gradually adds *noise* into data, whereas its backward process does the opposite. DoF utilizes a noise factorization function to ensure that the noise of multi-agent is equivalent to the combination of the noise of each agent. We show theoretically that the noise factorization function satisfies the IGD principle. As shown in Figure 1, DoF generates data that matches ground truth better than other methods, which demonstrates the effectiveness of the noise factorization function. Further, DoF utilizes a data factorization function to model the relationship among data generated by agents.

For evaluation, we conduct extensive experiments on the StarCraft II MARL tasks (Samvelyan et al., 2019; Ellis et al., 2023), the Multi-Particle Environment (MPE) (Lowe et al., 2017b), Multi-Agent Mujoco (de Witt et al., 2020), and several illustrative examples. The experimental results demonstrate the effectiveness of DoF.

## 2 BACKGROUND

### 2.1 DEC-POMDPS

We consider cooperative multi-agent reinforcement learning tasks, which can be modeled as Decentralized Partially Observable Markov Decision Processes (Dec-POMDPs) (Oliehoek & Amato, 2016). In this work, agents do not communication. The Dec-POMDPs is represented as tuple $G = \langle \mathcal{S}, \{\mathcal{U}_i\}_{i=1}^N, P, r, \{\mathcal{O}_i\}_{i=1}^N, \{\sigma_i\}_{i=1}^N, N, \gamma \rangle$ for $N$ agents. Please refer to Appendix A.1 for details.

### 2.2 THE INDIVIDUAL-GLOBAL-MAX PRINCIPLE

For Dec-POMDPs, value function factorization methods learn factorized Q value functions, which are used for the execution of each agent. The Individual-Global-Max (**IGM**) principle proposed in (Son et al., 2019) is essential for the realization of value function factorization. It is defined as follows.

**Definition 1** (IGM). *For a joint state-action value function $Q_{\mathrm{jt}} : \mathcal{T}^N \times \mathcal{U}^N \mapsto \mathbb{R}$, where $\boldsymbol{\tau}_{\mathbf{tot}} \in \mathcal{T}^N$ is a joint action-observation history and $\boldsymbol{u} \in \mathcal{U}^N$ is the joint action, if there exist individual state-action functions $[Q_i : \mathcal{T}_i \times \mathcal{U}_i \mapsto \mathbb{R}]_{i=1}^N$, such that the following conditions are satisfied*

$$\arg\max_{\boldsymbol{u}} Q_{\mathrm{jt}}(\boldsymbol{\tau}_{tot}, \boldsymbol{u}) = (\arg\max_{u_1} Q_1(\tau_1, u_1), \ \ldots, \ \arg\max_{u_N} Q_N(\tau_N, u_N)), \tag{1}$$

*then, $[Q_i]_{i=1}^N$ satisfy IGM for $Q_{\mathrm{jt}}$ under $\tau$. We can state that $Q_{\mathrm{jt}}(\boldsymbol{\tau}_{tot}, \boldsymbol{u})$ can be factorized in terms of $[Q_i(\tau_i, u_i)]_{i=1}^N$.*

### 2.3 DIFFUSION MODELS

Diffusion models (Sohl-Dickstein et al., 2015; Ho et al., 2020) are a type of generative model that learns the data distribution $p(\boldsymbol{x}^0)$ from a dataset $\mathcal{D}$. It consists of the forward noising process and the reverse denoising process. In the forward noising process, the data-generating procedure is modeled by $p(\boldsymbol{x}^{k+1}|\boldsymbol{x}^k) := \mathcal{N}(\sqrt{\alpha_k}\boldsymbol{x}^k, (1-\alpha_k)\boldsymbol{I})$, where $\boldsymbol{x}^0$ is a data sample, $\alpha_k \in \mathbb{R}$ determines the level of noise add to data $\boldsymbol{x}^k$. The reverse denoising process is a trainable process which can be modeled as $p_\theta(\boldsymbol{x}^{k-1}|\boldsymbol{x}^k) := \mathcal{N}(\mu_\theta(\boldsymbol{x}^k, k), \Sigma_k)$, where $\mu_\theta(\boldsymbol{x}^k, k)$ is a function of $\boldsymbol{x}^k$ and noise $\epsilon_\theta(\boldsymbol{x}^k, k)$. $\mathcal{N}(\mu, \Sigma)$ is a Gaussian distribution with its mean $\mu$ and variance $\Sigma$. DDPM (Ho et al., 2020) uses the following loss function to train the reverse denoising process.

$$\mathcal{L}(\theta) = E_{k \sim [1,K], \epsilon \sim \mathcal{N}(\mathbf{0}, \boldsymbol{I})}[|\epsilon - \epsilon_\theta(\boldsymbol{x}^k, k)|^2] \tag{2}$$

The *noise model* $\epsilon_\theta(\boldsymbol{x}^k, k)$ estimates the noise $\epsilon \sim \mathcal{N}(0, I)$ added to $\boldsymbol{x}^0$ for $\boldsymbol{x}^k$. Once the noise model is learned, it can be used to generate data. Please refer to Appendix A.2 for details.

## 3 RELATED WORK

Offline reinforcement learning algorithms (Fujimoto et al., 2019; Yang et al., 2021; Kumar et al., 2020; Chen et al., 2021; Janner et al., 2021; Meng et al., 2023; Tseng et al., 2022) learn policies from static operational logs, which circumstances the need for costly online exploration. DoF learns policy from static operational logs too.

In cooperative multi-agent reinforcement learning (MARL), the widely used IGM principle requires agents to work together for a common goal. This work introduces the IGD principle as a generalization of IGM. To satisfy IGM, researchers have proposed various multi-agent reinforcement learning (MARL) value factorization methods (Sunehag et al., 2018; Rashid et al., 2018; Son et al., 2019; Rashid et al., 2020a; Sun et al., 2021; Qiu et al., 2021; Haoyuan et al., 2024), which decompose a joint Q-value function into individual Q-values for each agent. These factorization methods, along with value/policy regularization and heuristics, have also been adapted for offline MARL (Yang et al., 2021; Jiang & Lu, 2021; Fujimoto & Gu, 2021; Shao et al., 2023; Pan et al., 2022), but they often suffer from value function approximation and off-policy learning issues.

Diffusion-based approaches make decisions by generating trajectories or actions. Diffuser (Janner et al., 2022) generates trajectories through classifier-guide diffusion (Dhariwal & Nichol, 2021) and

acts according to generated trajectories. Decision Diffuser (Ajay et al., 2023) enhances Diffuser by using classifier-free guidance (Ho & Salimans, 2021). DiffusionQL (Wang et al., 2023) uses diffusion models to generate actions. Although these approaches are flexible and high-performing, they are *not scalable* for cooperative multi-agent scenarios. For multi-agent settings, DOM2 (Li et al., 2023) uses an independent DiffusionQL diffusion process to make decision for each agent without fully considering cooperation.

The closest work to us is MADIFF (Zhu et al., 2024). It learns a centralized diffusion model (CDM) to generate trajectories. During execution, each agent uses the same CDM to generate trajectories. DoF learns a CDM, which can be factorized into multiple smaller diffusion models that are used by each agent. During decentralized execution, the input complexity of the MADIFF diffusion model is $o(n)$, where $n$ is the number of agents, whereas the input complexity of the DoF diffusion model is $o(1)$. DoF achieves better scalability than MADIFF, thanks to the noise factorization function. Please refer to Appendix A.3 for more discussion.

# 4 DoF: A Diffusion Factorization framework for offline MARL

## 4.1 Motivating Example

In Figure 1, three agents need to cooperatively explore all three landmarks in a short time while avoiding collisions. The most ideal case is that each cooperative agent goes to its closest distinct landmark. Figure 1(a) shows an implementation of the ideal case. Each agent is trained using a decision diffuser (Janner et al., 2022) with a human-given goal. In this Figure, the positions of each agent from 10 episodes are depicted as colored dots. Figure 1(b) shows the results for independent diffusion (ID), where cooperation is not considered. Each agent learns independently, which leads to many collisions. Figure 1(c) shows the result for MADIFF. It performs better than ID but causes collisions, too. Figure 1(d) depicts the results of DoF. For MARL, it is important for agents to collaboratively generate data that mimic cooperative behaviors in ground truth data.

## 4.2 The Individual-Global-identically-Distributed Principle

Under the centralized training with decentralized execution (CTDE) paradigm, the IGM principle is widely followed to address the scalability issues and to promote cooperation. Through satisfying the IGM principle, the collection of greedy local actions of decentralized agents is equal to the optimal jointed actions of centralized multi-agents. Similarly, for diffusion-based MARL, it is important to learn the decentralized diffusion process aligned with the centralized diffusion process. However, the IGM principle is designed for value-based methods, so it is not suitable for diffusion-based methods. A design principle that generalizes the IGM principle is needed. We extend the IGM principle to the Individual-Global-identically-Distributed (IGD) principle, which is defined as follows.

**Definition 2** (IGD). *For a joint total distribution $p_{\boldsymbol{\theta}_{tot}}(\boldsymbol{x}_{tot}^0) := \int p_{\boldsymbol{\theta}_{tot}}(\boldsymbol{x}_{tot}^{0:K}) d\boldsymbol{x}_{tot}^{1:K}$. which is called the reverse process, defined as a Markov chain $p_{\boldsymbol{\theta}_{tot}}(\boldsymbol{x}_{tot}^{0:K}) := p(\boldsymbol{x}_{tot}^K) \prod_{k=1}^{K} p_{\boldsymbol{\theta}_{tot}}(\boldsymbol{x}_{tot}^{k-1}|\boldsymbol{x}_{tot}^k)$ with learned Gaussian distribution starting as $p(\boldsymbol{x}_{tot}^K) = \mathcal{N}(\boldsymbol{0}, \boldsymbol{I}) \in \mathcal{R}^{N \times d}$, where $x_{tot}$ is the generated data, $N$ is the number of agent, $d$ is data dimension, $K$ is the diffusion steps. After $p_{\boldsymbol{\theta}_{tot}}(\boldsymbol{x}_{tot}^0)$ is learned to model ground truth distribution, if there exists a joint individual distribution function $[p_{\boldsymbol{\theta}_i}(\boldsymbol{x}_i^0) := \int p_{\boldsymbol{\theta}_i}(\boldsymbol{x}_i^{0:K}) d\boldsymbol{x}_i^{1:K}]_{i=1}^N$, where $\boldsymbol{x}_i^k \in \mathcal{R}^d$ is the data generated by agent $i$, $\boldsymbol{x}_i^K \sim \mathcal{N}(\boldsymbol{0}, \boldsymbol{I})$, such that the following conditions are satisfied.*

$$\prod_{i=1}^{N} p_{\boldsymbol{\theta}_i}(\boldsymbol{x}_i^0) = p_{\boldsymbol{\theta}_{tot}}(\boldsymbol{x}_{tot}^0) \quad \boldsymbol{\theta}_i \subset \boldsymbol{\theta}_{tot} \tag{3}$$

*It indicates that the collection of generated samples $\boldsymbol{x}_i^0$, identically distributed as $\boldsymbol{x}_{tot}^0$. We can state that $[p_{\boldsymbol{\theta}_i}(\boldsymbol{x}_i^0)]_{i=1}^N$ satisfy IGD for $p_{\boldsymbol{\theta}_{tot}}(\boldsymbol{x}_{tot}^0)$ and the diffusion model $p_{\boldsymbol{\theta}_{tot}}(\boldsymbol{x}_{tot}^0)$ is generatively factorized by diffusion models $[p_{\boldsymbol{\theta}_i}(\boldsymbol{x}_i)]_{i=1}^N$.*

The IGD principle is a generalization of the popular Individual-Global-Max (IGM) principle if the diffusion process generates deterministic actions exactly. Let's take an optimal discrete action generation (OAG) case as an example. If we view the generated data $x_{tot}$ as the optimal joint action $\bar{u}_{tot} = \arg max_{u_{tot}} Q_{tot}(\tau_{tot}, u_{tot})$, and each $x_i$ as the optimal local action $\bar{u}_i = \arg max_{u_i} Q_i(\tau_i, u_i)$. The IGD principle requires that $\prod_{i=1}^{N} p(x_i) = p(x_{tot})$. For the OAG case, the IGD principle requires that

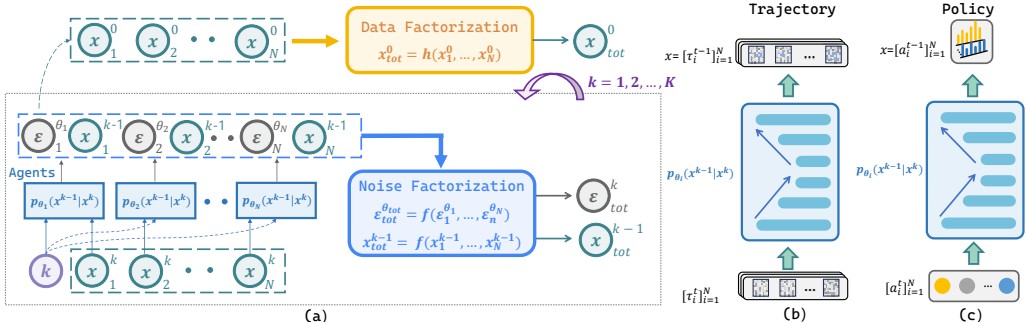

Figure 2: **DoF Overview:** (a) Diffusion Factorization: in each diffusion (forward and backward) step $1 \leq k \leq K$, the noise factorization function $f$ is used to factorize the noises $\epsilon_i^k$ and intermediate data $x_i^{k-1}$. In the last backward step $k = 0$, we apply the data factorization function $h$ to model the complex relationship among data generated by each agent. (b) DoF Trajectory Agent: generating trajectory for planning. (c) DoF Policy Agent: generating actions for execution.

$\prod_{i=1}^{N} p(\arg max_{u_i} Q_i(\tau_i, u_i)) = p(\arg max_{u_{tot}} Q_{tot}(\tau_{tot}, u_{tot}))$, which is an extension of the IGM principle. For more details, please refer to Theorems 4 and 5 in the Appendix B.

The IGD principle requires more than diffusion process factorization. It also requires that the diffusion process should match ground truth distribution by maximizing the likelihood of data via diffusion learning objective. After the diffusion processes satisfying IGD are learned, they can be used to generate data with desired properties through guidance.

The IGD principle is flexible. If we use diffusion to generate optimal policy, the IGD principle becomes the Individual-Global-Optimal principle (Zhang et al., 2021). For generating optimal risk-sensitive action, the IGD principle becomes the Risk-sensitive IGM principle (Shen et al., 2023).

## 4.3 DIFFUSION FACTORIZATION

The overview of the DoF method is depicted in Figure 2. In DoF, for the forward diffusion process, each agent samples a noise $\epsilon_i^k \in \mathbb{R}^d$, and they are combined to form the joint noise $\epsilon_{tot}^k \in \mathbb{R}^{N \times d}$ to noisify the data $x_{tot}$. In the backward diffusion process, the multi-agent system uses $\epsilon_{tot}^{\theta_{tot}}(\boldsymbol{x}_{tot}^k, k) \in \mathbb{R}^{N \times d}$ to generate $\boldsymbol{x}_{tot}^0$, where $\boldsymbol{x}_{tot}^k$ is the data generated in the $k$-th step. The forward joint noise $\epsilon_{tot}^k$, backward joint noise $\epsilon_{tot}^{\theta_{tot}}$, and joint data $\boldsymbol{x}_{tot}$ is factorized through using a noise factorization function $f$ and a data factorization $h$, where are described as follows.

$$\epsilon_{tot}^k = f(\epsilon_1^k, ..., \epsilon_N^k) \quad 0 \leq k \leq K \tag{4}$$

$$\epsilon_{tot}^{\theta_{tot}}(\boldsymbol{x}_{tot}^k, k) = f(\epsilon_1^{\theta_1}(\boldsymbol{x}_1^k, k), ..., \epsilon_N^{\theta_N}(\boldsymbol{x}_N^k, k)) \quad 0 \leq k \leq K \tag{5}$$

$$\boldsymbol{x}_{tot}^k = f(\boldsymbol{x}_1^k, ..., \boldsymbol{x}_N^k) \quad 1 \leq k \leq K \tag{6}$$

$$\boldsymbol{x}_{tot}^0 = h(\boldsymbol{x}_1^0, ..., \boldsymbol{x}_N^0) \tag{7}$$

The noise factorization function $f$ mixes the individual noise $\epsilon_i^k$ and data $\boldsymbol{x}_i^k$ of each agent to form joint noise $\epsilon_{tot}^k$ and joint data $\boldsymbol{x}_{tot}^k$. Thanks to $f$, after the noise model $\epsilon_{tot}^{\boldsymbol{\theta}_{tot}}(\boldsymbol{x}_{tot}^k, k)$, parameterized by $\boldsymbol{\theta}_{tot}$ is trained, it can be *factored into multiple small noise models* $\epsilon_i^{\boldsymbol{\theta}_i}(\boldsymbol{x}_i^k, k)$, each parameterized by $\boldsymbol{\theta}_i$. $\boldsymbol{\theta}_i \subset \boldsymbol{\theta}_{tot} \quad \forall i, \; \boldsymbol{\theta}_i \cap \boldsymbol{\theta}_j = \emptyset \; i \neq j$. During execution, agent $i$ uses the noise model $\epsilon_i^{\boldsymbol{\theta}_i}$ to generate data, its input complexity ($O(1)$) is only $1/N$ of the input complexity ($O(N)$) of the joint noise model $\epsilon^{\boldsymbol{\theta}_{tot}}$, where $N$ is the number of agents. In the last diffusion step, a data factorization function $h$ is applied to $[x_i^0]_{i=1}^N$ to form $x_{tot}^0$.

### 4.3.1 NOISE FACTORIZATION FUNCTION $f$

Noise factorization function $f$ factorizes the joint noise $\epsilon_{tot}$ into multiple individual noises $\epsilon_i$. In this work, we consider two noise factorization functions: Concat and WConcat.

**Concat** uses the concatenation function $\oplus$ as $f$. Concat assumes that the noise can be decomposed by dividing them according to data dimension. For example, given a noise $\epsilon_{tot} \sim \mathcal{N}(\boldsymbol{\mu}, \boldsymbol{\theta}) \in \mathbb{R}^{d \times N}$,

the function $f$ can factorize it into $[\epsilon_i]_{i=1}^N$, where $[\epsilon_i] \in \mathbb{R}^d$, and $\epsilon_i = \epsilon_{tot}[(i-1) \times d : i \times d]$. $\epsilon_i$ consists of the elements from the $(i-1) \times d$-th dimension to the $i \times d - 1$th dimension of $\epsilon_{tot}$.

In diffusion probability models, the noise $\epsilon_{tot}$ must be a Gaussian Noise with diagonal covariance. According to statistics, a concatenation of diagonal covariance Gaussian noises is still a Gaussian noise. So we can use $f$ to concatenate diagonal covariance Gaussian $[\epsilon_i]_{i=1}^N$ into $\epsilon_{tot}$. After training, each agent $i$ can use $\epsilon_i^{\theta_i}(x_i^k, k)$ to generate data $x_i^0$, which can be used for making decisions. Albeit the Concat function is simple, we show in Theorem 1 that the Concat noise factorization function satisfied the IGD principle.

**Theorem 1.** *A multi-agent diffusion model* $p_{\theta_{tot}}(x_{tot}^0)$

$$p_{\theta_{tot}}(x_{tot}^0) := \int p_{\theta_{tot}}(x_{tot}^{0:K}) \, dx_{tot}^{1:K} \tag{8}$$

$$\epsilon_{tot}^k = \oplus[\epsilon_i^k]_{i=1}^N \quad \epsilon \in \mathcal{N}(\mu, \sigma) \quad 0 \le k \le K \tag{9}$$

$$x_{tot}^k = \oplus[x_i^k]_{i=1}^N \quad 0 \le k \le K \tag{10}$$

$$\epsilon_{tot}^{\theta_{tot}}(x_{tot}^k, k) = \oplus[\epsilon_i^{\theta_i}(x_i^k, k)]_{i=1}^N \tag{11}$$

*is generatively factorized by* $[p_{\theta_i}(x_i)]_{i=1}^N$. *The noise ($\epsilon_{tot}$ and $\epsilon_i$) and the transition probability ($p_{\theta_{tot}}(x_{tot}^{k-1}|x_{tot}^k)$ and $p_{\theta_i}(x_i^{k-1}|x_i^k)$) follow diagonal Gaussian distributions. $\oplus$ is the Concat function. $p_{\theta_i}(x_i^0) := \int p_{\theta_i}(x_i^{0:K}) \, dx_i^{1:K}$. $\epsilon_i^t$ is the noise added during the forward process. $\epsilon_{\theta_i}(x_i^k, k)$ is used for the denoising process to predict the source noise $\epsilon_i^0 \sim \mathcal{N}(0, I)$ that determines $x_i^k$ from $x_i^0$.*

**WConcat** is a weighted version of Concat. It assigns an agent-specific weight $k_i$ to $\epsilon_i^{\theta_i}$. We show in Theorem 2 that WConcat satisfies the IGD principle. Please refer to Appendix B for proofs.

Algorithm 1 and Algorithm 2 present the pseudocode for the centralized training and decentralized execution phases of the DoF algorithm, respectively.

---

**Algorithm 1** Centralized Training

1: **repeat**
2:    $\mathbf{x}_{tot}^0 \sim q(\mathbf{x}_{tot})$ *(Sample global data)*
3:    $k \sim \text{Uniform}(\{1, \ldots, K\})$ *(Diffusion step)*
4:    $\epsilon \sim \mathcal{N}(\mathbf{0}, \mathbf{I}) \in \mathbb{R}^{d \times N}$ *(Sample global noise)*
5:    $\mathbf{x}_{tot}^k = \sqrt{\overline{\alpha}^k} \mathbf{x}_{tot}^{k-1} + \sqrt{1 - \overline{\alpha}^k} \epsilon$
6:    $\mathbf{x}_i^k = \mathbf{x}_{tot}^k[(i-1) \times d : i \times d], i \in [1, \ldots, N]$
7:    $\epsilon_{tot} = f(\epsilon_{\theta_1}^1(\mathbf{x}_1^k, k), \ldots, \epsilon_{\theta_N}^N(\mathbf{x}_N^k, k))$
8:    Take gradient descent step on:

$$\nabla_\theta \|\epsilon - \epsilon_{tot}\|^2$$

9: **until** convergence

---

**Algorithm 2** Decentralized Execution

1: $\mathbf{x}_i^K \sim \mathcal{N}(\mathbf{0}, \mathbf{I})$ *(Initialize for each agent $i$)*
2: **for** $k = K, \ldots, 1$ **do**
3:    $\epsilon_\theta^i(\mathbf{x}_i^k, k)$ *(Noise prediction by each agent $i$)*
4:    Update state for each agent $i$:

$$\mathbf{x}_i^{k-1} = \frac{1}{\sqrt{\alpha_k}} \left( \mathbf{x}_i^k - \frac{1 - \alpha_k}{\sqrt{1 - \overline{\alpha}_k}} \epsilon_\theta^i(\mathbf{x}_i^k, k) \right) + \sigma_k \mathbf{z},$$

   where $\mathbf{z} \sim \mathcal{N}(\mathbf{0}, \mathbf{I})$ if $k > 1$, else $\mathbf{z} = \mathbf{0}$.
5: **end for**
6: **return** $\mathbf{x}_i^0$ *(Final trajectory or action for each agent $i$)*

---

### 4.3.2 DATA FACTORIZATION FUNCTION $h$

The noise factorization function $f$ is used to learn factored diffusion processes. However, the modeling power of $f$ adopted in this work is limited. Thus, the generated $x_{tot}^0$ may not match closely as the real data. To improve the generation quality of $x_{tot}$, the data factorization function $h$ is used to mix $[x_i^0]_{i=1}^N$ to make $x_{tot}^0$ match real data closely.

The data factorization $h$ could model more powerful data relationships than the noise factorization function $f$. For example, if we consider each diffusion process generates individual Q value $x_i^0 = Q_i$, and $x_{tot}$ as the joint Q value function $Q_{tot}$, then $h$ can be *viewed as a value factorization function* (Rashid et al., 2020b) that modeling the relationship $Q_{tot} = h(Q_1, ..., Q_N)$. Besides Concat and WConcat, we explore the use of value factorization functions and their variants as $h$.

### 4.4 DoF AGENTS

DoF can be used for generating a trajectory for planning, and it can also be used for generating actions for execution. To demonstrate the flexibility of DoF, we implement two agents based on agents of

Decision Diffuser and DiffusionQL, respectively. When DoF is used for generating trajectories or actions, we call these methods **DoF-Trajectory** or **DoF-Policy**, respectively.

**DoF-Trajectory** use observation history as the data for diffusion. The clean data $x_{tot}^{t,0}$ used by the centralized diffusion process is defined as.

$$\boldsymbol{x}_{tot}^{t,0} := [\boldsymbol{o}_{tot}^{t}, \boldsymbol{o}_{tot}^{t+1}, \ldots, \boldsymbol{o}_{tot}^{t+H-1}]^0 \tag{12}$$

where $t$ is the time step of a MARL trajectory, $o_{tot}^t$ is the aggregated observations at $t$.

For each decentralized diffusion process $i$, the data $\boldsymbol{x}_i^{t,0} = [\boldsymbol{o}_i^t, \boldsymbol{o}_i^{t+1}, \ldots, \boldsymbol{o}_i^{t+H-1}]^0$ is used during diffusion, where $o_i^t$ is the observation of agent $i$ at $t$. Following (Ajay et al., 2023), we derive a policy from $\boldsymbol{x}_i^{0,t}$ by using an inverse dynamics model to estimate actions, which is defined as $\boldsymbol{u}_i^t := D_\phi(\boldsymbol{x}_i^{t,0}, \boldsymbol{x}_i^{t+1,0})$, where $D$ determines actions based on $\boldsymbol{x}_i^{t,0}$ and $\boldsymbol{x}_i^{t+1,0}$. Please refer the details of the agents in Appendix C.3.

**DoF-Policy** use continuous action as the data for diffusion. The data $x_{tot}^{t,0}$ is the action $u_{tot}^t$ for the multi-agent system. After the centralized diffusion process is trained, each agent $i$ uses a factored diffusion process, parameterized by $\boldsymbol{\theta}_i$ to generate its action $u_i^t$. Please refer the details of the agents in Appendix C.4.

For diffusion process $i$, we use condition $y_i$ to guide generated data toward desired properties. In cooperative MARL, a high-return value $R$ suggests cooperative behaviors. Thus, $R$ is included in $y_i$ to guide the diffusion process to generate high-return data. Further, the local observation history $\tau_i$ of agent $i$ is included in the condition $y_i$ to make the generated data align with $\tau_i$. For MADIFF, condition $y_i$ includes $R$ and $\tau_{tot}^i$, where $\tau_{tot}^i = (z_1, ..., \tau_i, ..., z_n)$, $z_1, ...z_n$ are random noises.

## 5 EVALUATION

In this section, we evaluate (1) the ability to generate data that match ground truth by comparing DoF against two diffusion-based MARL methods, (2) the importance of satisfying the IGD principle, (3) the ability to learn effective MARL policies from offline data, (4) the scalability of diffusion-based MARL. We justify the use of noise factorization and data factorization functions. In addition, we study the impact of different diffusion methods (Appendix D.5), demonstrate the ability of DoF to generate novel behaviors that satisfy multiple constraints (Appendix D.6), justify the use of condition information (Appendix D.7).

Unless otherwise specified, WConcat is employed as both the noise factorization function and the data factorization function. In default, the results for DoF-Trajectory are reported. Each experiment is repeated with five different seeds. Please refer to the Appendix D for details.

### 5.1 ILLUSTRATIVE EXAMPLES

The ability to generate data that match the ground truth distribution is the core ability of diffusion-based MARL methods. The studied diffusion-based MARL approaches are DoF, MADIFF, and Independent Diffusion (ID). The agent of DoF is the same as that of ID.

We evaluate the algorithms on three multi-agent cooperation tasks: (a) A matrix game generating two dimensional data, (b) A Landmark covering game, and (c) Q value generating Game. We demonstrate the superiority of the generation ability of DoF thanks to the IGD principle and factorization functions.

#### 5.1.1 A MATRIX GAME GENERATING TWO DIMENSIONAL DATA

The ground truth data consists of four two-dimension Gaussian distributed data. Their mean-value located in the top-left, top-right, bottom-left, and bottom-right of a data plane, and their variance are the same. The probability for generating the four Gaussian are 0.5, 0.2, 0.2, and 0.1, respectively.

Each agent is responsible for generating one dimension of the data. The closer the method mimics the ground truth, the better the algorithm. The scatter plot of the generated data from each method is depicted in Figure 3. The distribution of the data across quadrants is depicted in red in the center of the graph.

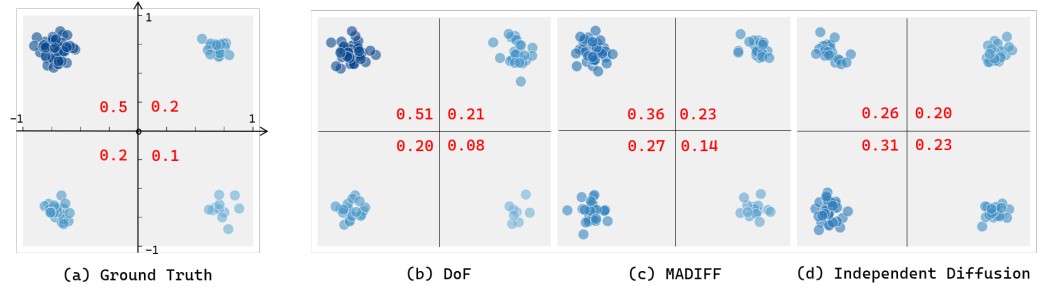

Figure 3: Generating two dimension data: (a) Ground Truth, (b) DoF, (c) MADIFF, (d) Independent Diffusion

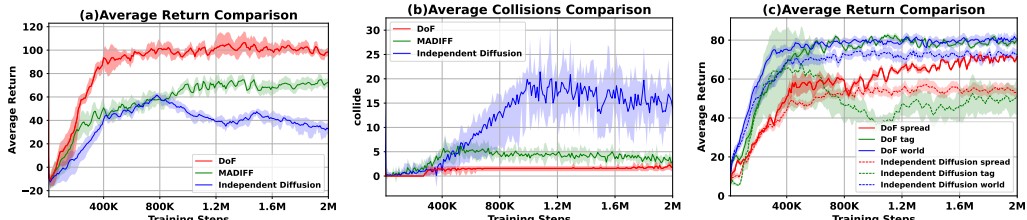

Figure 4: (a) Return and (b) Collisions for Trajectory Generation, (c) Return for Policy Generation.

Table 1: Payoff Matrix Games and Reconstructed Value Functions

| $u_2$ / $u_1$ | A | B |
|---|---|---|
| A | 1.0 | 0.0 |
| B | 18.0 | 1.0 |

(a) Game Payoff Matrix 1

| Q2 / Q1 | A | B |
|---|---|---|
| A | 0.9 | 0.0 |
| B | 17.9 | 1.2 |

(b) DoF

| $u_2$ / $u_1$ | A | B |
|---|---|---|
| A | 4.0 | 0.0 |
| B | 14.0 | 2.0 |

(c) Game Payoff Matrix 2

| Q2 / Q1 | A | B |
|---|---|---|
| A | 4.0 | 0.0 |
| B | 13.9 | 2.1 |

(d) DoF

As we can observe from Figure 3, DoF performs better than MADIFF and ID both visually and quantitatively. The data generated by DoF aligns most closely with the ground truth in both the scatter and probability distributions, with MADIFF performing the second. ID uses the same agent as DoF, but without using noise factorization and data factorization. ID does not satisfy the IGD principle. ID learns 2 separated diffusion processes, which cannot jointly model the data distribution well. MADIFF does not satisfy the IGD principle. As discussed in Section 4.4 and Appendix A.3, due to lack of factorization, each MADIFF agent uses a noisy condition $y_i$ to guide diffusion. It does not generate data match the ground truth data well.

### 5.1.2 LANDMARK COVERING GAME

In the landmark covering game, the motivating example, the trajectories of each method are plotted in Figure 1. It shows that DoF can generate trajectories similar to ground truth, whereas others do not. Figure 4(a) shows that more power modeling ability of DoF lead to higher rewards than others. And Figure 4(b) shows that the modeling ability of DoF lead to less collisions than others.

Further, we study the the action generation cases for the MPE dataset. As it is demonstrate in Figure 4 (c), DoF outperforms ID, which does not satisfy the IGD principle, by a large margins. *This game and the above games demonstrate the ability of DoF to generate data that matches ground truth, and the importance of satisfying the IGD principle.*

### 5.1.3 GENERATING Q VALUES

The goal of the two-agent game is to reconstruct a one-step payoff matrix $Q_{tot}$ (joint Q value function) through two agents. Agent $i$ use the diffusion process to generate individual utility value $Q_i$ and they are mixed into $Q_{tot} = h(Q_1, Q_2)$, where $h$ is the data factorization function $h(Q_1, Q_2) = k_1 Q_1 + K_2 Q_2$. $0 < k_i < 1$ are modelled following QAtten (Yang et al., 2020). As it is depicted in Table 1, DoF can reconstruct the payoff matrix $Q_{tot}$ well (including the optimal policies). This demonstrates the flexibility of the DoF framework in generating different types of

Table 2: The Average Return of the StarCraft Multi-Agent Benchmark (SMAC) Scenarios

| Maps | Data | MABCQ | MACQL | MAICQ | MADT | MADIFF | DoF |
|---|---|---|---|---|---|---|---|
| **3m** | Good | 3.7±1.1 | 19.1±0.1 | 18.7±0.7 | 19.0±0.3 | 19.3±0.5 | **19.8±0.2** |
| | Medium | 4.0±1.0 | 13.7±0.3 | 13.9±0.8 | 15.8±0.5 | 16.4±2.6 | **18.6±1.2** |
| | Poor | 3.4±1.0 | 4.2±0.1 | 8.4±2.6 | 4.2±0.1 | 10.3±6.1 | **10.9±1.1** |
| **8m** | Good | 4.8±0.6 | 5.4±0.9 | **19.6±0.2** | 18.5±0.4 | 18.9±1.1 | **19.6±0.3** |
| | Medium | 5.6±0.6 | 4.5±1.5 | 17.9±0.5 | 18.2±0.1 | 16.8±1.6 | **18.6±0.8** |
| | Poor | 3.6±0.8 | 3.5±1.0 | 11.2±1.3 | 4.8±0.1 | 9.8±0.9 | **12.0±1.2** |
| **5m_vs_6m** | Good | 2.4±0.4 | 7.4±0.6 | 11.0±0.6 | 16.8±0.1 | 16.5±2.8 | **17.7±1.1** |
| | Medium | 3.8±0.5 | 8.1±0.2 | 10.6±0.6 | 16.1±0.2 | 15.2±2.6 | **16.2±0.9** |
| | Poor | 3.3±0.5 | 6.8±0.1 | 6.6±0.2 | 7.6±0.3 | 8.9±1.3 | **10.8±0.3** |
| **2s3z** | Good | 7.7±0.9 | 17.4±0.3 | 18.3±0.2 | 18.1±0.1 | 15.9±1.2 | **18.5±0.8** |
| | Medium | 7.6±0.7 | 15.6±0.4 | 17.0±0.1 | 15.1±0.2 | 15.6±0.3 | **18.1±0.9** |
| | Poor | 6.6±0.2 | 8.4±0.8 | 9.9±0.6 | 8.9±0.3 | 8.5±1.3 | **10.0±1.1** |
| **3s5z_vs_3s6z** | Good | 5.9±0.3 | 7.8±0.5 | **13.5±0.6** | 12.8±0.2 | 7.1±1.5 | 12.8±0.8 |
| | Medium | 6.5±0.5 | 8.5±0.6 | 11.5±0.2 | 11.6±0.3 | 5.7±0.6 | **11.9±0.7** |
| | Poor | 6.1±0.6 | 5.9±0.4 | **7.9±0.2** | 5.6±0.3 | 4.7±0.6 | 7.5±0.2 |
| **2c_vs_64zg** | Good | 10.1±0.2 | 12.9±0.2 | 14.2±0.3 | 13.8±0.3 | 14.7±2.2 | **16.1±0.8** |
| | Medium | 9.9±0.2 | 11.6±0.1 | 12.0±0.1 | 11.8±0.2 | 12.8±1.2 | **13.9±0.9** |
| | Poor | 9.0±0.2 | 10.2±0.1 | 9.8±0.3 | 10.1±0.5 | 10.8±1.1 | **11.5±1.1** |

data. Moreover, this demonstrates that DoF can promote agent coordination through monotonicity among individual generated content $x_i$ and joint generated content $x_{tot}$.

## 5.2 COMPARISON STUDY

We evaluate the ability of DoF to learn effective MARL policies on the SMAC (Samvelyan et al., 2019), SMACv2 (Ellis et al., 2023), MPE (Lowe et al., 2017b), and MA-Mujoco (de Witt et al., 2020) environments against seven multi-agent algorithms.

The seven algorithms used for comparison are from three categories: (I) Offline MARL: MABCQ (Jiang & Lu, 2021), MACQL (Kumar et al., 2020), MAICQ (Yang et al., 2021), OMAR (Pan et al., 2022), and MA-TD3-BC (Fujimoto & Gu, 2021). (II) Transformer-based offline MARL: MADT (Meng et al., 2023). (III) Diffusion-based offline MARL: MADIFF (Zhu et al., 2024).

### 5.2.1 SMAC AND SMACV2

As shown in Table 2, DoF *achieves the best performance across most of the datasets*. MABCQ perform poorly. MACQL and MAICQ achieve good results on some good datasets but failed on moderate and poor datasets. Compared to these value-based algorithm, through using diffusion to generate trajectories, DoF does not learn value functions thus does not suffer from the challenges of value function approximation and off-policy learning. MADIFF, a diffusion-based approach, is able to model trajectory distributions and consider cooperation in some scenarios. However, it performs poorly in heterogeneous environments (e.g., 3s5z_vs_3s6z). DoF performs better than it thanks to the use of noise and data factorization functions. See Appendix D.3.1 for details and the win rate metric.

SMACv2 improves SMAC with more stochasticity. Its contents are procedurally generated with heterogeneous agents. In the Appendix D.3.2, Table 10 shows that DoF achieves the best results.

### 5.2.2 MULTI-AGENT PARTICLE ENVIRONMENTS (MPE) AND MULTI-AGENT MUJOCO

For MPE, the results are shown in Table 11 in the Appendix. MADIFF and OMAR perform the second and the third in most cases. MADIFF performs well in good-quality datasets but under-perform in low-quality datasets. DoF demonstrates *the best performance across various settings* thanks to the power modeling ability of diffusion and the effective collaborative strategies learned through noise factorization and data factorization functions.

In the Appendix D.3.4, Table 13 presents the HalfCheetah results from MA-MuJoCo. DoF performs best on the Medium dataset and ranks second on the Good and Poor datasets.

Table 3: Scalability Experiment: Comparison of DoF and MADIFF for Different Numbers of Agents

| Metric | Method | 4 Agents | 8 Agents | 16 Agents | 32 Agents | 64 Agents |
|---|---|---|---|---|---|---|
| GPU Memory (MB) | DoF | 1691 | 2123 | 2831 | 4322 | 5924 |
| | MADIFF | 3121 | 5387 | 8412 | 14981 | 21862 |
| Inference Time Cost (s) | DoF | 8.2 | 11.3 | 14.9 | 18.1 | 24.3 |
| | MADIFF | 12.9 | 16.5 | 23.9 | 31.5 | OOM |
| Reward | DoF | 60.12 | 75.91 | 120.31 | 154.62 | 210.42 |
| | MADIFF | 63.78 | 70.42 | 113.49 | 148.34 | OOM |

Table 4: DoF with different Noise Factorization Function $f$

| Maps | Dataset | Decentralized | | | Centralized | | MADIFF | DoF+MADIFF |
|---|---|---|---|---|---|---|---|---|
| | | Concat | WConcat | Dec-Atten | QMix | Atten | | |
| **3m** | Good | 19.7±0.6 | 19.8±0.5 | 4.3±2.3 | 3.8±1.3 | 19.8±0.4 | 19.3±0.5 | 19.7±0.4 |
| | Medium | 17.8±2.1 | 18.0±1.0 | 4.5±1.8 | 4.2±1.5 | 18.0±1.4 | 16.4±2.6 | 18.2±1.1 |
| | Poor | 10.6±1.6 | 11.4±0.7 | 3.2±1.5 | 3.5±1.4 | 11.3±1.3 | 10.3±1.5 | 10.8±1.2 |
| **5m_vs_6m** | Good | 16.7±1.4 | 17.0±0.8 | 3.6±1.5 | 4.1±1.2 | 17.1±0.8 | 16.5±2.8 | 16.7±1.2 |
| | Medium | 15.6±1.1 | 15.9±1.2 | 2.5±1.6 | 2.9±1.4 | 15.9±0.6 | 15.2±2.6 | 15.7±0.9 |
| | Poor | 9.8±1.1 | 10.7±0.8 | 2.9±1.4 | 2.3±1.1 | 10.2±0.7 | 8.9±1.3 | 10.0±0.8 |

## 5.3 SCALABILITY AND ABLATION STUDY

**Scalability Evaluation** We evaluate DoF and MADIFF in a customized environment developed based on MAgent (Zheng et al., 2018) with increasing number of agents. The experimental results are depicted in Table 3. It shows that through diffusion factorization, DoF achieves better scalability than MADIFF. When the number of agents reaches 64, MADIFF encounters out-of-memory (OOM) error, whereas DoF does not. We also conducted scalability evaluation on network parameters, inference time, and rewards. Please refer to Appendix D.3.5 for details.

**Noise Factorization function** $f$. We study the impact of different $f$, which can be categorized into decentralized and centralized execution functions. The experimental results are depicted in Table 4. Additional comparisons of Noise Factorization Function $f$ on more complex maps are provided in Appendix D.4. Please refer to Appendix D.4 for more details.

For decentralized functions, after the noise model $\epsilon_{tot}^{\theta_{tot}}$ is learned, it can be factored into multiple noise models $\epsilon_i^{\theta_i}$. For centralized functions, the noise model cannot be factored, and it should be executed centrally. For decentralized functions, The WConcat function performs better than Concat. For Dec-Atten, it is trained using an attention mechanism. During decentralized execution, the weights of $f$ considering other agents are omitted, which causes its poor performance. For centralized functions, the Atten function based on the attention mechanism performs the best. The QMix function performs poorly. This is due to the fact that through using QMix, the resulting noise may no longer be Gaussian noise, which is required for diffusion. Please refer to Appendix C.1 for details of $f$.

To demonstrate the flexibility of DoF, we introduce $f$ into MADIFF by replacing the DoF-trajectory agent with the MADIFF agent. The new method, DoF+MADIFF, performs better than MADIFF.

**Data factorization function** $h$. We study the impact of three functions: Concat, WConcat, and Atten. The results are depicted in Appendix Table 18. The experimental results show that Weight-Concat performs better than Concat but slightly weaker than Atten.

## 6 CONCLUSION

For diffusion-based multi-agent reinforcement learning (MARL), we extend the Individual-Global-Max (IGM) principle to the Individual-Global-identically-Distributed (IGD) principle, requiring that outcomes from a centralized diffusion process match those from combined individual processes. By satisfying this principle, after a diffusion model is learned, it can be factorized into multiple small diffusion models. We propose DoF, which employs the noise factorization function to decompose a joint noise into individual noises, and use the data factorization function to model data relationships. We prove that these noise factorization functions satisfy the IGD principle. Experiments on multiple benchmarks demonstrate DoF's effectiveness.

## ACKNOWLEDGEMENT

This work was partially supported by the Fundamental Research Funds for the Central Universities (No. 20720230033); by the National Natural Science Foundation of China (No. 62401225); by PDL (2022-PDL-12); by Xiaomi Young Talents Program, by the Fujian Provincial Natural Science Foundation of China (No. 2024J01115); by Natural Science Foundation of Xiamen, China (No. 3502Z202472018); the Jimei University Scientific Research Start-up Funding Project (No.ZQ2024034). We would like to thank the anonymous reviewers for their valuable suggestions.

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

# Appendix

## A  BACKGROUND AND COMPARING RELATED WORK

### A.1  DEC-POMDPS

In this study, we investigate cooperative multi-agent reinforcement learning scenarios, which can be effectively modeled using Decentralized Partially Observable Markov Decision Processes (Dec-POMDPs) (Oliehoek & Amato, 2016), a framework that handles environments where multiple agents must make coordinated decisions based on partial observations and incomplete information. A Dec-POMDP is formally represented by the tuple $G = \langle \mathcal{S}, \{\mathcal{U}_i\}_{i=1}^N, P, r, \{\mathcal{O}_i\}_{i=1}^N, \{\sigma_i\}_{i=1}^N, N, \gamma \rangle$ for $N$ agents.

Here, $\mathcal{S}$ represents a finite set of states, encapsulating all possible environmental configurations. Each agent $i$ interacts with the environment using a set of discrete actions $\mathcal{U}_i$. At any discrete time $t$, the joint action of all agents is $\mathbf{u}^t \in \mathcal{U}^N = \mathcal{U}_1 \times \ldots \times \mathcal{U}_N$, leading to a state transition to $s^{t+1} \in \mathcal{S}$ based on the transition function $s^{t+1} \sim P(\cdot \mid s^t, \mathbf{u}^t)$. Each agent receives a reward $r^t$ from this transition, critical for learning optimal policies.

Due to partial observability, each agent $i$ receives a local observation $o_i^t \sim \sigma_i(s^t)$, reflecting limited state information and complicating decision-making. The environment's partial observability is captured in each agent's local action-observation history $\tau_i = (\mathcal{O}_i \times \mathcal{U}_i)^*$, extending from the start to the time horizon $T$. The joint policy $\pi = \langle \pi_1, \ldots, \pi_N \rangle$ maps each agent's history to action probabilities.

### A.2  BACKGROUND ON DIFFUSION MODELS

Diffusion models have recently emerged as a powerful class of generative models, particularly successful in image and audio synthesis (Sohl-Dickstein et al., 2015; Ho et al., 2020; Song et al., 2021b). A diffusion model typically involves two main components: a forward (noising) process that gradually perturbs data, and a backward (denoising) process that learns to invert this perturbation to recover or generate new samples.

**Forward Noising Process.**  Given an initial data point $x^0 \sim q(x^0)$ from the empirical data distribution, the forward diffusion process adds noise step by step to produce a sequence of latent variables $\{x^1, x^2, \ldots, x^K\}$. Each transition follows a Gaussian:

$$q(x^k \mid x^{k-1}) = \mathcal{N}\big(x^k; \sqrt{\alpha^k}\, x^{k-1}, (1-\alpha^k)\mathbf{I}\big), \quad k = 1, \ldots, K, \tag{A.1}$$

where $\alpha^k \in (0,1)$ is a schedule parameter controlling the extent of noise injection at step $k$. After $K$ steps, $x^K$ becomes nearly an isotropic Gaussian variable if $\alpha^k$ is chosen such that $(1-\alpha^k)$ sums to a sufficiently large variance.

**Backward Denoising Process.**  To generate new samples, one starts from Gaussian noise $x^K \sim \mathcal{N}(0, \mathbf{I})$ and sequentially denoises via the reverse Markov chain:

$$p_{\boldsymbol{\theta}}(x^{k-1} \mid x^k) = \mathcal{N}\big(x^{k-1}; \boldsymbol{\mu}_{\boldsymbol{\theta}}(x^k, k), \boldsymbol{\Sigma}_{\boldsymbol{\theta}}(x^k, k)\big), \tag{A.2}$$

where $\boldsymbol{\theta}$ encapsulates the learned parameters. The mean $\boldsymbol{\mu}_{\boldsymbol{\theta}}(x^k, k)$ and variance $\boldsymbol{\Sigma}_{\boldsymbol{\theta}}(x^k, k)$ are trained (e.g., via variational bounds (Ho et al., 2020) or score matching (Song et al., 2021b)) to approximate the true reverse distribution $q(x^{k-1} \mid x^k)$ as closely as possible. After $K$ denoising steps, one obtains $x^0 \approx \tilde{x}^0$ as a generated sample.

The choice of Gaussian transitions in both forward and backward processes is not arbitrary but rather arises from practical and theoretical considerations. First, isotropic Gaussian noising is mathematically convenient, allowing closed-form derivations and efficient sampling. Second, the reverse-time formulation exploits properties of Gaussian distributions to make the training objective tractable. For more detailed derivations and discussions, see DDPM.

Hence, our use of Gaussian transitions in both the forward and backward stages of the Markov process aligns with the standard setup in DDPM and related diffusion-based generative models, rather than introducing a novel contribution in this work.

A.3 DIFFERENCES AMONG DoF, MADIFF, AND INDEPENDENT DIFFUSION

Here we discuss the differences among DoF, MADIFF and Independent Diffusion.

Table 5: Comparison of DoF, MADIFF, and Independent Diffusion Algorithms

| Methods | Decentrally executed | Input complexity | IGD | Num. of Processes |
|---|---|---|---|---|
| DoF ($f$=Concat) | Yes | $O(1)$ | Yes | $n$ |
| DoF ($f$=WConcat) | Yes | $O(1)$ | Yes | $n$ |
| MADIFF-D | Yes | $O(n)$ | No | 1 |
| MADIFF-C | No | $O(n)$ | *NA* | 1 |
| Independent Diffusion | Yes | $O(1)$ | No | $n$ |

In Table 5, *Decentrally Executed* indicates whether the trained agent can be executed decentrally without communication. *Input Complexity* is the input dimension for each trained backward diffusion step with respect to the number of agents $n$. All the methods listed in this table use U-net as its main neural network. Different from DoF and Independent Diffusion, the U-net used by MADIFF consists of an attention mechanism. *IGD* indicates whether following the IGD principle. *NA* is short for not applicable. For methods that cannot be decentrally executed, the IGD principle is not applicable for them. *Num. of Processes* indicates whether the number of factored diffusion processes.

$f$ is the noise factorization introduced in Section 4.3.1, and their details are described in Section C.1. In this table, we compare different noise factorization functions $f$ for DoF.

In DoF ($f$=Concat), the per-step backward-diffusion noise $\epsilon_{tot}^{\theta}$ is the concatenation of individual noises $\epsilon_i$. It is expressed as $\epsilon_{tot}^{\theta} = (\epsilon_1^{\theta}, ..., \epsilon_N^{\theta})$. In other words, $\epsilon_{tot}^{\theta}[(i-1) \times d : i \times d] = \epsilon_i^{\theta}$. For execution, each agent $i$ decentrally generates its noise $\epsilon_i^{\theta}$ using its noise model to generate $x_i$, which could be an action or a trajectory.

$f$=WConcat extends the basic $f$=Concat by introducing a learnable weight variable $k_i$, making the overall noise term $\epsilon_{tot}[(i-1) \times d : i \times d] = k_i \epsilon_i$. $f$=WConcat follows the paradigm of centralized training and decentralized execution. During execution, each agent generates $x_i$ based on the noise $\epsilon_i$ and the corresponding weight variable $k_i$, i.e., $k_i \epsilon_i$. $f$=WConcat satisfies the IGD principle.

$f$=Concat and $f$=Wconcat learn **n** diffusion processes. Each process is used by a separate agent $i$ to generate $x_i$ conditioned on observation $\tau_i$ and a return value $R$. For each agent, its input complexity is $O(1)$ with respect to the number of agents. The agents of $f$=Concat and $f$=WConcat can be executed decentrally.

MADIFF trains 1 diffusion process for all the agents. After the diffusion process is trained. Each agent uses the same diffusion process to generate data. There are two variants of MADIFF; they differ in the way the process is used for execution.

The centralized execution variant, MADIFF-C, conditions on aggregated observation-history $\tau_{tot} = (\tau_1, ..., \tau_n)$ and a return value $R$ to generate data $x_{tot}$. MADIFF-C can generate data that matches the original data distribution. If $R$ is high, MADIFF-C can sample high-return data. Its input complexity is $O(n)$.

The decentralized execution variant (**the default variant**), MADIFF-D conditions on $\tau_{tot}^i$ and a return value $R$ to generate data $x_i$. $\tau_{tot}^i = (z_1, ..., \tau_i, ..., z_n)$, where $\tau_i$ is the local observation-history of $i$, and $z_1, ...z_n$ are random noises. As agent $i$ can only observe $\tau_i$, except for the $i$-th element, all the other elements in $\tau_{tot}^i$ are filled with random noise. **Its signal-to-noise ratio is low, only** $1/n$**.** Thanks to the modeling ability of the diffusion model, MADIFF can still generate data. However, it is unclear how the generated data matches the original data distribution. Moreover, we show theoretically that MADIFF does not satisfy the IGD principle. The input complexity of MADIFF-D is $O(n)$. The input complexity is $n$ times higher than that for DoF($f$=Concat) and DoF ($f$=WConcat).

If we consider the combination of the diffusion model and inverse dynamics as the MADIFF-D agent network, then the condition $\tau_{tot}^i$ can be regarded as the input for this agent network. What we mean by random noise is that there is a large amount of noise $\tilde{x} \in \mathcal{N}(0, I)$ in the input $\tau_{tot}^i$ for the

MADIFF-D agent. We have borrowed the Formula 8 of MADIFF as follows, where the left part is denoted as $X$ and the right part as $X^0$.

$$X = \begin{bmatrix} \tilde{x}_{K,t}^0, & \cdots, & \tilde{x}_{K,t+H-1}^0 \\ \vdots & \cdots & \vdots \\ O_t^i, & \cdots, & \tilde{x}_{K,t+H-1}^i \\ \vdots & \cdots & \vdots \\ \tilde{x}_{K,t}^N, & \cdots, & \tilde{x}_{K,t+H-1}^N \end{bmatrix} \xRightarrow{\text{Iterative } K \text{ diffusion steps}} X^0 = \begin{bmatrix} \hat{O}_t^0, & \cdots, & \hat{O}_{t+H-1}^0 \\ \vdots & \cdots & \vdots \\ O_t^i, & \cdots, & \hat{O}_{t+H-1}^i \\ \vdots & \cdots & \vdots \\ \hat{O}_t^N, & \cdots, & \hat{O}_{t+H-1}^N \end{bmatrix},$$

(A.3)

The input $\tau_{tot}^i$ for the MADIFF-D agent network corresponds to the first column of $X$. In $\tau_{tot}^i$, except for the $i$-th entry $O_t^i$, the other $N-1$ entries consist of random noise $\tilde{x} \in \mathcal{N}(0, I)$. Therefore, MADIFF-D contains more noise compared to DoF, which limits its ability to fully utilize the learned coordination. Once the input $\tau_{tot}^i$ is provided to the agent network, the left column of $X^0$ (the condition) is improved through multiple diffusion steps (the teammate modeling mechanism of MADIFF), as shown in the first column of $X^0$. However, because MADIFF-D's condition consists of significant noise, its performance is unsatisfactory despite the use of the attention mechanism inside the diffusion process during training.

The random noise we refer to is the noise present in the input to the MADIFF-D agent network, not the noise generated during the diffusion process within the network itself. This excess noise in the input reduces the effectiveness of the coordination learned through diffusion. In contrast, MADIFF-C performs better than MADIFF-D as it suffers from less input noise. However, MADIFF-C is not scalable and faces limitations such as the curse of dimensionality in Multi-Agent Reinforcement Learning (MARL).

In Independent Diffusion (ID), each diffusion process is trained to mimic the behavior of all the agents rather than a specific agent due to the lack of noise and data factorization. For example, for a two-agent scenario, process 1 learns the behaviors of agents 1 and 2, and so does process 2. With the use of noise and data factorization, the diffusion process $i$ will learn the behaviors of agent $i$ rather than that of all the agents. As ID does not tried to model the collective behaviors, it does not satisfy the IGD principle.

DOM2 (Li et al., 2023) adopts an actor-critic MARL approach. Each actor is developed base on DiffuionQL (Wang et al., 2023). The corresponding critic (the Q function) of each actor are learned independently without considering others. As the critic does not fully taken into account the other agents, the learned agents do not fully consider cooperation.

# B  THE INDIVIDUAL-GLOBAL-IDENTICALLY-DISTRIBUTED PRINCIPLE

**Definition 1** (IGD). *For a joint total distribution $p_{\boldsymbol{\theta}_{tot}}(\boldsymbol{x}_{tot}^0) \coloneqq \int p_{\boldsymbol{\theta}_{tot}}(\boldsymbol{x}_{tot}^{0:K}) d\boldsymbol{x}_{tot}^{1:K}$. which is called the reverse process, defined as a Markov chain $p_{\boldsymbol{\theta}_{tot}}(\boldsymbol{x}_{tot}^{0:K}) \coloneqq p(\boldsymbol{x}_{tot}^K) \prod_{k=1}^K p_{\boldsymbol{\theta}_{tot}}(\boldsymbol{x}_{tot}^{k-1}|\boldsymbol{x}_{tot}^k)$ with learned Gaussian distribution starting as $p(\boldsymbol{x}_{tot}^K) = \mathcal{N}(\boldsymbol{0}, \boldsymbol{I}) \in \mathcal{R}^{N \times d}$, where $x_{tot}$ is the generated data, $N$ is the number of agent, $d$ is data dimension, $K$ is the diffusion steps. After $p_{\boldsymbol{\theta}_{tot}}(\boldsymbol{x}_{tot}^0)$ is learned, if there exists a joint individual distribution functions $[p_{\boldsymbol{\theta}_i}(\boldsymbol{x}_i^0) \coloneqq \int p_{\boldsymbol{\theta}_i}(\boldsymbol{x}_i^{0:K}) d\boldsymbol{x}_i^{1:K}]_{i=1}^N$, where $\boldsymbol{x}_i \in \mathcal{R}^d$ is the generated data, $\boldsymbol{x}_i^K \sim \mathcal{N}(\boldsymbol{0}, \boldsymbol{I})$, such that the following conditions are satisfied.*

$$\prod_{i=1}^N p_{\boldsymbol{\theta}_i}(\boldsymbol{x}_i^0) = p_{\boldsymbol{\theta}_{tot}}(\boldsymbol{x}_{tot}^0) \quad \boldsymbol{\theta}_i \subset \boldsymbol{\theta}_{tot} \tag{B.4}$$

*It indicates that the collection of generated samples $\boldsymbol{x}_i^0$, identically distributed as $\boldsymbol{x}_{tot}^0$. We can state that $[p_{\boldsymbol{\theta}_i}(\boldsymbol{x}_i^0)]_{i=1}^N$ satisfy IGD for $p_{\boldsymbol{\theta}_{tot}}(\boldsymbol{x}_{tot}^0)$ and the diffusion model $p_{\boldsymbol{\theta}_{tot}}(\boldsymbol{x}_{tot}^0)$ is generatively factorized by diffusion models $[p_{\boldsymbol{\theta}_i}(\boldsymbol{x}_i)]_{i=1}^N$.*

**Theorem 1.** *A multi-agent diffusion model* $p_{\boldsymbol{\theta}_{tot}}(\boldsymbol{x}_{tot}^0)$

$$p_{\boldsymbol{\theta}_{tot}}(\boldsymbol{x}_{tot}^0) \coloneqq \int p_{\boldsymbol{\theta}_{tot}}(\boldsymbol{x}_{tot}^{0:K})\, d\boldsymbol{x}_{tot}^{1:K} \tag{B.5}$$

$$\boldsymbol{\epsilon}_{tot}^k = \oplus[\boldsymbol{\epsilon}_i^k]_{i=1}^N \quad \boldsymbol{\epsilon} \in \mathcal{N}(\mu, \sigma) \quad 0 \le k \le K \tag{B.6}$$

$$\boldsymbol{x}_{tot}^k = \oplus[\boldsymbol{x}_i^k]_{i=1}^N \quad 0 \le k \le K \tag{B.7}$$

$$\boldsymbol{\epsilon}_{tot}^{\boldsymbol{\theta}_{tot}}(\boldsymbol{x}_{tot}^k, k) = \oplus[\boldsymbol{\epsilon}_i^{\boldsymbol{\theta}_i}(\boldsymbol{x}_i^k, k)]_{i=1}^N \tag{B.8}$$

*is generatively factorized by* $[p_{\boldsymbol{\theta}_i}(\boldsymbol{x}_i)]_{i=1}^N$. *The noise* ($\epsilon_{tot}$ *and* $\epsilon_i$) *and the transition probability* ($p_{\boldsymbol{\theta}_{tot}}(\boldsymbol{x}_{tot}^{k-1}|\boldsymbol{x}_{tot}^k)$ *and* $p_{\boldsymbol{\theta}_i}(\boldsymbol{x}_i^{k-1}|\boldsymbol{x}_i^k)$) *follow diagonal Gaussian distributions.* $\oplus$ *is the Concat function.* $p_{\boldsymbol{\theta}_i}(\boldsymbol{x}_i^0) \coloneqq \int p_{\boldsymbol{\theta}_i}(\boldsymbol{x}_i^{0:K})\, d\boldsymbol{x}_i^{1:K}$. $\boldsymbol{\epsilon}_i^t$ *is the noise added during the forward process.* $\boldsymbol{\epsilon}_{\boldsymbol{\theta}_i}(\boldsymbol{x}_i^k, k)$ *is used for the denoising process to predict the source noise* $\boldsymbol{\epsilon}_i^0 \sim \mathcal{N}(0, I)$ *that determines* $\boldsymbol{x}_i^k$ *from* $\boldsymbol{x}_i^0$.

*Proof.* In the forward diffusion process, the global data $x_{tot}$ is modified as follows.

$$x_{tot}^k = \sqrt{\bar{\alpha}^k} x_{tot}^{k-1} + \sqrt{1 - \bar{\alpha}^k}\, \epsilon \tag{B.9}$$

$$\bar{\alpha}^k = \prod_{i=1}^k \alpha^k \tag{B.10}$$

$$\boldsymbol{x}_{tot}^k = \oplus[\boldsymbol{x}_i^k]_{i=1}^N \quad 0 \le k \le K \tag{B.11}$$

We can view the data $x_i^k$ as it is polluted by adding noise to $x_i^{k-1}$ according to (B.13), where $\alpha^k$ is a pre-specified hyper-parameter, $k$ is the diffusion step, $\boldsymbol{\epsilon}^{k-1} \sim \mathcal{N}(0, \mathbf{I})$.

$$x_i^k = x_{tot}^k[(i-1) \times d : i \times d] \quad i \in [1, ..N] \tag{B.12}$$

$$x_i^k = \sqrt{\bar{\alpha}^k} x_i^{k-1} + \sqrt{1 - \bar{\alpha}^k}\, \boldsymbol{\epsilon}^{k-1} \tag{B.13}$$

After the model is trained, the noise model $\epsilon_{tot}^{\theta_{tot}}(x_{tot}, k)$ is factorized into multiple noise models $\epsilon_i^{\boldsymbol{\theta}_i}(x_i, k)$, $\boldsymbol{\theta_i} \subset \boldsymbol{\theta_{tot}}$, $\boldsymbol{\theta_i} \cap \boldsymbol{\theta_j} = \emptyset$. $\boldsymbol{\theta_{tot}} = \oplus[\boldsymbol{\theta_i}]_{i=1}^N$. The noise model $\epsilon_i^{\boldsymbol{\theta}_i}(x_i, k)$ can be used for backward diffusion of $x_i$. For the backward diffusion step, the probability $p_{\boldsymbol{\theta_i}}(\boldsymbol{x}_i^{k-1}|\boldsymbol{x}_i^k)$ of generating $\boldsymbol{x}_i^k$ based on $\boldsymbol{x}_i^{k-1}$ is defined as a Gaussian distribution described in (B.14). The mean $\boldsymbol{\mu}_{\boldsymbol{\theta_i}}(\boldsymbol{x}_i^k, k)$ and the variance $\boldsymbol{\Sigma}(k)$ of the Gaussian distribution is defined in (B.15) and in (B.16). Formulas (B.15) and (B.16) are adapted from Luo (2022).

$$p_{\boldsymbol{\theta_i}}(\boldsymbol{x}_i^{k-1}|\boldsymbol{x}_i^k) = \mathcal{N}(\boldsymbol{x}_i^{k-1}; \boldsymbol{\mu}_{\boldsymbol{\theta_i}}(\boldsymbol{x}_i^k, k), \boldsymbol{\Sigma}(k)) \tag{B.14}$$

$$\boldsymbol{\mu}_{\boldsymbol{\theta_i}}(\boldsymbol{x}_i^k, k) = \frac{1}{\sqrt{\alpha^k}} \boldsymbol{x}_i^k - \frac{1 - \alpha^k}{\sqrt{1 - \bar{\alpha}^k}\sqrt{\alpha^k}} \boldsymbol{\epsilon}_i^{\boldsymbol{\theta_i}}(\boldsymbol{x}_i^k, k) \tag{B.15}$$

$$\boldsymbol{\Sigma}(k) = \frac{(1 - \alpha_t)(1 - \bar{\alpha}_{t-1})}{1 - \bar{\alpha}_t} \mathbf{I} \tag{B.16}$$

The backward probability $p_{\boldsymbol{\theta_i}}(\boldsymbol{x}_i^{k-1}|\boldsymbol{x}_i^k)$ depends on $\theta_i$ only. For the factorization function **Concat**, $\theta_i \cap \theta_j = \emptyset$ $i \ne j$. Thus, $p_{\boldsymbol{\theta_i}}(\boldsymbol{x}_i^{k-1}|\boldsymbol{x}_i^k)$ is independent from $p_{\boldsymbol{\theta_j}}(\boldsymbol{x}_j^{k-1}|\boldsymbol{x}_j^k)$. We can derive in equation (B.17) the joint conditional probability is equal to the product of two independent conditional probabilities.

$$p_{\boldsymbol{\theta_i} \cup \boldsymbol{\theta_j}}(\boldsymbol{x}_i^{k-1}, \boldsymbol{x}_j^{k-1}|\boldsymbol{x}_i^k, \boldsymbol{x}_j^k) = p_{\boldsymbol{\theta_i}}(\boldsymbol{x}_i^{k-1}|\boldsymbol{x}_i^k) \times p_{\boldsymbol{\theta_j}}(\boldsymbol{x}_j^{k-1}|\boldsymbol{x}_j^k) \quad i \ne j \tag{B.17}$$

The probability of backward denoising step of $\boldsymbol{x_{tot}}$ is written as follows.

$$p_{\boldsymbol{\theta_{tot}}}(\boldsymbol{x}_{tot}^{k-1}|\boldsymbol{x}_{tot}^k) = p_{\boldsymbol{\theta_1} \cup, ..., \cup \boldsymbol{\theta_N}}(\boldsymbol{x}_1^{k-1}, ..., \boldsymbol{x}_N^{k-1}|\boldsymbol{x}_1^k, ..., \boldsymbol{x}_N^k) \tag{B.18}$$

$$= p_{\boldsymbol{\theta_1}}(\boldsymbol{x}_1^{k-1}|\boldsymbol{x}_1^k) \times p_{\boldsymbol{\theta_2}}(\boldsymbol{x}_2^{k-1}|\boldsymbol{x}_2^k) \times ... p_{\boldsymbol{\theta_N}}(\boldsymbol{x}_N^{k-1}|\boldsymbol{x}_N^k) \tag{B.19}$$

$$= \prod_{i=1}^N p_{\boldsymbol{\theta_i}}(\boldsymbol{x}_i^{k-1}|\boldsymbol{x}_i^k) \tag{B.20}$$

The generation process of $\boldsymbol{x}_{tot}^0$, parameterized by $\boldsymbol{\theta}_{tot}$, can be written as a Markov chain as follows.

$$p_{\boldsymbol{\theta}_{tot}}(\boldsymbol{x}_{tot}^0) = \int p_{\boldsymbol{\theta}_{tot}}(\boldsymbol{x}_{tot}^{0:K}) d\boldsymbol{x}_{tot}^{1:K} \tag{B.21}$$

$$= p(\boldsymbol{x}_{tot}^K) \int p_{\boldsymbol{\theta}_{tot}}(\boldsymbol{x}_{tot}^{k-1}|\boldsymbol{x}_{tot}^k) d\boldsymbol{x}_{tot}^{1:K} \tag{B.22}$$

$$= \prod_{i=1}^{N} p(\boldsymbol{x}_i^K) \int p_{\boldsymbol{\theta}_{tot}}(\boldsymbol{x}_{tot}^{k-1}|\boldsymbol{x}_{tot}^k) d\boldsymbol{x}_{tot}^{1:K} \quad (\boldsymbol{x}_i^K \sim \mathcal{N}(0,\mathbf{I}),\ \boldsymbol{x}_{tot}^K \sim \mathcal{N}(0,\mathbf{I})) \tag{B.23}$$

$$= \prod_{i=1}^{N} p(\boldsymbol{x}_i^K) \int \prod_{i=1}^{N} p_{\boldsymbol{\theta}_i}(\boldsymbol{x}_i^{k-1}|\boldsymbol{x}_i^k) d\boldsymbol{x}_{tot}^{1:K} \tag{B.24}$$

$$= \prod_{i=1}^{N} p(\boldsymbol{x}_i^K) \int \prod_{i=1}^{N} p_{\boldsymbol{\theta}_i}(\boldsymbol{x}_i^{k-1}|\boldsymbol{x}_i^k) d\left[\boldsymbol{x}_1^{1:K}, \boldsymbol{x}_2^{1:K}, ..., x_N^{1:K}\right] \tag{B.25}$$

$$= \prod_{i=1}^{N} p(\boldsymbol{x}_i^K) \int \prod_{i=1}^{N} p_{\boldsymbol{\theta}_i}(\boldsymbol{x}_i^{k-1}|\boldsymbol{x}_i^k) d\boldsymbol{x}_1^{1:K} d\boldsymbol{x}_2^{1:K} ..., d\boldsymbol{x}_N^{1:K} \quad (x_i \cap x_j = \emptyset, i \neq j) \tag{B.26}$$

$$= p(\boldsymbol{x}_1^K) \int p_{\boldsymbol{\theta}_1}(\boldsymbol{x}_i^{k-1}|\boldsymbol{x}_i^k) d\boldsymbol{x}_1^{1:K} ... p(\boldsymbol{x}_N^K) \int p_{\boldsymbol{\theta}_N}(\boldsymbol{x}_N^{k-1}|\boldsymbol{x}_N^k) d\boldsymbol{x}_N^{1:K} \tag{B.27}$$

$$= p_{\boldsymbol{\theta}_1}(\boldsymbol{x}_1^0) p_{\boldsymbol{\theta}_2}(\boldsymbol{x}_2^0) ... p_{\boldsymbol{\theta}_N}(\boldsymbol{x}_N^0) \tag{B.28}$$

$$= \prod_{i=1}^{N} p_{\boldsymbol{\theta}_i}(\boldsymbol{x}_i^0) \tag{B.29}$$

We have shown that the probability $p_{\boldsymbol{\theta}_{tot}}(\boldsymbol{x}_{tot}^0)$ of centrally generating $\boldsymbol{x}_{tot}^0$ is equal to the product of probability $p_{\boldsymbol{\theta}_i}(\boldsymbol{x}_i^0)$ for generating $\boldsymbol{x}_i^0$ decentrally. $\square$

**Theorem 2.** *A multi-agent diffusion model $p_{\boldsymbol{\theta}_{tot}}(\boldsymbol{x}_{tot}^0)$*

$$p_{\boldsymbol{\theta}_{tot}}(\boldsymbol{x}_{tot}^0) := \int p_{\boldsymbol{\theta}_{tot}}(\boldsymbol{x}_{tot}^{0:K}) d\boldsymbol{x}_{tot}^{1:K} \tag{B.30}$$

$$\boldsymbol{\epsilon}_{tot}^k = \uplus[\boldsymbol{\epsilon}_i^k]_{i=1}^N \quad \boldsymbol{\epsilon} \in \mathcal{N}(\mu,\sigma) \quad 0 \leq k \leq K \tag{B.31}$$

$$\boldsymbol{x}_{tot}^k = \uplus[\boldsymbol{x}_i^k]_{i=1}^N \quad 0 \leq k \leq K \tag{B.32}$$

$$\boldsymbol{\epsilon}_{tot}^{\boldsymbol{\theta}_{tot}}(\boldsymbol{x}_{tot}^k, k) = \uplus[\boldsymbol{\epsilon}_i^{\boldsymbol{\theta}_i}(\boldsymbol{x}_i^k, k)]_{i=1}^N \tag{B.33}$$

$$\boldsymbol{\theta}_{tot} = \oplus[\boldsymbol{\theta}_i]_{i=1}^N \tag{B.34}$$

*is generatively factorized by $[p_{\boldsymbol{\theta}_i}(\boldsymbol{x}_i)]_{i=1}^N$. The noise ($\epsilon_{tot}$ and $\epsilon_i$) and the transition probability ($p_{\boldsymbol{\theta}_{tot}}(\boldsymbol{x}_{tot}^{k-1}|\boldsymbol{x}_{tot}^k)$ and $p_{\boldsymbol{\theta}_i}(\boldsymbol{x}_i^{k-1}|\boldsymbol{x}_i^k)$) follow diagonal Gaussian distributions. $\uplus$ is the WConcat function, and $\oplus$ is the Concat function. $p_{\boldsymbol{\theta}_i}(\boldsymbol{x}_i^0) := \int p_{\boldsymbol{\theta}_i}(\boldsymbol{x}_i^{0:K}) d\boldsymbol{x}_i^{1:K}$. $\boldsymbol{\epsilon}_i^t$ is the noise added during the forward process. $\boldsymbol{\epsilon}_{\boldsymbol{\theta}_i}(\boldsymbol{x}_i^k, k)$ is used for the denoising process to predict the source noise $\boldsymbol{\epsilon}_i^0 \sim \mathcal{N}(0, I)$ that determines $\boldsymbol{x}_i^k$ from $\boldsymbol{x}_i^0$.*

*Proof.* This theorem can be proved in the same way as the proof for Theorem 1 $\square$

**Theorem 3.** *The multi-agent diffusion model $p_{\theta_{tot}}(\boldsymbol{x}_{tot}^0)$ learned by MADIFF does not satisfy the IGD principle.*

*Proof.* The diffusion model learned by MADIFF is defined as follows.

$$p_{\theta_{tot}}(\boldsymbol{x}_{tot}^0) := \int p_{\theta_{tot}}(\boldsymbol{x}_{tot}^{0:K}) d\boldsymbol{x}_{tot}^{1:K} \tag{B.35}$$

After the model is trained, the diffusion model used by agent $i$ is $p_{\theta_i}(\boldsymbol{x}_i^0)$, it is the same as $p_{\boldsymbol{\theta}_{tot}}(\boldsymbol{x}_{tot}^0)$, parameterized by $\boldsymbol{\theta}_{tot}$. Thus, $\boldsymbol{\theta}_i = \boldsymbol{\theta}_{tot}$. It does not meet the factorization requirement of the IGD principle that $\boldsymbol{\theta}_i \subset \boldsymbol{\theta}_{tot}$ $\square$

**Theorem 4.** *IGD as a Generalization of the IGO Principle*

**The definition of the IGO Principle** (Zhang et al., 2021): For an optimal joint policy $\pi_{tot}^*(\mathbf{u_{tot}} \mid \tau_{\mathbf{tot}}) : \mathcal{T} \times \mathcal{U} \to [0, 1]$, where $\tau_{\mathbf{tot}} \in \mathcal{T}$ is a joint trajectory, if there exist individual optimal policies $[\pi_i^*(u_i \mid \tau_i) : \mathcal{T} \times \mathcal{U} \to [0, 1]]_{i=1}^N$, such that the following holds:

$$\pi_{tot}^*(\mathbf{u_{tot}} \mid \tau_{\mathbf{tot}}) = \prod_{i=1}^N \pi_i^*(u_i \mid \tau_i), \tag{B.36}$$

then we say that $[\pi_i]_{i=1}^N$ satisfy **IGO** for $\pi_{tot}$ under $\tau_{\mathbf{tot}}$.

The IGD principle requires that

$$\prod_{i=1}^N p_{\theta_i}(x_i^0) = p_{\theta_{tot}}(x_{tot}^0), \tag{B.37}$$

where $x_i^0$ is the generated data of agent $i$, and $x_{tot}^0$ is the generated data of the whole multi-agent system. Here, $p_{\theta_i}(x_i^0)$ is the probability of $x_i^0$, and $p_{\theta_{tot}}(x_{tot}^0)$ is the probability of $x_{tot}^0$.

We can use the IGD principle to generate multi-agent actions. Let us treat the generated data $x_i^0$ as $u_i$, where $u_i$ is the action taken by agent $i$. Then, the probability $p_{\theta_i}(u_i)$ becomes $\pi_i(u_i \mid \tau_i)$, where $\pi_i$ is the policy of agent $i$ and $\tau_i$ is the local observation. Further, let us treat the generated total data $\mathbf{x_{tot}^0}$ as $\mathbf{u_{tot}}$. The probability $p_{\theta_{\mathbf{tot}}}(\mathbf{u_{tot}})$ becomes $\pi_{tot}(\mathbf{u_{tot}} \mid \tau_{\mathbf{tot}})$, where $\pi_{tot}$ is the policy of the multi-agent system, and $\tau_{tot}$ is the aggregated observations. Then the IGD principle becomes the following formulas:

$$\prod_{i=1}^N p_{\theta_i}(u_i) = p_{\theta_{\mathbf{tot}}}(\mathbf{u_{tot}}), \tag{B.38}$$

$$\prod_{i=1}^N \pi_i(u_i \mid \tau_i) = \pi_{tot}(\mathbf{u_{tot}} \mid \tau_{\mathbf{tot}}). \tag{B.39}$$

The above formula requires that for any policy $\pi$, the total policy $\pi_{tot}$ is equal to the product of its per-agent policies. By substituting the optimal policy $\pi_i^*$ for the policy $\pi_i$ and the optimal total policy $\pi_{tot}^*$ for the policy $\pi_{tot}$, we obtain the following formula:

$$\prod_{i=1}^N \pi_i^*(u_i \mid \tau_i) = \pi_{tot}^*(\mathbf{u_{tot}} \mid \tau_{\mathbf{tot}}), \tag{B.40}$$

$$\pi_{tot}^*(\mathbf{u_{tot}} \mid \tau_{\mathbf{tot}}) = \prod_{i=1}^N \pi_i^*(u_i \mid \tau_i). \tag{B.41}$$

The formula is exactly the requirement of the IGO principle. Thus, we have shown that the IGD principle is a generalization of the IGO principle.

**Theorem 5.** *IGD as a Generalization of the IGM Principle.*

We have shown that the IGD principle is a generalization of the IGO principle. And the IGO paper Zhang et al. (2021) shows that the IGO principle is a generalizatioin of the IGM principle, thus the IGD principle is a generalization of the IGM principle.

**Theorem 6.** *IGD as a Generalization of the RIGM Principle.*

**The definition of the RIGM Principle** (Shen et al., 2023): Given a risk metric $\psi_\alpha$, a set of individual return distribution utilities $[Z_i(\tau_i, u_i)]_{i=1}^N$, and a joint state-action return distribution $Z_{tot}(\boldsymbol{\tau_{tot}}, \boldsymbol{u_{tot}})$, if the following conditions are satisfied:

$$\arg\max_{\mathbf{u}} \psi_\alpha\left(Z_{tot}(\tau_{\mathbf{tot}}, \mathbf{u_{tot}})\right) = \left[\arg\max_{u_1} \psi_\alpha\left(Z_1(\tau_1, u_1)\right), \ldots, \arg\max_{u_N} \psi_\alpha\left(Z_N(\tau_N, u_N)\right)\right], \tag{B.42}$$

where $\psi_\alpha : Z \times R \to R$ is a risk metric such as the VaR or a distorted risk measure, $\alpha$ is its risk level. Then, $[Z_i(\tau_i, u_i)]_{i=1}^N$ satisfy the RIGM principle with risk metric $\psi_\alpha$ for $Z_{jt}$ under under $\tau$. We can state that $Z_{tot}(\boldsymbol{\tau_{tot}}, \boldsymbol{u_{tot}})$ can be distributionally factorized by $[Z_i(\tau_i, u_i)]_{i=1}^N$ with risk metric $\psi_\alpha$.

Let's define probability functions $\pi_{tot}(\mathbf{u_{tot}} \mid \tau_{\mathbf{tot}})$ and $\pi_i(u_i \mid \tau_i)$. $\pi_{tot}(\mathbf{u_{tot}} \mid \tau_{\mathbf{tot}}) = 1$, when $\mathbf{u_{tot}} = \arg\max_{\mathbf{u}} \psi_\alpha(Z_{tot}(\tau_{\mathbf{tot}}, \mathbf{u_{tot}}))$, and it is 0 otherwise. For $\pi_i(u_i \mid \tau_i)$, $\pi_i(u_i \mid \tau_i) = 1$, when $u_i = \arg\max_u \psi_\alpha(Z_i(\tau_i, u_i))$, otherwise 0. The RIGM principle becomes the following formula:

$$\pi_{tot}(\mathbf{u_{tot}} \mid \tau_{\mathbf{tot}}) = \prod_{i=1}^N \pi_i(u_i \mid \tau_i). \tag{B.43}$$

The same as the IGO case, we can use a diffusion model to generate risk-sensitive multi-agent action. Let's treat the generated data $x_i^0$ as risk-sensitive action $u_i$ of agent $i$. Then, the probability $p_{\theta_i}(x_i^0) = p_{\theta_i}(u_i)$ becomes $\pi_i(u_i \mid \tau_i)$, where $\pi_i$ is the risk-sensitive policy of agent $i$ and $\tau_i$ is the local observation. Further, let's treat the generated total data $\mathbf{x_{tot}^0}$ as $\mathbf{u_{tot}}$. The probability $p_{\theta_{tot}}(\mathbf{x_{tot}}^0) = p_{\theta_{\mathbf{tot}}}(\mathbf{u_{tot}})$ becomes $\pi_{tot}(\mathbf{u_{tot}} \mid \tau_{\mathbf{tot}})$, where $\pi_{tot}$ is the risk-sensitive policy of the multi-agent system, and $\tau_{tot}$ is the aggregated observation.

$$\prod_{i=1}^N p_{\theta_i}(x_i^0) = p_{\theta_{tot}}(x_{tot}^0), \tag{B.44}$$

$$\prod_{i=1}^N p_{\theta_i}(u_i) = p_{\theta_{\mathbf{tot}}}(\mathbf{u_{tot}}), \tag{B.45}$$

$$\prod_{i=1}^N \pi_i(u_i \mid \tau_i) = \pi_{tot}(\mathbf{u_{tot}} \mid \tau_{\mathbf{tot}}), \tag{B.46}$$

$$\pi_{tot}(\mathbf{u_{tot}} \mid \tau_{\mathbf{tot}}) = \prod_{i=1}^N \pi_i(u_i \mid \tau_i). \tag{B.47}$$

We have shown that the IGD principle is a generalization of the general RIGM principle. Thus, the IGD principle is a generalization of the RIGM principle.

# C DoF Algorithm

In this section, we will provide a detailed explanation of the noise factorization functions proposed in the main text. We will discuss the basic principles of each method, explaining the mathematical models and algorithms that form the core of their functionality.

## C.1 Noise Factorization Functions

### C.1.1 $f$=Concat

In this section, we provide a detailed overview of the $f$=Concat noise factorization function. We start with a noise term $\boldsymbol{\epsilon}_{tot} \sim \mathcal{N}(\boldsymbol{\mu}, \boldsymbol{\theta}) \in \mathbb{R}^{d \times N}$, where $\boldsymbol{\epsilon}_{tot}$ is modeled as a Gaussian noise with mean vector $\boldsymbol{\mu}$ and covariance matrix $\boldsymbol{\theta}$, covering $d \times N$ dimensions. The mixer function $f$ decomposes $\epsilon_{tot}$ into $N$ smaller noise vectors $[\boldsymbol{\epsilon}_i]_{i=1}^N$, each $\epsilon_i \in \mathbb{R}^d$. This decomposition is specifically done as $\epsilon_i = \epsilon_{tot}[(i-1) \times d : i \times d]$, meaning each $\epsilon_i$ contains the elements of $\epsilon_{tot}$ from the $(i-1) \times d$-th dimension to the $i \times d - 1$-th dimension. By this partitioning, we ensure that each $\epsilon_i$ retains the properties of the original Gaussian noise within its respective subspace.

In diffusion probability models, the noise $\epsilon_{tot}$ must be Gaussian with diagonal covariance. This requirement ensures that the noise components are uncorrelated and independently distributed across different dimensions. A notable property of Gaussian noise with diagonal covariance is that the concatenation of multiple such Gaussian noises remains Gaussian. Mathematically, if each $\epsilon_i$ follows a Gaussian distribution with diagonal covariance, then the combined noise vector $\epsilon_{tot}$ formed by the mixer function $f$ also maintains a Gaussian distribution with diagonal covariance.

We further analyze the statistical properties of Gaussian distributions. Assume $\epsilon_i \sim \mathcal{N}(\mu_i, \theta_i)$ for $i = 1, 2, \cdots, N$, where each $\theta_i$ represents a diagonal covariance matrix. When these noise vectors are concatenated, the resulting noise vector $\epsilon_{tot} = \oplus_{i=1}^{N} \epsilon_i$ has a mean vector $\boldsymbol{\mu}_{tot} = [\mu_1, \mu_2, \cdots, \mu_N]$ and a block-diagonal covariance matrix $\boldsymbol{\theta}_{tot}$, where each block on the diagonal corresponds to $\theta_i$. This confirms that $\epsilon_{tot}$ remains Gaussian with diagonal covariance.

## C.1.2   $f$=WCONCAT

In this section, we provide a detailed overview of the $f$=Wconcat noise factorization function, which utilizes learnable weight variables to combine Gaussian noise. The mixing function $f$ for $f$=Wconcat is defined as follows:

$$\epsilon_{tot}(\boldsymbol{x}^k, \boldsymbol{y}, k)[(i-1) \times d : i \times d] = k_i \epsilon_i(\boldsymbol{x}_i^k, \boldsymbol{y}, k) \tag{C.48}$$

Here, the coefficients $k_i$ are trained through learnable weight variables. For the $f$=Wconcat case, since each $\epsilon_i$ follows a Gaussian distribution with diagonal covariance, the scaled noise term $k_i \epsilon_i$ also retains a Gaussian distribution with diagonal covariance. Therefore, when the scaled vectors $k_i \epsilon_i$ are concatenated, the resulting vector $\epsilon_{tot} = \oplus_{i=1}^{N} k_i \epsilon_i$ still follows a Gaussian distribution with diagonal covariance.

## C.1.3   $f$=ATTEN

In this section, we provide a detailed overview of the $f$=Atten noise factorization function, which utilizes an attention mechanism to combine Gaussian noise. Specifically, the mixing function $f$ for $f$=Atten is defined as follows:

$$\epsilon_{tot}(\boldsymbol{x}^k, \boldsymbol{y}, k)[(i-1) \times d : i \times d] = \sum_{j=1}^{N} w_i^j \epsilon_j(\boldsymbol{x}_j^k, \boldsymbol{y}, k) \tag{C.49}$$

Here, $w_i^j$ are weights computed using a multi-head attention mechanism, reflecting the relative importance of each source of noise in the current context.

The multi-head attention mechanism is widely used in natural language processing and computer vision due to its ability to dynamically assign different weights to different inputs, thereby highlighting significant information. In the $f$=Atten , the multi-head attention mechanism captures information across different dimensions through multiple attention heads. Each attention head computes a set of weights, and the results are then aggregated to generate the final combined noise. This design enhances the model's expressiveness and flexibility, making it more effective in handling complex tasks.

Crucially, despite the complexity of the attention mechanism, the $f$=Atten function maintains the statistical properties of Gaussian noise. Since a linear combination of diagonal covariance Gaussian noises remains Gaussian, the total noise $\epsilon_{tot}$ generated using the mixing function $f$ retains its Gaussian distribution. This characteristic is vital for ensuring the stability and consistency of the generation process.

However, the $f$=Atten method has certain limitations when it comes to decentralized execution. During the backward denoising steps, each agent $i$ needs access to the noise information from other agents to generate its state $x_i^k$. This requirement limits the applicability of the $f$=Atten method in decentralized settings, as each agent cannot independently complete the denoising process.

## C.1.4   $f$=QMIX

Similar to $f$=Atten, through using $f$=QMIX, each agent cannot independently complete the denoising process. The $f$=QMIX noise factorization function adapts the QMIX architecture, to combine Gaussian noise. This method employs a mixing network that takes individual agent noises as input and produces a combined total noise. The mixing network's weights are generated by a hypernetwork, allowing for state-dependent mixing of noises.

## C.1.5   LOSS FUNCTION FOR NOISE FACTORIZATION

In this section, we discuss the loss functions for different noise factorization functions: $f$=Concat, $f$=WConcat.

For $f$=Concat, the loss function is defined as follows:

$$L(\theta, \phi) := \mathbb{E}_{k, \tau \in \mathcal{D}, \beta \sim \text{Bern}(p)} \left[ \left\| \epsilon - \epsilon_\theta^{tot} \left( x_k(\tau), (1 - \beta) y(\tau) + \beta \emptyset, k \right) \right\|^2 \right]$$
$$+ \mathbb{E}_{(s, u, s') \in \mathcal{D}} \left[ \left\| u - f_\phi(o, o') \right\|^2 \right] \tag{C.50}$$

where

$$\epsilon_{tot} \left( \mathbf{x}^k, \mathbf{y}, k \right) [(i - 1) \times d : i \times d] = \epsilon_i \left( \mathbf{x}_i, \mathbf{y}, k \right) \tag{C.51}$$

$\epsilon_\theta$ represents the parameterized noise model applied across all agents.

For each trajectory $\tau$, noise $\epsilon$ is sampled from a normal distribution $\mathcal{N}(0, I)$, and a timestep $k$ is selected from a uniform distribution $\mathcal{U}\{1, \ldots, K\}$. A noisy array of states $x_k(\tau)$ is constructed, and the model predicts the noise as $\hat{\epsilon}_\theta := \epsilon_\theta(x_k(\tau), y(\tau), k)$. With probability $p$, the conditioning information $y(\tau)$ may be ignored, and the inverse dynamics model $f_\phi(o, o')$ is trained to predict the action $u$ leading from observation $o$ to $o'$. The term $\epsilon_\theta^{tot}$ is a predicted noise function, and $\beta$, sampled from a Bernoulli distribution, determines whether the condition $y(\tau)$ is used or replaced by an empty set. The dataset $\mathcal{D}$ contains the trajectories utilized for training. This loss function effectively balances the model's capability to predict noise and capture the underlying dynamics of the system.

For $f$=WConcat, the loss function is as follows:

$$L(\theta, \phi) := \mathbb{E}_{k, \tau \in \mathcal{D}, \beta \sim \text{Bern}(p)} \left[ \left\| \epsilon - \epsilon_\theta^{tot} \left( x_k(\tau), (1 - \beta) y(\tau) + \beta \emptyset, k \right) \right\|^2 \right]$$
$$+ \mathbb{E}_{(s, u, s') \in \mathcal{D}} \left[ \left\| u - f_\phi(o, o') \right\|^2 \right] \tag{C.52}$$

where

$$\epsilon_{tot} \left( \mathbf{x}^k, \mathbf{y}, k \right) [(i - 1) \times d : i \times d] = k_i \epsilon_i \left( \mathbf{x}_i, \mathbf{y}, k \right) \tag{C.53}$$

$k_i$ is trained through learnable weight variables, assigning a specific weight to each noise model.

## C.2 Data Factorization Functions $h$

In this work, we use the data factorization function $h$ to model relationship among generated data. All the noise factorization function $f$ (Concat, WConcat, Atten, and QMIX) can be used the data factorization $h$.

For $h$ = Concat, the data factorization function is as follows:

$$x_{tot}[(i - 1) \times d : i \times d] = x_i \tag{C.54}$$

For $h$ = WConcat, the data factorization function is as follows:

$$x_{tot}[(i - 1) \times d : i \times d] = k_i x_i \tag{C.55}$$

$k_i$ is trained through learnable weight variables.

For $h$ = Atten, the data factorization function is as follows:

$$x_{tot} [(i - 1) \times d : j \times d] = \sum_{j=1}^{N} w_i^j x_j \tag{C.56}$$

$k_i$ is trained through learnable weight variables.

For $h$ = Qmix, the data factorization function is as follows:

$$x_{tot} = h(x_1, x_2, \cdots, x_n) \tag{C.57}$$

Where, $h$ is monotonic non-linear function.

## C.3 DoF-Trajectory

In the trajectory-based version of our reinforcement learning model, we focus on modeling states using diffusion processes. Since action sequences in reinforcement learning are typically discrete and noisy, while states are continuous, we apply the diffusion model to states rather than actions. The state sequence within a trajectory segment is represented as:

$$x^k(\tau) := [o_t, o_{t+1}, \ldots, o_{t+H-1}]^k \tag{C.58}$$

where $k$ denotes the time step and $t$ specifies the time of accessing states in trajectory $\tau$. The sequence $x^k(\tau)$ is treated as a noisy state sequence over the $H$-step prediction horizon. In the forward training process, to leverage the diffusion model for planning, the diffusion process is conditioned on the trajectory return $y(\tau)$, employing unsupervised classification and low-temperature sampling, thereby extracting high-probability optimal trajectories from a dataset containing suboptimal paths.

For multiple agents, we introduce a Noise factorization function $f(\cdot)$ to integrate the noise generated by each agent's diffusion model:

$$\epsilon_{\text{tot}}^{\theta} = f(\epsilon_{\theta_1}, \epsilon_{\theta_2}, \ldots, \epsilon_{\theta_n}) \tag{C.59}$$

Noise factorization functions can be stacking, attention mechanisms, or adaptive dynamic programming, among others. To derive a policy from the sampled states, we use an inverse dynamics model to estimate actions:

$$u_t := D_\phi(o_t, o_{t+1}) \tag{C.60}$$

The designed loss function aims to minimize noise prediction error and action prediction error:

$$
\begin{aligned}
L(\theta, \phi) := & \mathbb{E}_{k, \tau \in D, \beta \sim \text{Bern}(p)} \left[ \left\| \boldsymbol{\epsilon} - \boldsymbol{\epsilon}_\theta^{tot}(\boldsymbol{x}_k(\tau), (1 - \beta)y(\tau) + \beta \emptyset, k) \right\|^2 \right] \\
& + \mathbb{E}_{(\boldsymbol{o}, \boldsymbol{u}, \boldsymbol{o}') \in D} \left[ \left\| \boldsymbol{u} - D_\phi(\boldsymbol{o}, \boldsymbol{o}') \right\|^2 \right]
\end{aligned}
\tag{C.61}
$$

The loss function $L(\theta, \phi)$ optimizes the model's decision-making capability in complex environments.

Next, we will provide a detailed explanation of the implementation of DoF in Trajectory. The DoF algorithm is described in Algorithm 3.

## C.4 DoF-Policy

DoP-Policy is a combination of the DiffusionQL algorithm and the noise and data factorization functions. The DoF-Policy agent is built using the DiffusionQL (Wang et al., 2023) algorithm. It is designed for continuous action domains only. DiffusionQL initially ensures that the Actor's behavior policy closely aligns with the offline dataset and subsequently enhances policy performance through policy gradient optimization based on the Critic's estimations. DOM2 Li et al. (2023), extend the diffusion-QL to Offline MARL by learning a separated critic for each diffusion process. It does not address inter-agent cooperation and credit assignment in multi-agent reinforcement learning. Moreover, they overlook the environmental instability caused by interactions among agents in multi-agent scenarios.

In DiffusionQL, the diffusion policy represents each agent's action $u_i$, based on its observation, as a Gaussian distribution $x_i$. This is achieved by utilizing a reverse process of conditional diffusion model to represent $\pi_{\theta_i}$ as

$$\pi_{\theta_i}(\boldsymbol{u_i} \mid \boldsymbol{o_i}) = p_{\theta_i}(\boldsymbol{u_i}^{0:K} \mid \boldsymbol{o_i}) = \mathcal{N}(\boldsymbol{u_i}^K; \boldsymbol{0}, \boldsymbol{I}) \prod_{k=1}^{K} p_{\theta_i}(\boldsymbol{u_i}^{k-1} \mid \boldsymbol{u_i}^k, \boldsymbol{o_i}), \tag{C.62}$$

where $p_{\theta_i}(\boldsymbol{u_i}^{k-1} \mid \boldsymbol{u_i}^k, \boldsymbol{o_i})$ can be reparameterized as $\mathcal{N}(\boldsymbol{u_i}^{k-1}; \boldsymbol{\mu}_{\theta_i}(\boldsymbol{u_i}^k, \boldsymbol{o_i}, k), \boldsymbol{\Sigma}_{\theta_i}(\boldsymbol{u_i}^k, \boldsymbol{o_i}, k))$ whose mean constructed as

$$\boldsymbol{\mu}_{\boldsymbol{\theta}_i}(\boldsymbol{u_i}^k, \boldsymbol{o_i}, k) = \frac{1}{\sqrt{\alpha_i}} \left( \boldsymbol{u_i}^k - \frac{\beta_k}{\sqrt{1 - \bar{\alpha}_k}} \boldsymbol{\epsilon}_{\boldsymbol{\theta}_i}(\boldsymbol{u_i}^k, \boldsymbol{o_i}, k) \right). \tag{C.63}$$

To gain ground action $\boldsymbol{u_i}^0$ of each agent, we need to start sampling from $\boldsymbol{u_i}^K \sim \mathcal{N}(\boldsymbol{0}, \boldsymbol{I})$ to $\boldsymbol{u_i}^0$ via

$$\boldsymbol{u_i}^{k-1} \mid \boldsymbol{u_i}^k = \frac{\boldsymbol{u_i}^k}{\sqrt{\alpha_k}} - \frac{\beta_k}{\sqrt{\alpha_k(1 - \bar{\alpha}_k)}} \boldsymbol{\epsilon}_{\boldsymbol{\theta}_i}(\boldsymbol{u_i}^k, \boldsymbol{o_i}, k) + \sqrt{\beta_k} \boldsymbol{\epsilon}, \boldsymbol{\epsilon} \sim \mathcal{N}(\boldsymbol{0}, \boldsymbol{I}), \text{for} : k = K, \ldots, 1. \tag{C.64}$$

---

**Algorithm 3** DoF

---

*# Training Process*

**Initialize**: Offline dataset $\mathcal{D}$, Agent Nums $N$, Inverse Dynamic $f_\phi$, Batch Size $M$.

 1: **for** $n = 1$ **to** $n\_epoch$ **do**
 2:     Sample trajectory sequence $[\tau_i]_{i=1}^N$ of $H$ and condition $[y^i(\tau^i)_{i=1}^N]$ from $\mathcal{D}$ with batch size $\mathcal{M}$
 3:     Sample noise $\epsilon \sim \mathcal{N}(\mathbf{0}, \boldsymbol{I})$
 4:     **for** each Agent $i \in N$ do **do**
 5:         Sample a timestep $t \sim \mathcal{U}\{1, \cdots, T\}$
 6:         Construct a noisy array of states $x_t(\tau_i)$
 7:         Predict the noise $\hat{\epsilon}_\theta^i := \epsilon_\theta^i(x_t(\tau_i, y(\tau_i), t)$
 8:         Omit the condition $y(\tau_i)$ with probability $\beta_i \sim \mathrm{Bern}(p_i)$
 9:     **end for**
10:     $\epsilon_\theta^{tot} \leftarrow f(\epsilon_{\theta_1}, \epsilon_{\theta_2}, \ldots, \epsilon_{\theta_N})$
11:     $o_{tot} \leftarrow h(o_1, o_2, \ldots, o_N)$ ($o_i$ is the local observation of agent $i$, it is the first element of $\tau_i^0$)
12:     Get the $L(\theta, \phi) := \mathbb{E}_{k, \tau \in D, \beta \sim \mathrm{Bern}(p)} \left[ \|\boldsymbol{\epsilon} - \boldsymbol{\epsilon}_\theta^{\mathrm{tot}} \left( \boldsymbol{x}_k(\tau), (1 - \beta)y(\tau) + \beta\emptyset, k \right)\|^2 \right] + \mathbb{E}_{s,u,s'} \in$
        $D \left[ ||u - f_\phi(o, o')||^2 \right]$
13:     Update $[\epsilon^i]_{i=1}^N$ model
14: **end for**

*# Trajectory sampling Process*

 1: **Input**: Noise model $\epsilon_\theta$, Inverse Dynamic $I_\phi$, guidance scale $w$, History length $C$, condition $y$
 2: Initialize $h \leftarrow \mathrm{Queue}(\mathrm{length} = C)$; $t \leftarrow 0$
 3: **while** not done do **do**
 4:     Observe joint observation $o$; $h.insert(o)$; Initialize $\tau_K \sim \mathcal{N}(0, \alpha I)$
 5:     **for** $t = T$ to 1 do **do**
 6:         $\tau_t[: \mathrm{length}\,(h)] \leftarrow h$
 7:         **for** agent $i \in \{1, 2, \ldots, N\}$ **do**
 8:             $\hat{\epsilon}^i \leftarrow \epsilon_\theta^i\left(\tau_t^i, t\right) + \omega \left(\epsilon_\theta^i\left(\tau_t^i, y^i, t\right) - \epsilon_\theta^i\left(\tau_t^i, t\right)\right)$
 9:         **end for**
10:         $\hat{\epsilon}_\theta^{tot} \leftarrow f(\hat{\epsilon}_{\theta_1}, \hat{\epsilon}_{\theta_2}, \ldots, \hat{\epsilon}_{\theta_N})$
11:         $\left(\mu_{t-1}^{tot}, \Sigma_{t-1}^{tot}\right) \leftarrow \mathrm{Denoise}\left(\tau_t, \hat{\epsilon}_\theta^{tot}\right)$
12:         $\tau_{t-1}^{tot} \sim \mathcal{N}(\mu_{t-1}^{tot}, \alpha\Sigma_{t-1}^{tot})$
13:     **end for**
14:     $o_{tot} \leftarrow h(o_1, o_2, \ldots, o_N)$ ($o_i$ is the local observation of agent $i$, it is the first element of $\tau_i^0$)
15:     **for** agent $i \in \{1, 2, \cdots, N\}$ **do**
16:         $u_t^i \leftarrow f_{\phi^i}\left(o_t^i, o_{t+1}^i\right)$
17:     **end for**
18:      Execute $u_t$ in the environment; $t \leftarrow t + 1$
19: **end while**

---

Through the noise factorization function $f$, DoF-Policy factorizes the global noise $\epsilon_\theta^{tot}$ into noises $\epsilon_{\theta_i}$ generated by each agent's diffusion model via

$$\boldsymbol{\epsilon}_{\theta_i} := \boldsymbol{\epsilon}_{\theta_i}(\sqrt{\overline{\alpha_k}}\boldsymbol{u}_i + \sqrt{1 - \overline{\alpha_k}}\boldsymbol{\epsilon}, \boldsymbol{o}_i, k), i \in (1, n) \qquad (C.65)$$

$$\boldsymbol{\epsilon}_\theta^{tot} = f(\boldsymbol{\epsilon}_{\theta_1}, \boldsymbol{\epsilon}_{\theta_2}, \ldots, \boldsymbol{\epsilon}_{\theta_n}) \qquad (C.66)$$

where the $f$ is the noise factorization function, enabling the agent to account for the non-stationarity of the environment caused by other agents' behaviors during training. Consequently, in both policy learning and action value function training, we attained global training goal policy $\pi^*$ with parameters $\theta$ by considering global information to minimize the loss function $\mathcal{L}(\theta)$:

$$\mathcal{L}(\theta) = \mathcal{L}_{\mathrm{diff}}(\theta) + \mathcal{L}_{\mathrm{pg}}(\theta) \qquad (C.67)$$

$$= \mathbb{E}_{\boldsymbol{\epsilon} \sim \mathcal{N}(\boldsymbol{0}, \boldsymbol{I}), (\boldsymbol{s}, \boldsymbol{u}) \sim \mathcal{D}} \left[ ||\boldsymbol{\epsilon} - \boldsymbol{\epsilon}_\theta^{tot}||^2 \right] - \alpha \cdot \mathbb{E}_{\boldsymbol{s} \sim \mathcal{D}, \boldsymbol{u}^0 \sim \pi_\theta} \left[ Q_\Phi(\boldsymbol{s}, \boldsymbol{u}^0) \right] \qquad (C.68)$$

where the second term $\mathcal{L}_{\mathrm{pg}}(\theta)$ can utilize various methods while in this case we follow the DiffusionQL (Wang et al., 2023) to learn $Q_\Phi(\boldsymbol{s}, \boldsymbol{u}^0)$ for policy improvement, and the $\alpha = \frac{\eta}{\mathbb{E}_{(\boldsymbol{s}, \boldsymbol{u}) \sim \mathcal{D}}[|Q_\Phi(\boldsymbol{s}, \boldsymbol{u}^0)|]}$ is a hyperparameter to balance the two loss terms.

During execution, each agent samples its own action through the diffusion model based on the IGD principle and Equation(C.63). This allows for decentralized execution of the agents.

The DoF-P algorithm is described in Algorithm 4.

---

**Algorithm 4** DoF-POLICY

---

*# Training Process*

**Initialize**: Offline dataset $\mathcal{D}$, Agent Nums $N$, Policy network $\pi_\theta$, Critic network $Q_\Phi$ (double Q-learning Hasselt (2010) could be added), and target network $\pi_{\theta'}$, $Q_{\Phi'}$.

1: **for** $n = 1$ **to** $n\_epoch$ **do**
2:     Sample mini-batch $\mathcal{B} = \{(\mathcal{O}_t^N, \mathcal{U}_t^N, r_t, \mathcal{O}_{t+1}^N)\} \sim \mathcal{D}$
    *# Critic learning*
3:     **for** each Agent $i \in N$ **do**
4:         Sample action $\boldsymbol{u}_{t+1,i}^0 \sim \pi_{\theta_i'}(\boldsymbol{u}_{t+1,i} \mid \boldsymbol{o}_{t+1,i})$
5:     **end for**
6:     $\boldsymbol{u}_{tot} \leftarrow h(\boldsymbol{u}_1, \boldsymbol{u}_2, \ldots, \boldsymbol{u}_N)$
7:     Update $Q_\Phi$ by Q-learning method
    *# Policy learning*
8:     **for** each Agent $i \in N$ do **do**
9:         Sample a timestep $k \sim \mathcal{U}\{1, \cdots, K\}$
10:       Sample a random noisy distribution of action $x_i^k$
11:       Predict the noise $\boldsymbol{\epsilon}_{\theta_i} := \boldsymbol{\epsilon}_{\theta_i}(\sqrt{\overline{\alpha}_k}\boldsymbol{u}_i + \sqrt{1 - \overline{\alpha}_k}\boldsymbol{\epsilon}, \boldsymbol{o_i}, k)$
12:     **end for**
13:     $\epsilon_\theta^{tot} \leftarrow f(\epsilon_{\theta_1}, \epsilon_{\theta_2}, \ldots, \epsilon_{\theta_N})$
14:     Update policy network $\pi_\theta$ by Equation C.67
15:     Update the two target networks $\pi_{\theta'}$ and $Q_{\Phi'}$ through soft update.
16: **end for**

---

# D EXPERIMENT DETAILS

## D.1 EXPERIMENTAL SETUP

We select four categories of MARL algorithms for comparison: (I) uses online algorithms to train offline datasets, such as QMIX (Rashid et al., 2018). (II) offline multi-agent reinforcement learning algorithms based on the Centralized Training with Decentralized Execution (CTDE) paradigm, including MABCQ (Jiang & Lu, 2021), MACQL (Kumar et al., 2020), MAICQ (Yang et al., 2021), and OMAR (Pan et al., 2022), and MA-TD3-BC (Fujimoto & Gu, 2021). These methods optimize multi-agent collaborative strategies by combining the advantages of centralized training and decentralized execution. (III) Offline multi-agent algorithms based on the Decision Transformer, such as MADT (Meng et al., 2023). (IV) Offline multi-agent reinforcement learning algorithms based on diffusion models: MADIFF (Zhu et al., 2024), DOM2(Li et al., 2023), and Independent Diffusion. These methods use diffusion models to generate more effective multi-agent strategies. To demonstrate robustness, was tested with experiments using five different seeds.

The work used for comparison is listed as shown in table 6.

## D.1.1 COMPUTING RESOURCES

The experiments are conducted on a high-performance computing cluster equipped with multiple NVIDIA GeForce RTX 3090 GPUs, which provide the necessary computational power. The CPUs

---

[1]https://github.com/oxwhirl/pymarl

[2]https://github.com/instadeepai/og-marl

[3]https://github.com/instadeepai/og-marl

[4]https://github.com/YiqinYang/ICQ

[5]https://github.com/ling-pan/OMAR

[6]https://github.com/sfujim/TD3_BC

[7]https://github.com/ReinholdM/Offline-Pre-trained-Multi-Agent-Decision-Transformer

[8]https://github.com/zbzhu99/madiff

Table 6: Baseline algorithms

| Algorithms | Brief Description |
|---|---|
| QMIX[1] (Rashid et al., 2018) | Facilitates a monotonic combination of individual agent utilities. |
| MABCQ[2] (Jiang & Lu, 2021) | Offline decentralized multi-agent reinforcement learning using value deviation and transition normalization for coordinated policies. |
| MACQL[3] (Kumar et al., 2020) | Prevents overestimation by adjusting Q-values for policy samples and dataset state-action pairs. |
| MAICQ[4] (Yang et al., 2021) | Mitigates extrapolation error by trusting only dataset-provided state-action pairs |
| OMAR[5] (Pan et al., 2022) | Combining first-order and zero-order methods improves conservative value function optimization. |
| MA-TD3+BC[6] (Fujimoto & Gu, 2021) | A behavior cloning term is added to TD3 (Fujimoto et al., 2018) to regularize the policy. |
| MADT[7] (Meng et al., 2023) | uses transformer-based offline RL to integrate global information into agents' policies via centralized critic gradients. |
| MADIFF[8] (Zhu et al., 2024) | incorporates attention mechanisms into Unet diffusion models to model trajectories. The version we adopt is MADIFF-D. |

used in the cluster are Intel(R) Xeon(R) Silver 4216 processors, each running at 2.10GHz. To ensure the robustness of our results, we run the DoF algorithm five times with different random seeds in each experimental setup. In the MPE environment, a single run of the DoF algorithm takes approximately 5 hours to complete, with convergence typically achieved within the first 1 to 2 hours.

### D.1.2 HYPERPARAMETERS OF DoF

We implement DoF based on the source code of MADIFF, Decision Diffuser, and DiffusionQL.

The hyperparameters of the DoF for trajectory-generation are shown in Table 7. The hyperparameters of DoF algorithm for policy-generation are shown in Table 8.

**DoF-Trajectory**   As shown in Table 7, the learning rate is set to 0.0002, which dictates the step size for each parameter update. The condition guidance weight is selected from $\{1.2, 1.4, 1.6, 1.8\}$, depending on the task requirements. The number of diffusion steps is chosen from $\{100, 200, 300\}$, based on the specific task, and adjusts the model's reliance on conditional information during generation. The planning horizon is set to 20, representing the number of future time steps considered by the model for planning and prediction. The history horizon is set to 8, indicating the number of past time steps used for decision making. The condition dropout is set to 0.25, randomly dropping parts of the conditional information during training to prevent overfitting and enhance the model's generalization ability. The agent share noise is set to False, meaning different agents do not share noise during training, which helps improve the diversity and robustness of the multi-agent system. The discount factor is set to 0.99 for calculating the discounted future rewards, allowing the model to consider both immediate and future rewards in long-term planning. The loss type is set to L2, using mean squared error as the loss function, which penalizes larger deviations more heavily and helps reduce significant prediction errors. The batch size is set to 32, indicating the number of samples used in each training iteration.

**DoF-Policy**   As shown in Table 8, we use values of $\{1e^{-3}, 1e^{-4}, 1e^{-5}\}$ for the policy learning rate of the Adam optimizer, while a fixed learning rate of $3e^{-4}$ is used for the Q-networks' Adam optimizer. The parameter $\tau$ is set to 0.005, which is the update rate for the target networks. For $\eta$ in the loss function $\mathcal{L}(\theta)$, we use a fixed value of 0.5 to balance the two loss terms. The diffusion step is set to $K = 10$ for action inference. We normalize the action space to $[-1, 1]$ using the hyperparameter max action. The model is evaluated 200 times in total, with evaluations occurring every 1,000 training steps.

Next, we will provide a detailed introduction of the experimental environments for MPE, SMAC, and MA mujoco in Sections D.1.4, D.1.3, and D.1.5, and explain the sources of the offline datasets.

Table 7: Hyperparameter Settings for DoF in Trajectory

| Hyper-parameter | Value |
|---|---|
| learning rate | 0.0002 |
| horizon | 20 |
| history horizon | 8 |
| condition dropout | 0.25 |
| condition guidance weight | 1.2 |
| number of diffusion steps | 200 |
| discount | 0.99 |
| loss type | L2 |
| batch size | 32 |
| agent share noise | False |
| optimizer | Adam optimizer |

Table 8: Hyperparameter Settings for DoF in Policy

| Hyper-parameter | Value |
|---|---|
| diffusion Step | 10 |
| discount | 0.99 |
| max action | 1.0 |
| beta schedule | linear |
| $\tau$ | 0.005 |
| $\eta$ | 0.5 |
| learning rete | 0.0003 |
| eval iterations | 200 |
| train iterations step | 1000 |
| optimizer | Adam optimizer |

### D.1.3 STARCRAFT MULTI-AGENT CHALLENGE (SMAC) AND SMACV2

The SMAC environment (Samvelyan et al., 2019) is a widely used benchmark for evaluating co-operative multi-agent reinforcement learning (MARL) algorithms. In SMAC, two teams engage in real-time battles, with one controlled by built-in AI and the other by learned multi-agent policies. Agents must independently make decisions while cooperating to achieve victory. The environment includes diverse scenarios with varying difficulty based on unit composition, terrain constraints, and team balance. It features both homogeneous settings, where all agents share the same unit type, and heterogeneous settings, where different unit types require coordinated strategies. We evaluate our approach on six SMAC maps: 3m, 8m, 5m_vs_6m, 2c_vs_64zg, 2s3z, and 3s5z_vs_3s6z, covering both symmetric and asymmetric matchups as well as homogeneous and heterogeneous unit compositions. This variety allows for a comprehensive assessment of an algorithm's capability to coordinate agents in diverse combat settings.

SMACv2 (Ellis et al., 2023) is an updated version of the SMAC benchmark, designed specifically for research in cooperative multi-agent reinforcement learning (MARL). The update introduces three major changes: randomizing start positions, randomizing unit types, and adjusting unit sight and attack ranges. The first two changes address a key limitation of the original SMAC benchmark, which is its lack of sufficient randomness in many maps, making it less challenging for modern MARL algorithms. The adjustment to unit sight and attack ranges increases agent diversity and aligns these attributes more closely with their actual values in StarCraft, enhancing the benchmark's realism and complexity. We conducted experiments on three SMACv2 scenarios: terran_5_vs_5, zerg_10_vs_10, and zerg_5_vs_5.

**Dataset:** We utilized the datasets from the off-the-grid offline dataset (Formanek et al., 2023), where each map is divided into three datasets: good, medium, and poor, based on the quality of the joint policies. This dataset enhances diversity by leveraging several different joint policies and adding a small amount of exploration noise.

### D.1.4 MULTI-AGENT PARTICLE ENVIRONMENT (MPE)

MPE (Lowe et al., 2017a) is a straightforward multi-agent particle environment where particles can perform continuous observations and discrete actions. The experiments described in this study utilized three distinct environments. The Spread environment comprises three agents and three landmarks; the agents must learn to avoid collisions while covering all landmarks. The Tag environment includes one pre-trained prey, three predators, and two obstacles. The predators must cooperate to apprehend the faster prey. The World environment also includes one pre-trained prey and three predators; the prey agent needs to locate food on the map and can hide in a forest to avoid detection.

**Dataset:** The dataset used in this study, collected by OMAR (Pan et al., 2022), consists of multiple datasets of varying quality, developed by introducing noise into the behavioral policy in MATD3 to enhance diversity. Random-quality datasets were generated using a randomly initialized policy for one million steps. The medium replay dataset was obtained by recording all samples in the buffer when training reached a medium performance level. The medium and expert datasets were derived from either a partially pretrained policy with a medium performance level or a fully trained policy.

We normalized the average scores of MPE tasks to better compare the performance of different algorithms, as shown in Table 1. We used expert scores and random scores as the benchmarks for normalization. Let the original episodic return be $S$. The normalized score $S_{\text{norm}}$ is calculated using the formula $S_{\text{norm}} = 100 \times \frac{(S - S_{\text{random}})}{(S_{\text{expert}} - S_{\text{random}})}$, where $S_{\text{random}}$ is the score obtained by a random policy and $S_{\text{expert}}$ is the score obtained by an expert policy. This normalization formula follows the work of Pan et al. [2022] and Fu et al. [2020], ensuring the method's reliability and validity. For specific MPE tasks, we used the following expert and random scores: for the Spread task, the expert and random scores are $516.8$ and $159.8$, respectively; for the Tag task, the expert and random scores are $185.6$ and $-4.1$, respectively; and for the World task, the expert and random scores are $79.5$ and $-6.8$, respectively.

### D.1.5 MULTI-AGENT MUJOCO (MA MUJOCO)

MA Mujoco (Peng et al., 2021) is based on the Mujoco physics engine and provides a high-precision multi-agent simulation platform. The robots are composed of multiple intelligent agents that must learn to cooperate to move faster while maintaining balance. We conducted experiments using the 2-agent halfcheetah (2halfcheetah) configuration, where two different agents control the front half and the back half of the cheetah, respectively.

**Dataset:** We utilized the datasets from the off-the-grid offline dataset (Formanek et al., 2023), where each map is categorized into three datasets: good, medium, and poor, based on the quality of the joint policies.

### D.2 ILLUSTRATIVE EXAMPLES

We demonstrate the superiority of DoF's generation capability under the IGD principle through three tasks: (a) a matrix game generating two-dimensional data, (b) the Landmark covering game, and (c) the Q-value generation game. We will now provide a detailed introduction to the setup of each environment.

### D.2.1 A MATRIX GAME GENERATING TWO DIMENSIONAL DATA

The Matrix-like Game is a simple experimental environment designed to study the generation capabilities of three algorithms: DoF, MADIFF, and Independent Diffusion. In this game, we developed a multi-agent system where each agent is responsible for generating a different dimension of the data, learning to reproduce the ground-truth data. The ground-truth data consists of four sets of two-dimensional Gaussian-distributed data, with their means located at the top-left (-0.75, 0.75), top-right (0.75, 0.75), bottom-left (-0.75, -0.75), and bottom-right (-0.75, 0.75) of the data plane, all with a variance of 0.05. These four sets of data are generated with different probabilities: 0.5 for the top-left, 0.2 for the top-right, 0.2 for the bottom-left, and 0.1 for the bottom-right.

In generative multi-agent reinforcement learning (MARL), generating data that matches the ground truth is a key metric for evaluating the performance of algorithms. We used the DoF, MADIFF, and Independent Diffusion algorithms to generate data, and assessed their performance by analyzing the

Table 9: Payoff matrix of one-step matrix games and reconstructed value functions to approximate the optimal policy.

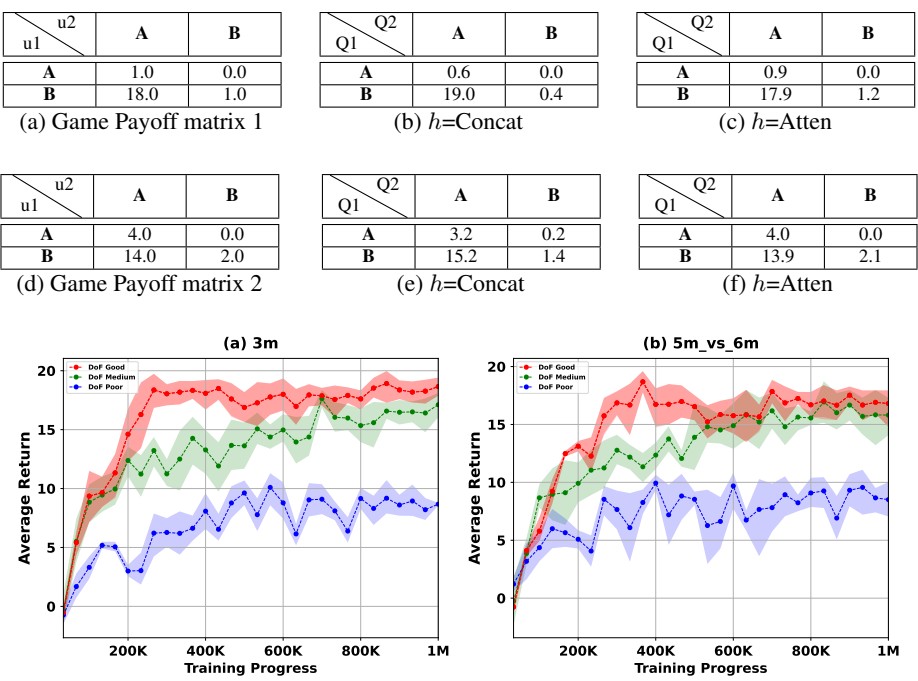

| u2 / u1 | A | B |
|---|---|---|
| A | 1.0 | 0.0 |
| B | 18.0 | 1.0 |

(a) Game Payoff matrix 1

| Q2 / Q1 | A | B |
|---|---|---|
| A | 0.6 | 0.0 |
| B | 19.0 | 0.4 |

(b) $h$=Concat

| Q2 / Q1 | A | B |
|---|---|---|
| A | 0.9 | 0.0 |
| B | 17.9 | 1.2 |

(c) $h$=Atten

| u2 / u1 | A | B |
|---|---|---|
| A | 4.0 | 0.0 |
| B | 14.0 | 2.0 |

(d) Game Payoff matrix 2

| Q2 / Q1 | A | B |
|---|---|---|
| A | 3.2 | 0.2 |
| B | 15.2 | 1.4 |

(e) $h$=Concat

| Q2 / Q1 | A | B |
|---|---|---|
| A | 4.0 | 0.0 |
| B | 13.9 | 2.1 |

(f) $h$=Atten

Figure 5: SMAC Return Curves: (a) 3m environment, (b) 5m_vs_6m environment.

data distribution generated by each method and how well it matched the ground truth. The result are depicted in Figure 3.

### D.2.2 LANDMARK COVERING GAME

In this game, three agents must cooperative cover three landmarks without collision in short-time. This game is developed based the mpe-spread environment. In order to ensure the uniformity of the assessment and reduce random interference, we fixed the initial position of the agent and landmarks in the game. For each algorithm, we ran 10 iterations of planning, each iteration with 10 trajectory samples per agent, and visualized these trajectory points using different colors. The goal was for agents to learn to select the nearest landmark while avoiding overlap, exhibiting cooperative behavior.

### D.2.3 GENERATING Q VALUE

The goal of the game is to reconstruct the one-step payoff matrix $Q_{tot}$ through two agents. Agent $i$ use the diffusion process to generate individual utility value $Q_i$ and they are mixed into $Q_{tot} = h(Q_1, Q_2)$, where $h$ is the data factorization function. We consider two data factorization functions: Concat and Atten. The results are depicted in Table 9. DoF can reconstruct the payoff matrix $Q_{tot}$ well. The Atten function performs better than the Concat function.

### D.3 COMPARISON RESULTS

Table 10: The Average Return of the SMACv2 Scenarios

| Map | Data | BC | MABCQ | MACQL | MAICQ | MADIFF | DoF |
|---|---|---|---|---|---|---|---|
| **terran_5_vs_5** | replay | 7.3±1.0 | 13.8±4.4 | 11.8±0.9 | 13.7±1.7 | 13.3±1.8 | **15.4±1.3** |
| **Zerg_5_vs_5** | replay | 6.8±0.6 | 10.3±1.2 | 10.3±3.4 | 10.6±0.7 | 10.2±1.1 | **12.0±1.1** |
| **terran_10_vs_10** | replay | 7.4±0.5 | 12.7±2.0 | 11.8±2.0 | 14.4±0.7 | 13.8±1.3 | **14.6±1.1** |

Table 11: The Average Return of the Multi-agent Particle Environments (MPE)

| Dataset | Task | MAICQ | MA-TD3+BC | MACQL | OMAR | MADIFF | DoF | DoF-P |
|---|---|---|---|---|---|---|---|---|
| **Expert** | Spread | 101.4±3.4 | 110.3±3.3 | 85.3±4.6 | 113.9±2.6 | 120.1±6.3 | **136.4±3.9** | **126.3±3.1** |
| | Tag | 95.2±10.1 | 113.1±11.6 | 84.3±10.2 | 115.8±13.6 | **120.8±11.3** | 125.6±8.6 | 120.1±6.3 |
| | World | 98.5±21.8 | 95.3±18.3 | 65.4±20.2 | 113.4±23.1 | 124.7±20.1 | **135.2±19.1** | **138.4±20.1** |
| **Medium** | Spread | 29.3±5.5 | 32.3±3.8 | 35.3±10.3 | 45.0±18.8 | **67.5±8.5** | **75.6±8.7** | 60.5±8.5 |
| | Tag | 58.3±18.0 | 63.3±25.6 | 62.3±27.8 | 55.3±16.7 | 78.6±12.3 | **86.3±10.6** | 83.9±9.6 |
| | World | 69.9±20.1 | 72.4±9.3 | 56.4±6.4 | 69.2±21.5 | 80.1±13.4 | 85.2±11.2 | **86.4±10.6** |
| **Md-Replay** | Spread | 13.7±5.6 | 14.4±5.8 | 19.2±6.4 | 35.3±14.0 | **48.4±3.4** | **57.4±6.8** | 48.1±3.6 |
| | Tag | 29.5±21.8 | 25.7±20.1 | 23.9±16.2 | 52.4±18.3 | **57.4±13.4** | **65.4±12.5** | 51.7±10.1 |
| | World | 12.0±9.1 | 15.4±8.1 | 21.3±10.3 | 42.6±28.2 | 51.6±12.1 | **58.6±10.4** | **58.1±11.5** |
| **Random** | Spread | 5.3±3.4 | 8.8±4.4 | 20.5±5.8 | 30.4±8.2 | 20.6±7.6 | **35.9±6.8** | 34.5±5.4 |
| | Tag | 2.2±2.6 | 3.7±3.5 | 2.7±4.4 | 10.9±3.8 | 13.3±3.4 | **16.5±6.3** | 14.8±3.2 |
| | World | 1.0±2.2 | 2.8±3.5 | 2.4±3.2 | 9.2±3.6 | 6.1±2.2 | **13.1±2.1** | **15.1±3.0** |

Table 12: The win rate for the SMAC Scenarios

| Maps | Data | QMIX | MABCQ | MACQL | MAICQ | MADT | MADIFF | DoF |
|---|---|---|---|---|---|---|---|---|
| **3m** | Good | 0.0 | 0.0 | 0.92 | 0.88 | 0.91 | **0.94** | **0.96** |
| | Medium | 0.0 | 0.0 | 0.25 | 0.28 | 0.60 | 0.63 | **0.82** |
| | Poor | 0.0 | 0.0 | 0.0 | 0.0 | 0.0 | 0.0 | **0.06** |
| **8m** | Good | 0.0 | 0.0 | 0.0 | **0.93** | 0.86 | 0.90 | **0.94** |
| | Medium | 0.0 | 0.0 | 0.0 | 0.75 | **0.81** | 0.65 | **0.83** |
| | Poor | 0.0 | 0.0 | 0.0 | 0.05 | 0.0 | 0.0 | **0.08** |
| **5m_vs_6m** | Good | 0.0 | 0.0 | 0.0 | 0.04 | 0.68 | 0.66 | **0.72** |
| | Medium | 0.0 | 0.0 | 0.0 | 0.0 | **0.62** | 0.58 | **0.62** |
| | Poor | 0.0 | 0.0 | 0.0 | 0.0 | 0.0 | 0.0 | **0.03** |
| **2s3z** | Good | 0.0 | 0.0 | 0.70 | **0.78** | 0.76 | 0.60 | **0.78** |
| | Medium | 0.0 | 0.0 | 0.56 | 0.68 | 0.50 | 0.56 | **0.70** |
| | Poor | 0.0 | 0.0 | 0.0 | 0.0 | 0.0 | 0.0 | **0.0** |
| **3s5z_vs_3s6z** | Good | 0.0 | 0.0 | 0.0 | **0.21** | **0.18** | 0.0 | **0.18** |
| | Medium | 0.0 | 0.0 | 0.0 | 0.08 | **0.09** | 0.0 | **0.10** |
| | Poor | 0.0 | 0.0 | 0.0 | 0.0 | 0.0 | 0.0 | 0.0 |
| **2c_vs_64zg** | Good | 0.0 | 0.0 | 0.18 | 0.38 | 0.24 | 0.42 | **0.61** |
| | Medium | 0.0 | 0.0 | 0.14 | 0.17 | 0.06 | 0.18 | **0.23** |
| | Poor | 0.0 | 0.0 | 0.0 | 0.0 | 0.0 | 0.0 | **0.05** |

### D.3.1 SMAC

In offline multi-agent reinforcement learning, the SMAC return metric is commonly used to evaluate performance. To provide a more comprehensive assessment of our algorithm, we also evaluated the win rate. While win rate is a meaningful indicator of performance, especially in environments with well-structured data, offline multi-agent reinforcement learning presents additional challenges, often resulting in lower win rates compared to online methods. In cases where the dataset quality is poor and returns are consistently low (e.g., below 11), the win rate may drop to 0, reducing the comparability between algorithms in these settings.

The SMAC win rate results we present are shown in the table 12. In the Good and Medium datasets, DoF achieved the best win rate in most environments. In simpler environments, such as the 3m and 8m environments in the Good dataset, DoF's win rate was about 96%, followed by the MADIFF and MAICQ algorithms. In the 3m environment, MADIFF's win rate was 94%, while in the 8m environment, MAICQ's win rate was 93%. In the 5m_vs_6m and 2c_vs_64zg environments, DoF achieved the best win rates. In heterogeneous environments, such as the 2s3z and 3s5z_vs_3s6z environments, both the DoF and MAICQ algorithms achieved the best win rates.

To further analyze, we plotted the return curves for all datasets on the 3m and 5m_vs_6m maps. During training, we saved the models periodically throughout the diffusion process, training for a total of 1 million steps and saving a model every 30,000 steps. In the sampling phase, we evaluated each saved Diffusion model and recorded the corresponding return data, as shown in Figure 5.

As shown in Figure 5, in the Good dataset, DoF converges around 200,000 to 300,000 steps, with some fluctuations around the mean afterward. In the Medium dataset, the DoF algorithm only begins

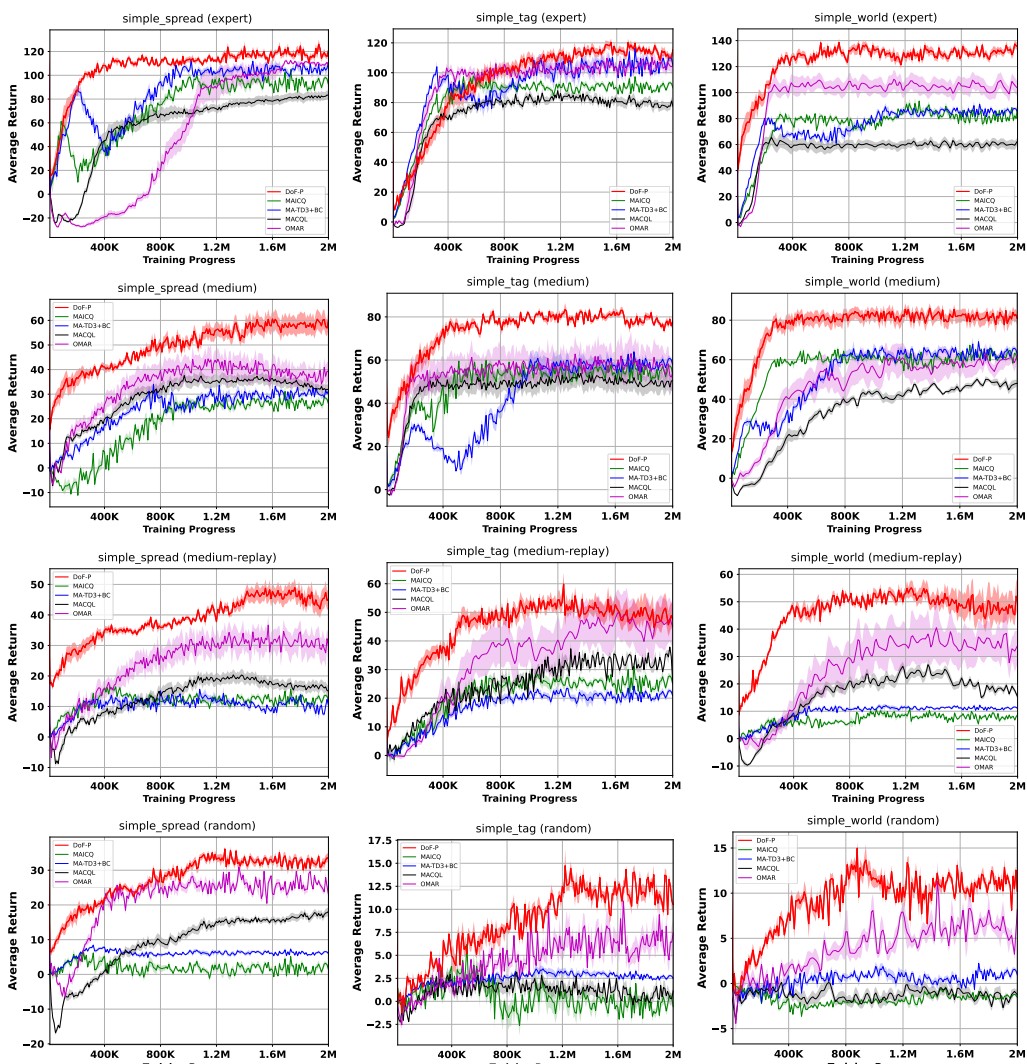

Figure 6: Training Curves of DoF-P, MAICQ, MA-TD3+BC, and OMAR in MPE

to slowly converge around 600,000 steps. In comparison, for the Poor dataset, DoF exhibits larger fluctuations, but the return curve still shows an upward trend.

### D.3.2   SMACv2

We further evaluated our algorithm on the SMACv2 replay dataset in three scenarios: terran_5_vs_5, zerg_10_vs_10, and zerg_5_vs_5. As shown in Table 10, DoF algorithm outperforms other approaches, achieving state-of-the-art (SOTA) performance across these benchmarks.

### D.3.3   MPE

In Figure 6, we present the training curves of DoF-Policy (DoF-P) in the Multi-Agent Particle Environment (MPE). These graphs cover the performance across three different environments and four distinct datasets. We compare the training results of four algorithms: DoF-P, MAICQ, MA-TD3+BC, and OMAR. The solid lines in the graphs represent the mean values, while the shaded areas indicate the variance, providing insight into the central tendency and variability of the model performance. It is worth noting that the DoF-P algorithm displayed here employs the DoF-concat noise decomposition method.

Table 13: The Average Return of the Multi-Agent MuJoCo Benchmark

| Maps | Data | MACQL | MAICQ | OMAR | MA-TD3+BC | DoF |
|---|---|---|---|---|---|---|
| **HalfCheetah-v2** | Good | 2886.2±651.7 | 3044.2±311.4 | 3124.2±411.4 | **3412.7±281.3** | 3400.1±310.5 |
| | Medium | 1243.2±455.1 | 2621±281.4 | 2864.3±322.4 | 3011.2±178.6 | **3123.1±161.7** |
| | Poor | 1045.3±376.7 | 744.3±141.7 | **1968.1±141.7** | 1651.9±156.1 | 1869.9±129.8 |

As shown in Figure 6, it is evident that DoF-P achieves superior performance in the presented experimental results, consistently outperforming the MAICQ, MA-TD3+BC, MACQL, and OMAR algorithms. Notably, DoF-P demonstrates a faster convergence rate, particularly in the simple_world environment, where it converges in just 400,000 steps. In contrast, OMAR achieves the second-best results in most environments.

### D.3.4 MULTI-AGENT MUJOCO (MA MUJOCO)

Table 13 depicts the experimental results for the HalfCheetah task of MA-MuJoCo (de Witt et al., 2020). DoF performs the best in the Medium dataset, and ranks second in the Good and Poor datasets.

### D.3.5 SCALABILITY EVALUATION

We have developed a customized combat game based on MAgent (Zheng et al., 2018) environment, a grid-world specifically designed for large-scale multi-agent reinforcement learning. The combat game is a drone combat game, where multiple drones fight against other drones controlled by in-game AI. The goal of the game is to train RL controlled drones to defeat all the opponent drones. In this game, each drone agent has a 120° observation field and a smaller 120° attack range. It is a cooperative MARL game, all the agents share the same reward: 1 point for hitting an enemy drone, 10 for neutralizing one, and 50 for eliminating all enemy drones. Negative rewards are assigned to following scenarios: -1 for being hit, -10 for being neutralized, and -3 if no enemy drones are taken down in a timestep. This large-scale environment presents significant challenges in agent coordination, requiring drones to cooperate effectively to maximize enemy drone elimination while minimizing their own casualties. We consider multiple game scenarios with different number of drones fighting against the same number of agents. For example, the 64x64 scenario indicates 64 drones fighting against 64 opponents. For each scenario, we train agents using MADDPG (Lowe et al., 2017a) for 2,000 episodes (with 550 steps on average), and the replay logs are collected for offline training.

We conducted the Scalability Experiment: Comparison of DoF and MADIFF for Different Numbers of Agents in the main text. As shown in Table 3, this experiment compares GPU Memory, Inference Time Cost, and Reward for both DoF and MADIFF across different numbers of agents. Specifically, Inference Time Cost represents the total time taken for a single inference, which includes the time for the entire process of generating samples using the model, with a batch size of 32 and 200 diffusion steps. GPU Memory measures the memory usage for running the model, and Reward reflects the performance of the agents during the game. The results demonstrate that through diffusion factorization, DoF achieves better scalability than MADIFF. DoF consistently uses less GPU Memory and achieves faster Inference Time Cost compared to MADIFF, particularly as the number of agents increases. In terms of Reward, DoF maintains high performance, outperforming MADIFF in larger-scale scenarios. These results validate the efficiency and scalability of DoF in comparison to MADIFF.

To further evaluate the scalability, we compare the network parameter efficiency of MADIFF and DoF. Table 14 presents a comparison of network parameter counts (in MB) across different numbers of agents, aimed at evaluating the model's parameter efficiency and scalability. We measure the network parameter counts *per agent* to directly compare how both methods scale as the number of agents increases.

We also investigate the inference time costs of various baseline methods. Table 15 provides a comparison of inference time (in seconds) for DoF with enhanced sampling techniques alongside other baseline approaches. Specifically, we explore sampling acceleration techniques such as DDIM and Consistency Model, and include comparisons with non-diffusion methods (e.g., MACQL and MABCQ) to offer a more comprehensive view of the computational trade-offs involved.

Table 14: Network Parameter Count (in MB) Comparison Between DoF and MADIFF for Different Numbers of Agents

| Metric | Method | 4 agents | 8 agents | 16 agents | 32 agents | 64 agents |
|---|---|---|---|---|---|---|
| **Network Parameter Count** | MADIFF | 109 MB | 135 MB | 174 MB | 228 MB | 310 MB |
| | DoF | 71 MB | 72 MB | 76 MB | 81 MB | 91 MB |

Table 15: Inference Time Cost (s) for Different Methods Across Agent Configurations

| Metric | Method | 4 agents | 8 agents | 16 agents | 32 agents | 64 agents |
|---|---|---|---|---|---|---|
| | DoF(DDPM) | 8.2s | 11.3s | 14.9s | 18.1s | 24.3s |
| | DoF(DDIM) | 5.1s | 7.8s | 9.6s | 12.2s | 14.8s |
| **Inference Time Cost (s)** | DoF(consistency model) | 1.3s | 1.4s | 1.6s | 1.9s | 2.4s |
| | MACQL | 1.1s | 1.2s | 1.2s | 1.3s | 1.5s |
| | MABCQ | 1.1s | 1.2s | 1.3s | 1.4s | 1.6s |
| | MADIFF | 12.9s | 16.5s | 23.9s | 31.5s | Out Of Memory |

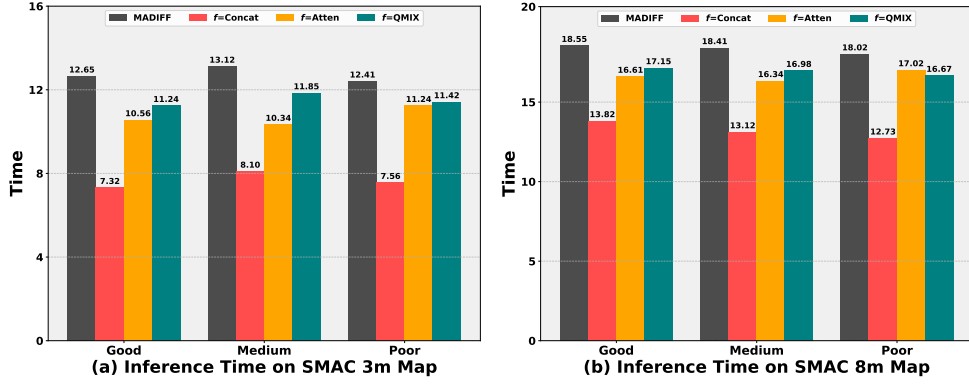

Figure 7: Diffusion Inference Time in SMAC Environment

Table 16 shows the reward comparisons using the same sets of methods. By comparing rewards across different sampling techniques (e.g., DDPM, DDIM, and Consistency Model) and non-diffusion methods, we can observe variations in both performance and efficiency. These experiments highlight the effectiveness of advanced sampling strategies and underscore the importance of selecting appropriate techniques under different computational constraints.

Table 16: Reward Comparison for Different Methods Across Agent Configurations

| Metric | Method | 4 agents | 8 agents | 16 agents | 32 agents | 64 agents |
|---|---|---|---|---|---|---|
| | DoF(DDPM) | 60.1 | 75.9 | 120.3 | 154.6 | 210.4 |
| | DoF(DDIM) | 61.3 | 73.8 | 118.5 | 151.7 | 208.3 |
| **Reward** | DoF(consistency model) | 55.3 | 70.2 | 116.3 | 148.9 | 202.1 |
| | MACQL | 50.7 | 65.6 | 100.3 | 135.1 | 190.6 |
| | MABCQ | 42.1 | 49.4 | 90.4 | 119.2 | 162.3 |
| | MADIFF | 63.8 | 70.4 | 113.5 | 148.3 | Out Of Memory |

### D.3.6 INFERENCE TIME

We evaluate the inference time, which refers to the duration taken by the diffusion model to complete one reverse process during sampling. This process involves $N$ diffusion steps, and for this experiment, we set $N = 200$. Our primary focus is to compare the inference times of DoF using different noise factorization function $f$ ($f$=Concat, $f$=Atten, and $f$=QMIX) and MADIFF.

Experiments were conducted on the 3m and 8m maps of the SMAC environment. The detailed results are shown in Figure 7. As illustrated, the inference time for $f$=Concat is faster than both $f$=Atten and $f$=QMIX. On the 3m map, executing a single reverse process takes approximately 7.32 seconds,

Table 17: Performance Comparison Across Different Maps and Methods

| Maps | Dataset | $f$ = Concat | $f$ = WConcat | $f$ = Atten |
|---|---|---|---|---|
| 3s5z_vs_3s6z | Good | 12.0±0.8 | 12.8±0.8 | 14.7±0.7 |
| | Medium | 10.4±0.7 | 11.9±0.7 | 12.6±0.6 |
| | Poor | 7.0±0.2 | 7.5±0.2 | 8.4±0.3 |
| 2c_vs_64zg | Good | 15.7±0.9 | 16.1±0.8 | 18.0±0.6 |
| | Medium | 13.3±0.8 | 13.9±0.9 | 14.7±0.7 |
| | Poor | 11.2±0.9 | 11.5±1.1 | 12.0±0.6 |

while on the 8m map, it takes around 13.82 seconds. The inference times for $f$=Attn and $f$=QMIX are similar, with around 11 seconds on the 3m map and approximately 17 seconds on the 8m map. MADIFF has the slowest inference time, taking around 12.65 seconds on the 3m map and 18.55 seconds on the 8m map. This experiment confirms that DoF achieves faster inference times than MADIFF, and that the noise factorization method $f$=Concat outperforms $f$=Atten and $f$=QMIX in terms of inference speed. $f$=Concat performs the fastest thanks to its ability to factor diffusion process.

### D.4 ABLATION STUDY

**How does the Noise Factorization Function $f$ Affect Agent Performance?** In the main text (see Table 4), we briefly compared different noise factorization functions $f$ and observed that decentralized functions like WConcat generally outperform simpler approaches such as Concat, while centralized attention-based functions Atten excel but require centralized execution.

To further investigate the impact of noise factorization functions $f$, Table 17 presents the results of using the noise factorization functions $f$ = WConcat and $f$ = Atten in the homogeneous 2c_vs_64zg and heterogeneous 3s5z_vs_3s6z environment. Notably, the centralized attention-based method ($f$ = Atten) demonstrates a clear performance advantage over the WConcat approach ($f$ = WConcat), particularly in this challenging heterogeneous setting.

**How does the Data Factorization Function $h$ Affect Agent Performance?** In this experiment, we compare three data factorization methods: Concat, WConcat, and Atten. Concat uses a simple concatenation operation to combine individual agent data; WConcat builds upon Concat by adding learnable weights; while Atten employs an attention mechanism for more sophisticated data integration. We conducted tests on the SMAC 3m, 5m_vs_6m, 3s5z_vs_3s6z, and 2s3z scenarios, using datasets of varying quality (Good, Medium, and Poor). As shown in Table 18, WConcat and Atten consistently outperform Concat across all four scenarios and dataset qualities. The performance gap is particularly noticeable in more complex maps, indicating that the weighted concatenation method WConcat and the attention-based approach Atten are more effective at capturing inter-agent relationships and extracting relevant information from the joint state.

**How Does Parameter Sharing Affect Agent Performance?** Similar to other MARL approaches, we use parameter sharing for the agent network. The parameters of the noise prediction network $\epsilon_i^\theta$ are shared among agents too. In this way, $\theta_i = \theta_j, \quad i \neq j$. To discern whether the performance improvement is due to parameter sharing or noise factorization, we study the performance of DoF without parameter sharing for the noise prediction network on the SMAC 3m and 5m_vs_6m scenarios. The results are depicted in Table 19. DoF without parameter-sharing is depicted as DoF (No_share). As shown in the Table 19, there is no significant difference between the parameter-sharing and the non-parameter-sharing approaches. This suggests that parameter sharing plays a minor role in multi-agent cooperation tasks.

**How sensitive is DoF to different hyper-parameters?** We investigate the impact of different hyper-parameters for the DoF trajectory. Figure 8 shows the results for different diffusion steps, history horizons, and conditional guidance scales. As shown in Figure 8 (a), Longer diffusion steps does not always lead to higher returns. Using a longer history horizon $H$ does not always lead to better performance, as it is depicted in Figure 8 (b). Horizon $H = 24$ performs better than $H = 36$. As it is depicted in Figure 8 (c), the conditional guidance scale does not lead to significant performance differences.

Table 18: Ablation Study on Data Factorization Functions $h$ Across Different Maps

| Maps | Dataset | $h$ = Concat | $h$ = WConcat | $h$ = Atten |
|------|---------|--------------|---------------|-------------|
| **3m** | Good | 19.7±0.6 | 19.8±0.2 | 19.9±0.1 |
| | Medium | 17.8±2.1 | 18.6±1.2 | 18.7±1.0 |
| | Poor | 10.6±1.6 | 10.9±1.1 | 10.8±0.9 |
| **5m_vs_6m** | Good | 15.8±1.4 | 17.7±1.1 | 18.2±0.9 |
| | Medium | 14.9±1.1 | 16.2±0.9 | 16.8±0.8 |
| | Poor | 9.8±1.1 | 10.8±0.3 | 11.0±0.5 |
| **3s5z_vs_3s6z** | Good | 11.3±0.9 | 12.8±0.8 | 15.2±0.7 |
| | Medium | 9.4±0.7 | 11.9±0.7 | 12.8±0.5 |
| | Poor | 6.8±0.3 | 7.5±0.2 | 8.2±0.3 |
| **2s3z** | Good | 15.5±1.0 | 18.5±0.8 | 19.5±0.3 |
| | Medium | 14.8±0.8 | 18.1±0.9 | 18.5±0.3 |
| | Poor | 9.6±1.1 | 10.0±1.1 | 10.2±0.7 |

Table 19: DoF w/wo for Parameter Sharing

| Maps | Dataset | Share | No_share |
|------|---------|-------|----------|
| **3m** | Good | 19.8±0.2 | 19.7±0.5 |
| | Medium | 18.6±1.2 | 18.1±0.7 |
| | Poor | 10.9±1.1 | 11.2±0.4 |
| **5m_vs_6m** | Good | 17.7±1.1 | 17.5±0.8 |
| | Medium | 16.2±0.9 | 16.3±0.7 |
| | Poor | 11.4±0.7 | 10.8±0.9 |

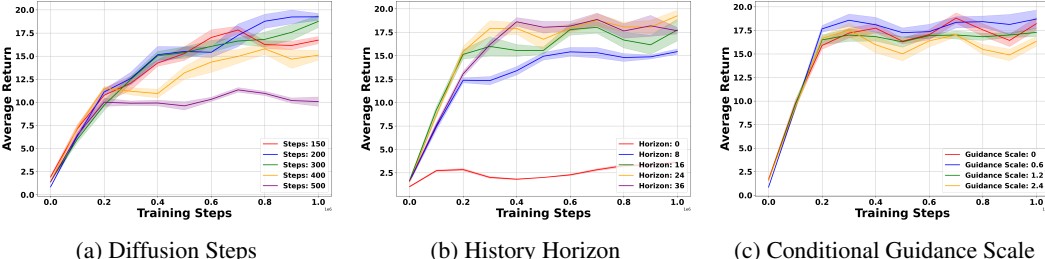

| (a) Diffusion Steps | (b) History Horizon | (c) Conditional Guidance Scale |
|---|---|---|

Figure 8: Sensitive analyze: (a) Diffusion Steps, (b) History Horizon, and (c) Conditional Guidance Scale.

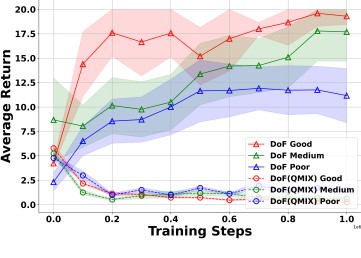

(a) Linear vs. Non-linear

Figure 9: Comparison of linear and non-linear noise combination methods in the SMAC 3m environment. The non-linear method underperforms due to non-Gaussian noise profiles. Each subplot shows the average episode reward over training steps. The shaded regions indicate the standard deviation of the rewards.

**Can we use Monotonic-increasing Noise Factorization Methods?** The normal distribution is linearly additive, so we can use the linear combination of noises in DoF to form a larger noise. However, it is unclear whether monotonic increasing mixers such as QMIX (Rashid et al., 2018) can be used for noise factorization. To investigate the impact of the monotonic increasing noise

factorization combination method, we conducted experiments in the SMAC 3m environment using good, medium, and poor datasets. In this experiment, we use QMIX as the noise factorization method. It is depicted as DoF (QMIX) in Figure 9. As it is shown in the figure, using the monotonic increasing function as the mixer hurt the performance of DoF significantly. This is due to the fact that through using such a monotonic increasing mixer, the resulting noise may no longer be Gaussian noise, which is required during the diffusion process.

## D.5 DIFFERENT DIFFUSION GENERATION PROCESSES

The diffusion model is computationally intensive despite its flexible modeling ability; in DoF, we leverage the modeling ability of the diffusion model to model the cooperative behaviors among agents. As a generation-based MARL approach, our work relies on generation models to generate data. In this work, we use DDPM (Ho et al., 2020). The major testing time of DDPM is spent on the long diffusion steps to sample data. Our work can be built on recent advancements of the diffusion model to accelerate the sampling time. For example, DDIM (Song et al., 2021a) can reduce the sampling steps significantly with a performance drop. The consistency model (Song et al., 2023) requires only one sampling step.

The experimental results for DoF and MADIFF using DDPM and DDIM on SMAC 3m, using different steps, are shown in Table 20 and Table 21. As shown in Table 20, the testing time is lower with fewer diffusion steps. The diffusion step 50 achieves the lowest testing time at the cost of the lowest return. The testing time of MADIFF is always longer than that of DoF. And MADIFF always performs weaker than DoF. For Table 21, it shows that using DDIM leads to faster testing time. We observe similar trends demonstrated in Table 20.

Table 20: Comparison for DoF and MADIFF across Diffusion Steps with DDPM in SMAC 3m.

| Diffusion Step | Time (DoF) | Time (MADIFF) | Reward (DoF) | Reward (MADIFF) |
|---|---|---|---|---|
| 50 | 515.4 | 689.4 | 13.5 | 12.7 |
| 100 | 625.8 | 925.8 | 16.4 | 15.1 |
| 200 | 1018.8 | 1413.6 | 19.3 | 19.1 |
| 300 | 1492.8 | 1679.4 | 19.2 | 19.0 |

Table 21: Comparison for DoF and MADIFF Across Diffusion Steps with DDIM in SMAC 3m.

| Diffusion Step | Cost Time (DoF) | Cost Time (MADIFF) | Reward (DoF) | Reward (MADIFF) |
|---|---|---|---|---|
| 50 | 367.2 | 475.2 | 11.8 | 11.3 |
| 100 | 533.4 | 738.6 | 15.8 | 14.7 |
| 200 | 813.6 | 1185.6 | 19.1 | 19.0 |
| 300 | 1124.4 | 1304.4 | 18.9 | 18.8 |

The training time and testing time of DoF can be further reduced by following the approach of StableDiffusion (Rombach et al., 2022) and LatentDiffusin (Venkatraman et al., 2024). They use a Variation Auto Encoder (VAE) to encode input data into latent space with a lower dimensions than before. The diffusion is happens in the latent space. In the end of diffusion, the data is recovered using a decoder from the latent space. We implement this idea and named it as DoF+VAE, and conduct experiment on the SMAC 3m good dataset. The VAE compress the data from 33 dimensions to 17 dimensions. The experimental results are shown in table 22.

Table 22: Comparison of Training Time and Return for DoF and DoF+VAE on SMAC 3m Good Dataset

| Method | Training Time | Return |
|---|---|---|
| DoF | 48h | 18.96 |
| DoF+VAE | 39h | 15.80 |

Table 22 shows that with the use of VAE, the training time is reduced from 48h to 39h, a decrease of approximately 27%, with a 16.6% decline in performance. This suggests that within an acceptable range of performance loss, using a VAE to compress data can effectively reduce training time and

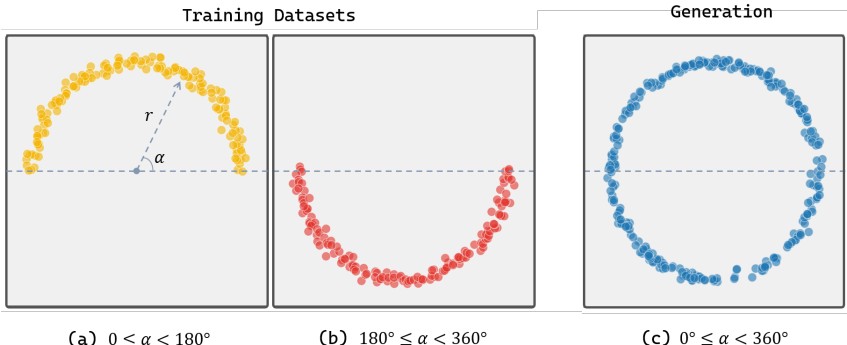

Figure 10: DoF is able to generate novel behaviors through using multiple constraints. (a) Dataset 1: data is located in upper half-circle ($0° \leq \alpha < 180°$), (b) Dataset 2: data is located in lower-half-circle $180° \leq \alpha < 360°$, and (c) Generated data: full circle $0° \leq \alpha < 360°$

enhance training efficiency, particularly in scenarios where resources are limited or there is a high demand for rapid training.

### D.6  SATISFY MULTIPLE CONSTRAINTS

Consider a multi-agent system comprising two agents: one responsible for learning the angle $\alpha$, and the other for learning the radius $r$. These two agents work collaboratively to generate point distributions in a two-dimensional plane. We are given two datasets, each representing different constraint conditions. As shown in Figure 10: (a) $0° \leq \alpha < 180°$: In the first dataset, the angle agent is constrained to learn within the upper half-circle. (b) $180° \leq \alpha < 360°$: In the second dataset, the angle agent is constrained to learn within the lower half-circle. (c) $0° \leq \alpha < 360°$: In the generation phase, we demonstrate how the two agents collaborate to generate a complete circular distribution.

In this work, we train two diffusion models for the two datasets: model A for the upper half-circle dataset, with constraint $(1, 0)$, model B for the lower half-circle dataset, with constraint $(0, 1)$. Each model can be factored into two diffusion models for generating data. After factorization, one model is used to generate the radius $r$, and another model is used to generate the angle $\alpha$. During generation, we utilize the model A and B to generate novel data, the full circle, with the constraint $(1, 1)$. We follow the approach of Decision Diffusion (Please refer to its Appendix D) to guide the factored diffusion processes to generate data following multiple constraints.

Figure 10 illustrates that the multi-constraints diffusion processes generate a point distribution covering the entire circle. This example demonstrates the flexibility and adaptability of our approach in handling multi-agent systems, where different agents are responsible for different parameters but can collaboratively satisfy complex constraint conditions.

### D.7  CONDITIONING ON LOCAL OBSERVATIONS

After the diffusion processes satisfying the IGD principle are learned, they can be used to generate data with desired properties with guidance. Researchers (Ho & Salimans, 2021) have shown that using classifier-free guidance can lead to better performance. In this work, we adopt classifier-free guidance to guide the agent to learn cooperative behaviors.

For diffusion process $i$, we use condition $y_i$ to guide the generation process toward desired properties. In cooperative MARL a high-return value $R$ suggests cooperative behaviors, thus $R$ is included into $y_i$ to guide the diffusion process to generate high-return data. Further, we include the local observation history $\tau_i$ of each agent into the condition $y_i$ to make the generate data aligns with its local observation.

Table 23 presents an ablation study on the condition of our proposed method. We compare the performance of DoF with two different types of conditions: return value $R$ only, return $R$ and local observation $\tau_i$. The results demonstrate that using both $R$ and $\tau_i$ as condition $y_I$ consistently

Table 23: Ablation Study on Condition Components

| Datasets | DoF $R$ | DoF $R$ and $\tau_i$ |
|---|---|---|
| **3m medium** | $7.34 \pm 0.89$ | $18.58 \pm 1.22$ |
| **5m6m medium** | $4.79 \pm 0.64$ | $16.22 \pm 0.83$ |
| **8m medium** | $6.66 \pm 0.86$ | $18.64 \pm 0.86$ |

outperforms the return-only condition across all tested scenarios. This justify the selection of using both $R$ and $\tau_i$ as guidance to guide generation process toward desired cooperative behaviors.

## E  DISCUSSION

### E.1  DIFFUSION-BASED METHODS AND OUT-OF-DISTRIBUTION (OOD) SCENARIOS

Diffusion-based methods exhibit certain advantages in handling out-of-distribution (OOD) situations. When using DoF-Trajectory, from the perspective of trajectory modeling, diffusion methods bypass the traditional Q-value estimation step commonly employed in reinforcement learning. This bypass naturally alleviates the issue of Q-value overestimation, which is particularly prominent in offline scenarios with limited or biased data. By mitigating this overestimation, diffusion methods effectively address OOD challenges. On the other hand, when using DoF-Policy, from the perspective of policy modeling, diffusion models model the distribution of the underlying policy well. In this way, it effectively reduces extrapolation errors caused by sampling out-of-distribution actions.

### E.2  POTENTIAL APPLICATIONS IN COMPETITIVE AND MIXED-MOTIVATION SCENARIOS

While our current work focuses on cooperative multi-agent reinforcement learning tasks, extending diffusion-based methods to competitive or mixed-motivation scenarios is a valuable direction for future research. In competitive settings, the diffusion process could condition on adversarial strategies or payoff structures during sampling to model the interplay between competing agents. For mixed-motivation scenarios, diffusion methods could incorporate both individual motivations and shared objectives into the conditional space.

This enhancement would allow the model to capture the complexity and dynamics of such environments more effectively. For example, in a competitive game, the diffusion model could learn to anticipate and counter adversarial moves, while in mixed-motivation tasks, it could optimize individual agent goals while ensuring collective success. Incorporating motivations into the diffusion conditions provides a flexible mechanism to extend the applicability of our framework to these challenging scenarios, paving the way for broader applications in multi-agent systems.

## F  LIMITATION

Our method may face challenges when applied to tasks with high-dimensional observations. While our current experiments focus on standard MARL benchmarks, extending the approach to handle more complex, high-dimensional observation spaces remains an open direction for future work.

