# OpenReview forum: "DoF: A Diffusion Factorization Framework for Offline Multi-Agent Reinforcement Learning"
_ICLR.cc/2025/Conference — ICLR 2025 Poster_

### Official Review · Reviewer_YHzY · 2024-10-23

**Soundness:** 3
**Presentation:** 2
**Contribution:** 3
**Rating:** 8
**Confidence:** 2

**Summary:**

The paper studies the use of diffusion models in offline cooperative multi-agent decision-making (MADM). The authors propose the Individual-Global-identically-Distribute (IGD) principle, a generalization of the IGM principle. IGD requires that collectively generated outcome of individual agents follows the same distribution as the generated outcome of a whole multi-agent system. The authors propose DoF, a diffusion factorization framework for offline MADM which adopts a noise factorization function to factorize a centralized diffusion model into multiple diffusion models. The authors show theoretically that the noise factorization functions in DoF satisfy the IGD principle. Experiments are used to shown the effectiveness of DoF.

**Strengths:**

- The paper provides a novel definition of the IGD principle to generalize the IGM for the use of diffusion models.
- The authors theoretically show that the proposed DoF added with particular factorization functions satisfy the IGD principle.
- Strong empirical section showing the effectiveness of the proposed method

**Weaknesses:**

Potential limitations of noise and data factorization functions - the primary options used in the paper are Concat and Wconcat, which might lack modeling power.

**Questions:**

- Table 6 compares decentralized methods (WConcat) with 'centralized' methods (attention), showing similar performance. To properly interpret these results:
  - How is the centralized attention method implemented during execution?
  - What observations/information does centralized attention have access to versus decentralized methods? Is centralized attention meant to serve as an upper-bound baseline (with access to information not available in real deployments) rather than a competing method?
- The authors claim in section 4.3.2 that 'the data factorization h could model more powerful data relationships than the noise factorization function f'. However, Table 14 shows minimal performance improvement when using attention (a more complex function) versus WConcat for data factorization h. Are there theoretical or empirical evidence that supports the claim that a more complex h can compensate for using simple noise factorization functions f (as is mainly used in the paper)?

---

> ### Author Response · Authors · 2024-11-18
>
> We would like to express our gratitude for your time and effort in reviewing our work. Thanks for recognizing the novelty of the IGD principle that generalizes IGM, the theoretical proof of DoF satisfying the IGD principle, and the solid empirical experiments. We address your reviews as follows.
>
>
> >Potential limitations of noise and data factorization functions - the primary options used in the paper are Concat and Wconcat, which might lack modeling power.
>
> Thank you for your insightful comment. We agree that Concat and Wconcat have limited modeling power, they are preferred due to their ability to satisfy the IGD property and conform to the Centralized Training with Decentralized Execution (CTDE) paradigm. In addition to using Concat and Wconcat for noise and data factorization, we also explored alternative methods such as QMix, which has better modeling ability.
>
>
> QMix is a famous value factorization method, it models the monotonic increasing relationship among individual value function $Q_i$ and global value function $Q_{tot}$. For the QMix-based DoF, it uses the QMix function to factorize the noise. Each individual noise $\epsilon_i$ is mixed with a learned monotonic increasing function into the global noise $\epsilon_{tot}$. QMix-based DoF performs poorly in diffusion tasks, as shown in Table 6.
>
> When using the QMix function as the noise factorization function, it has two drawbacks: non-Gaussian distribution and non-decentralized execution. For the non-Gaussian distribution drawback, QMix models the monotonic relationship among noises. It does not restrict the monotonic relationship as linear additive relationship. Thus, the joint noise $\epsilon_{tot}$ modeled by QMix cannot guarantee to follow the Gaussian distribution. As DoF is built on the Diffusion model which assumes that the noise follows the Gaussian distribution, this violation of the Gaussian distribution assumption leads to poor performance, as shown in Table 6 (column QMix). The non-decentralized execution drawback is that the agent of QMix-based DoF cannot be decentrally executed. In each diffusion step, each agent relies on the noise and the data of other agents.
>
> Therefore, we believe Concat and Wconcat offer a better trade-off between performance, scalability, and theoretical constraints. The IGD principle proposed in our paper highlights an important direction for future research. We hope this principle inspires further exploration, potentially leading to more powerful and scalable noise factorization and data factorization functions in the future.
>
> > Table 6 compares decentralized methods (WConcat) with 'centralized' methods (attention), showing similar performance. To properly interpret these results:
> > How is the centralized attention method implemented during execution?
>
> Thank you for your insightful question. Centralized attention adopts a centralized training and centralized execution Paradigm, as detailed in the explanation of the noise factorization method using the Atten implementation in Appendix C.1.3. The noise factorization function is expressed as:
>
> $$
> \epsilon_{t o t}\left(\mathbf{x}^k, \mathbf{y}, k\right)[(i-1) \times d: i \times d]=\sum_{j=1}^N w_i^j \epsilon_j\left(\mathbf{x}_{\mathbf{j}}^k, \mathbf{y}, k\right)
> $$
>
> Where  $w_i^j$  represents a set of weights learned through an attention mechanism. Its implementation is similar to Wconcat.
>
>
> While the 3m and 5m_vs_6m homogeneous environments (as shown in Table 6)  are relatively simple, resulting in comparable performance between $f=Wconcat$ and $f=Atten$,   We conducted additional experiments in more challenging map environments, including 2c_vs_64zg (homogeneous) and 3s5z_vs_3s6z (heterogeneous), to better illustrate the differences. The performance comparison is shown below：
>
> | Maps | Dataset  | $f=\text{Concat}$ |$f = \text{WConcat}$ | $f = \text{Atten}$ |
> |------|----------------|-------|-------------------------|-------------------------|
> |      | Good     |12.0±0.8 |12.8±0.8             | 14.7±0.7         |
> |  3s5z_vs_3s6z    | Medium| 10.4±0.7 | 11.9±0.7               | 12.6±0.6     |
> |      | Poor      | 7.0±0.2| 7.5±0.2  | 8.4±0.3              |
> |   | Good    | 15.7±0.9|16.1±0.8                | 18.0  ± 0.6       |
> |  2c_vs_64zg     | Medium |13.3±0.8 |  13.9±0.9                | 14.7±0.7               |
> |      | Poor     | 11.2±0.9|11.5±1.1           | 12.0±0.6               |
>
>
> From the table, it is evident that the centralized attention method ($f=Atten$) outperforms Wconcat ($f=Wconcat$) in both the homogeneous 2c_vs_64zg and the heterogeneous 3s5z_vs_3s6z environments.

---

> ### Author Response · Authors · 2024-11-18
>
> > What observations/information does centralized attention have access to versus decentralized methods? Is centralized attention meant to serve as an upper-bound baseline (with access to information not available in real deployments) rather than a competing method?
>
>
> Thank you for your question. In the case of centralized attention, during centralized training, agent $i$ generates noise based on the actions and observations of other agents. However, during execution, due to the requirement for consistency between forward noise addition and denoising, agents must still access the observations of other agents. It means that agents  must be executed centrally rather than decentrally.
>
> In contrast, decentralized methods like Concat and WConcat require only the noise information from agent $i$ itself during execution, without the need for information from other agents.
>
> DoF seeks to address the scalability issue of diffusion-based MARL. **We agree with the reviewer's observation that centralized attention represents an upper bound on achievable reward, as it utilizes information beyond what decentralized methods can access.** Although centralized attention  could achieve higher rewards, it lacks the scalability of Concat and WConcat due to increased computational and memory demands, which makes Concat and WConcat the lower-bound in terms of scalability.

---

> ### Author Response · Authors · 2024-11-18
>
> > The authors claim in section 4.3.2 that 'the data factorization h could model more powerful data relationships than the noise factorization function f'. However, Table 14 shows minimal performance improvement when using attention (a more complex function) versus WConcat for data factorization h. Are there theoretical or empirical evidence that supports the claim that a more complex h can compensate for using simple noise factorization functions f (as is mainly used in the paper)?
>
>
> Thank you for your question. Our claim that a more complex data factorization $h$ can compensate for the simpler noise factorization function $f$ is supported by an illustrative example and empirical evidence.
>
> First, in Table 11, different data factorization $h$ are employed to reconstruct a one-step payoff matrix. The results show that $h=Atten$ reconstructs a better payoff matrix than other functions.
>
>
> Second, Table 14 shows the results for the 3m and the 5m vs 6m scenarios, where agents are homogeneous. Although it shows that $h=Atten$ is slightly better than $h=Wconcat$, it is not significant. This is due to the fact that in such a homogeneous setting, a linear weighted relationship among agents could be enough. To further validate the effect of a more complex data factorization function, we test these methods in a heterogeneous 2s3z and 3s5z_vs_3s6z environment, where agents are heterogeneous. The results are presented below:
>
>
> | Maps | Dataset | $h = \text{Concat}$ | $h = \text{WConcat}$ | $h = \text{Atten}$ |
> |------|---------|-----------------------|-------------------------|-------------------------|
> |   | Good    | 11.3±0.9            | 12.8±0.8     |      15.2±0.7       |
> |  3s5z_vs_3s6z     | Medium  | 9.4±0.7            | 11.9±0.7    |     12.8±0.5          |
> |      | Poor    | 6.8±0.3    | 7.5±0.2|    8.2±0.3       |
> |   | Good    | 15.5±1.0              | 18.5±0.8                | 19.5±0.3               |
> |  2s3z     | Medium  | 14.8±0.8              | 18.1±0.9                | 18.5±0.3               |
> |      | Poor    | 9.6±1.1               | 10.0±1.1                 | 10.2±0.7               |
>
> As shown in both Table 14 and the heterogeneous 2s3z and 3s5z_vs_3s6z environment, the choice of data factorization $h$ has a significant impact on performance. The trend $h = \text{Atten} > h = \text{WConcat} > h = \text{Concat}$ demonstrates that **more complex data factorization methods provide a clear advantage, supporting our claim**.
>
> To make it easier for the reviewer to examine the results, we have included both Table 11 and Table 14 directly below.
>
> Table 11: Payoff matrix of one-step matrix games and reconstructed value functions to approximate the optimal policy.
>
> | u1\u2 | A   | B   |
> |-------|-----|-----|
> | A     | 1.0 | 0.0 |
> | B     | 18.0| 1.0 |
>
> (a) Game Payoff Matrix 1
>
> | Q1\Q2 | A   | B   |
> |-------|-----|-----|
> | A     | 0.6 | 0.0 |
> | B     | 19.0| 0.4 |
>
> \(b\) $h = \text{Concat}$
>
> | Q1\Q2 | A   | B   |
> |-------|-----|-----|
> | A     | 0.9 | 0.0 |
> | B     | 17.9| 1.2 |
>
> \(c\) $h = \text{Atten}$
>
> | u1\u2 | A   | B   |
> |-------|-----|-----|
> | A     | 4.0 | 0.0 |
> | B     | 14.0| 2.0 |
>
> \(d\) Game Payoff Matrix 2
>
> | Q1\Q2 | A   | B   |
> |-------|-----|-----|
> | A     | 3.2 | 0.2 |
> | B     | 15.2| 1.4 |
>
> \(e\) $h = \text{Concat}$
>
> | Q1\Q2 | A   | B   |
> |-------|-----|-----|
> | A     | 4.0 | 0.0 |
> | B     | 13.9| 2.1 |
>
> \(f\) $h = \text{Atten}$
>
>
> Table 14: Ablation Study Data Factorization functions $h$
> | Maps       | Dataset | $h = \text{Concat}$ | $h = \text{WConcat}$ | $h = \text{Atten}$ |
> |------------|---------|------------------------|--------------------------|-------------------------|
> |          | Good    | 19.7±0.6              | 19.8±0.4                | 19.8±0.2               |
> |   **3m**      | Medium  | 17.8±2.1              | 18.5±1.4                | 18.6±1.2               |
> |            | Poor    | 10.6±1.6              | 10.8±1.0                | 10.9±1.1               |
> |        | Good  | 15.8±1.4              | 17.5±1.3                | 17.7±1.1               |
> |  **5m_vs_6m**           | Medium  | 14.9±1.1              | 16.0±1.0                | 16.2±0.9               |
> |            | Poor    | 9.8±1.1               | 10.9±0.5                | 10.8±0.3               |
>
> We hope this presentation helps clarify our results and makes it easier for the reviewer to assess the evidence.

---

### Official Review · Reviewer_y4iS · 2024-11-04

**Soundness:** 3
**Presentation:** 2
**Contribution:** 3
**Rating:** 5
**Confidence:** 3

**Summary:**

The paper presents Diffusion Factorization Framework (DoF), a new method for offline cooperative multi-agent decision-making (MADM) using diffusion models. DoF generalizes the traditional Individual-Global-Max (IGM) principle with a new Individual-Global-identically-Distributed (IGD) principle. This IGD principle aligns decentralized outcomes in multi-agent diffusion models with centralized outcomes, fostering effective cooperation. DoF includes two key functions: a noise factorization function that decentralizes a centralized diffusion model per the IGD principle, and a data factorization function that models inter-agent data relationships to ensure alignment with ground truth. Extensive experiments demonstrate DoF’s advantages in scalability, cooperation, and alignment with ground truth, outperforming existing MADM methods across various benchmarks.

**Strengths:**

1.	The paper generalizes the well-known IGM principle to the IGD principle, which broadens the applicability of diffusion models in multi-agent decision-making and improves coordination among agents. It addresses the limitations of existing cooperative MADM methods, enhancing scalability and cooperative performance.
2.	DoF introduces novel noise and data factorization methods that align agent-generated data with a centralized diffusion model while enabling decentralized execution. This factorization process addresses the complexity of multi-agent coordination and enhances resource efficiency across a range of multi-agent tasks.
3.	The extensive experiments conducted on the SMAC and MPE show that DoF outperforms competing methods in multi-agent coordination tasks. Experimental results indicate that DoF performs well with an increasing number of agents. This scalability suggests DoF’s potential for large-scale applications in complex multi-agent environments.

**Weaknesses:**

1.	The paper lacks clarity in some important parts. It introduces DoF-Policy and DoF-Trajectory but does not specify the suitable scenarios for each. The results section also fails to indicate which algorithm was used, and Figure 6 only includes DoF-Policy results, making it difficult to assess the relative strengths of the two methods. Additionally, implementation details for the inverse dynamics model in DoF-Trajectory are missing.
2.	In Figure 2(a), there is a mislabeling of the reverse diffusion process: the output should be  x^{k-1} instead of x^k , which may mislead the understanding of the process. Additionally, the caption of Figure 2 lacks necessary descriptions, which reduces its readability.
3.	The paper's benchmarks are based on state vectors, making it unclear whether the proposed method can be applied to environments with high-dimensional observations. The Scalability Experiment section only compares GPU memory and time with MADIFF, without providing comparisons to other baseline algorithms. Since diffusion-based methods require multiple iterations to generate the final output, does this lead to significantly higher computational costs compared to traditional methods?

**Questions:**

1.	Given that QMIX is an online algorithm not designed for offline reinforcement learning, could you clarify the rationale for including it in the comparisons? What insights does this comparison provide?
2.	Can diffusion-based methods effectively handle out-of-distribution situations? Additionally, can these methods be applied to competitive or mixed-motivation scenarios?

Please respond to the aforementioned weaknesses.

---

> ### Author Response · Authors · 2024-11-18
>
> Thanks for your time and effort to read our work. We highly appreciate your comments about the novelty of the IGD principle, the usefulness of the noise factorization and data factorization, and the extensitive experimental results.
>
>
>
> > The paper lacks clarity in some important parts. It introduces DoF-Policy and DoF-Trajectory but does not specify the suitable scenarios for each. making it difficult to assess the relative strengths of the two methods. The results section also fails to indicate which algorithm was used, and Figure 6 only includes DoF-Policy results.
>
>
> Sorry for the confusion, most of the results reported are DoF-Trajectory. In the experimental section, we had written "the generated data are trajectories." in line 343 of the initial subsmission. It was written to indicate that most of the results are from DoF-Trajectory. However, it is not work as expected. We have updated the text as "in default, the results for DoF-Trajectory are reported." in the experiment section. Specifically, the experiments for SMAC, SMAC v2, and MPE in the main text use the DoF-Trajectory agent, while Appendix D.3, Table 13, and the corresponding figures present results with the DoF-Policy agent.
>
> The core contribution of our work is the IGD principle and the diffusion factorization methods (noise factorization and data factorization). The DoF-Trajectory and DoF-Policy agents are developed to demonstrate that DoF can be used for both the trajectory-generation and the policy-generation cases. We do not present all the results of DoF-policy and DoF-trajectory as we want to focus on the factorization ability of DoF rather than the ability of a specific agent.
>
> In essence, DoF-Policy is a combination of DoF factorization functions with the DiffusionQL[1] agent. And DoF-Trajectory is a combination of DoF factorization functions with the Decision Diffuser[2] agent. We have added the following sentence *"To demonstrate the flexibility of DoF, we implement two agents based on agents of Decision Diffuser and DiffusionQL, respectively. "* into Sec 4.4.  DiffusionQL and Decision Diffuser are two seminar Diffusion-based approaches. DoF-Policy models the relationship between environment states and actions, making it suitable for continuous action spaces. DoF-Trajectory models trajectories and plans based on them. It is suitable for both continuous and discrete action environments. **In most cases, DoF-trajectory is recommended.** However, it needs to learn the inverse dynamic model which could be slower than DoF-policy.
>
>
> > The results section also fails to indicate which algorithm was used, and Figure 6 only includes DoF-Policy results.
>
> Following your advice, we have conducted experiments comparing DoF-Trajectory and DoF-Policy in MPE tasks, the experimental results are shown in the following table. It will be included in the appendix E.4 (Table 20) for completeness.
>
> | Dataset      | Task   | DoF-Trajectory        | DoF-Policy          |
> |--------------|--------|----------------|----------------|
> | Expert       | Spread | 126.4±3.9      | 126.3±3.1      |
> |              | Tag    | 125.6±8.6      | 120.1±6.3      |
> |              | World  | 132.2±19.1     | 138.4±20.1     |
> | Md-Replay    | Spread | 57.4±6.8       | 48.1±3.6       |
> |              | Tag    | 65.4±12.5      | 51.7±10.1      |
> |              | World  | 58.6±10.4      | 58.1±11.5      |
> | Medium       | Spread | 75.6±6.8      | 60.5±8.5       |
> |              | Tag    | 86.3±12.5      | 83.9±9.6       |
> |              | World  | 85.2±11.2      | 86.4±10.6      |
> | Random       | Spread | 35.9±6.8       | 34.5±5.4       |
> |              | Tag    | 16.5±6.3       | 14.5±3.2       |
> |              | World  | 13.1±2.1       | 15.1±3.0       |
>
>
> > Additionally, implementation details for the inverse dynamics model in DoF-Trajectory are missing.
>
> The inverse dynamics model used in DoF-Trajectory is taken from the Decision Diffuser [2], with details provided in Appendix C.3 (C.44) and (C.45) of the initial submission. This specific model is not included in the main text as it is not the central contribution of this work. We have added "Please refer the details of the agents in Appendix C.3." in Sec 4.4.
>
> **Reference**
>
> [1] Zhendong Wang, Jonathan J. Hunt, and Mingyuan Zhou. Diffusion policies as an expressive policy class for offline reinforcement learning. In ICLR 2023.
>
> [2] Ajay, Anurag, et al. "Is Conditional Generative Modeling all you need for Decision Making?" ICLR 2023.

---

> ### Author Response · Authors · 2024-11-18
>
> > In Figure 2(a), there is a mislabeling of the reverse diffusion process: the output should be $x^{k-1}$ instead of $x^k$ , which may mislead the understanding of the process. Additionally, the caption of Figure 2 lacks necessary descriptions, which reduces its readability.
>
> Thank you for pointing this out. There was a mislabeling in Figure 2(a), where the output should have been labeled as $x^{k-1}$ instead of $x^k$. We have corrected this error in the revised manuscript. Additionally, we have updated the caption of Figure 2 to provide more detailed descriptions to improve its clarity and readability. The updated caption is described as follows:
>
> Figure 2: DoF Overview: (a) Diffusion Factorization: in each diffusion (forward and backward) step $1\le k \le K$, the noise factorization function $f$ is used to factorize the noises $\epsilon_i^k$ and intermediate data $x_i^{k-1}$. In the last backward step $k=0$, we apply the data factorization function $h$ to model the complex relationship among data generated by each agent. (b) DoF Trajectory Agent: generating trajectory for planning. \(c\) DoF Policy Agent: generating actions for execution.
>
>
> > The paper's benchmarks are based on state vectors, making it unclear whether the proposed method can be applied to environments with high-dimensional observations. The Scalability Experiment section only compares GPU memory and time with MADIFF, without providing comparisons to other baseline algorithms. Since diffusion-based methods require multiple iterations to generate the final output, does this lead to significantly higher computational costs compared to traditional methods?
>
> Thank you for raising this question. We fully recognize the importance of high-dimensional observation environments in practical applications and understand the challenges they present. In the field of multi-agent reinforcement learning, research typically focuses on learning cooperative strategies. The use of state vectors is a widely adopted approach, particularly in typical multi-agent environments such as SMAC, MPE. Compared to directly processing high-dimensional image inputs, state vectors indeed have lower dimensionality, which introduces certain limitations to the applicability of our approach. However, this is not only a limitation of our method but also a common challenge in the current state of the MARL field. We hope that there could be more advanced MARL environments that enable us to evaluate the performance of DoF with vision-based observations. We have added such discussion into the limitaion section in Appendix E.2.
>
>
> We agree with the reviewer's concerns about computational costs. In our current scalability experiments, we mainly compared with MADIFF, another diffusion-based method, in terms of inference speed and GPU memory usage. Indeed, compared to traditional methods, diffusion-based approaches require multiple sampling iterations, which incur higher computational overhead. To provide a more comprehensive evaluation of performance, we have evaluated DoF and MADIFF in terms of the neural network parameters of their agents, and we have studied the impact of different accelerated sampling methods and compared them with two non-diffusion methods.
> 1. Parameter Efficiency: We measure network parameter counts per agent to evaluate the model's efficiency and scalability. The following table shows the parameter counts for DoF and MADIFF agent network with different numbers of agents.
>
> | Metric               | Method                   | 4 agents | 8 agents | 16 agents | 32 agents | 64 agents |
> |----------------------|--------------------------|----------|----------|-----------|-----------|-----------|
> | **Network Parameter Count** | MADIFF            |  109 MB   | 135 MB   |  174 MB   | 228 MB     | 310 MB   |
> |                      | DoF               | 71 MB   | 72 MB    | 76 MB   |  81 MB | 91 MB     |

---

> ### Author Response · Authors · 2024-11-18
>
> 2. DoF with enhanced sampling techniques and more baseline comparison: We explore sampling acceleration techniques, including DDIM[3] and Consistency Model[4], and added comparisons with non-diffusion methods (e.g., MACQL[1], MABCQ[2]) to better understand the computational trade-offs.
>
> Inference Time Costs (in seconds) and reward：
> | Metric               | Method                   | 4 agents | 8 agents | 16 agents | 32 agents | 64 agents |
> |----------------------|--------------------------|----------|----------|-----------|-----------|-----------|
> | **Inference Time Cost(s)** | DoF(DDPM)                | 8.2s     | 11.3s    | 14.9s     | 18.1s     | 24.3s     |
> |                      | DoF(DDIM)                | 5.1s     | 7.8s     | 9.6s      | 12.2s     | 14.8s     |
> |                      | DoF(consistency model)   | 1.3s     | 1.4s     | 1.6s      | 1.9s      | 2.4s      |
> |                      | MACQL                     | 1.1s     | 1.2s     | 1.2s      | 1.3s      | 1.5s      |
> |                      | MABCQ                    | 1.1s     | 1.2s     | 1.3s      | 1.4s      | 1.6s      |
> | |MADIFF| 12.9s | 16.5s | 23.9s | 31.5s | Out Of Memory |
>
> | Metric               | Method                   | 4 agents | 8 agents | 16 agents | 32 agents | 64 agents |
> |----------------------|--------------------------|----------|----------|-----------|-----------|-----------|
> | **Reward**           | DoF(DDPM)                | 60.1     | 75.9     | 120.3     | 154.6     | 210.4     |
> |                      | DoF(DDIM)                | 61.3     | 73.8     | 118.5     | 151.7     | 208.3     |
> |                      | DoF(consistency model)   | 55.3     | 70.2     | 116.3     | 148.9     | 202.1     |
> |                      | MACQL                     | 50.7     | 65.6     | 100.3     | 135.1     | 190.6     |
> |                      | MABCQ                    | 42.1     | 49.4     | 90.4      | 119.2     | 162.3     |
> | | MADIFF| 63.8| 70.4|113.5|148.3|Out Of Momory|
>
> These results demonstrate that, while diffusion-based methods generally have higher computational costs, techniques like DDIM and Consistency Model significantly reduce inference times of DoF with DDPM, enhancing their practical scalability. With fast sampling techniques such as DDIM and the consistency model, the rewards of DoF slightly decrease, but its inference time significantly decreases. For example, when DoF adopts the consistency model technique, it always performs better than the other non-diffusion methods (MACQL and MABCQ), at the cost of moderately higher inference time (2.4s vs 1.6s). The above results are included in Appendix E.4 .
>
>
> Our choice of diffusion models for multi-agent decision making is primarily motivated by their powerful modeling ability, the avoidance of estimating value function, and the ability to generate novel behaviors. We have shown the powerful modeling ability of DoF in Section 5.1. We have shown that through modeling the trajectory rather than modeling the value function, DoF-Trajectory circumstances the need for value function estimation, which is error-prone. And DoF-Trajectory can obtain better performance than others (Table 2, 3, 4). Through a flexible combination of conditions, we can generate novel behaviors at test time, such as combining multiple skills or constraints. We validate this property through specific case studies in Appendix D.6. In Appendix D.6, two datasets, each containing a semi-circle, are used to train DoF-trajectory, and through the combination of multiple conditions, DoF-trajectory can generate a full circle.
>
> **Reference**
>
> [1] Aviral Kumar, Aurick Zhou, George Tucker, and Sergey Levine. Conservative q-learning for offline reinforcement learning. In NeurIPS, 2020.
>
> [2] Jiechuan Jiang, Zongqing Lu. Offline decentralized multi-agent reinforcement learning. In ECAI, 2023.
>
> [3] Jiaming Song, Chenlin Meng, Stefano Ermon. Denoising Diffusion Implicit Models. In ICLR, 2021.
>
> [4] Yang Song, Prafulla Dhariwal, Mark Chen, Ilya Sutskever. Consistency Models. In ICML, 2023

---

> ### Author Response · Authors · 2024-11-18
>
> > Given that QMIX is an online algorithm not designed for offline reinforcement learning, could you clarify the rationale for including it in the comparisons? What insights does this comparison provide?
>
> Including QMIX in our comparisons follows the approach of prior offline MARL approaches such as MADiff [1] and OGMARL [2]. Although QMIX is primarily an online algorithm, its inclusion allows us to evaluate the effectiveness of famous online value-decomposition methods, in offline reinforcement learning scenarios. **Following the advice of the reviewer, we have removed the results of QMIX from Table 3.**
>
>
> > Can diffusion-based methods effectively handle out-of-distribution situations?
>
> Thank you for the insightful question. We believe diffusion methods exhibit certain advantages in handling out-of-distribution (OOD) situations. When using DoF-Trajectory, from the perspective of trajectory modeling, diffusion methods bypass the traditional Q-value estimation step commonly used in reinforcement learning [3]. This characteristic naturally alleviates the issue of Q-value overestimation (e.g., extrapolation and bootstrapping) in offline scenarios, especially when learning from limited or biased data, thereby mitigating OOD problems. When using DoF-Policy, from the perspective of policy modeling, diffusion models model the distribution of the underlying offline policy well. In this way, it effectively reduces extrapolation errors caused by sampling out-of-distribution actions [4]. We have added this paragraph to the discussion section  in Appendix E.3.1 of our submission.
>
> > Additionally, can these methods be applied to competitive or mixed-motivation scenarios?
>
> Thank you for your question. As it is written in the abstract, in the introduction, and the background of the initial submission, We consider cooperative multi-agent decision-making tasks.  We have not delved into competitive or mixed-motivation settings. However, we believe this is a highly valuable direction for future research. To address this, in future studies, we could extend our framework by explicitly incorporating motivations into the diffusion conditions. For instance, in competitive scenarios, the diffusion model could condition adversarial strategies or payoff structures during sampling. In mixed-motivation scenarios, it could integrate both individual motivations and shared objectives into the diffusion process. This enhancement would enable the model to better handle the complexity of competitive and mixed-motivation environments, paving the way for broader applications of our approach. We have added this paragraph to the discussion section in Appendix E.3.2 of our submission.
>
> **Reference**
>
> [1] Zhu, et al. "MADiff: Offline Multi-agent Learning with Diffusion Models." NeurIPS 2024.
>
> [2] Formanek, et al. "Off-the-Grid MARL: Datasets and Baselines for Offline Multi-Agent Reinforcement Learning." AAMAS 2023.
>
> [3] Zhendong Wang, Jonathan J. Hunt, and Mingyuan Zhou. Diffusion policies as an expressive policy class for offline reinforcement learning. In ICLR 2023.
>
> [4] Anurag Ajay, Yilun Du, Abhi Gupta, Joshua B. Tenenbaum, Tommi S. Jaakkola, and Pulkit Agrawal. Is conditional generative modeling all you need for decision making? In ICLR, 2023

---

> ### Author Response · Authors · 2024-11-26
>
> Dear Reviewer y4iS,
>
> We have responded to your reviews 9 days ago. Can you please read our response? Are there still any remaining concerns?
>
> Best regards,
>
> Authors

---

> > ### Author Response · Authors · 2024-11-27
> >
> > Dear Reviewer y4iS,
> >
> > We submitted our response to your reviews on November 18th, and it has now been 10 days. We kindly ask if you could review our responses at your earliest convenience.
> >
> > Best regards,
> >
> > Authors

---

> ### Author Response · Authors · 2024-11-29
>
> Dear Review y4iS,
>
> Thank you for your thoughtful feedback. We appreciate your recognition of the novelty of the IGD principle, the usefulness of noise and data factorization, and the extensive experiments.
>
> In response, we have conducted scalability experiments, clarified scenarios for DoF-Policy and DoF-Trajectory, and revised the experimental section for clarity based on your suggestions.
>
> We have received positive feedback from another reviewer, noting that our responses addressed the reviewer's concern and led to an improved rating. We hope our responses address your concerns and would appreciate it if you could reconsider your rating.
>
> Best Regards,
>
> Authors

---

> > ### Author Response · Authors · 2024-12-02
> >
> > Dear Review y4iS,
> >
> > We wanted to follow up on our previous message. It has been over **14  days** since we addressed your valuable feedback, and with only **one day** remaining in the rebuttal period, we are eager to engage with you further.
> >
> > We have carefully considered your comments, particularly regarding the clarity of our paper and the inclusion of scalability metrics. We have made revisions to enhance these aspects and have addressed the concerns you raised.
> >
> > Additionally, we have successfully addressed the issues pointed out by two other expert reviewers, which led them to improve their ratings of our work. Moreover, another reviewer has discussed with us in a back-and-forth fashion, which is the key feature of the ICLR conference. We are hopeful that our responses address your concerns.
> >
> > If there are any other questions or areas you’d like us to clarify, please let us know. We are keen to discuss them with you to ensure our work meets your expectations.
> >
> > Thank you for your time and consideration.
> >
> > Best regards,
> >
> > Authors

---

### Official Review · Reviewer_zeea · 2024-11-05

**Soundness:** 1
**Presentation:** 1
**Contribution:** 2
**Rating:** 3
**Confidence:** 2

**Summary:**

This paper proposes an individual-global-identically distributed (IGD) principle (property) for model design in cooperative multi-agent decision-making.  The paper claims that the property generalizes the popular individual-global-max principle (IGM).  They further propose a diffusion factorization framework that 1) satisfies the IGD principle and 2) performs well on several datasets.

**Strengths:**

The empirical results show their method perform slightly better than other methods.

**Weaknesses:**

I am unfamiliar with this domain of literature, and my comment would mostly focus on mathematical writing.  Overall, I have strong concerns about the claim that IGD generalizes IGM, and I feel the theory about diffusion factorization framework under Gaussian assumption satisfying IGD uninteresting.  Thus, I suggest rejection.

The technical writing quality needs significant improvement. Here are some comments.
1. Does IGD property apply to the initial distribution $p_{\theta_{tot}}(x_{tot}^0)$  or the whole series $p_{\theta_{tot}}(x_{tot}^{0:K})$? Does $x_{tot}^K$ need to be sampled from Gaussian?  Does the reverse process need to be a Markov chain?
2.  The claim that IGD generalizes IGM is not well-defined.  IGM needs a value function but does not necessarily have a distribution over action or even under a diffusion model.  Please give an explicit statement on the generalization: For instance, given a value function $Q_{jt}$ satisfies IGM what is the corresponding $p_{\theta_{tot}}$ such that Eqn.(1) holds for any history $\tau_{tot}$.
3.  At the end of section 4.2, there should be formal statements of IGD vs individual global optimal principle or risk-sensitive IGM principles.

If I understand correctly, Theorem 1 basically states that each component of isotropic Gaussian is independent.  I feel this result is very limited and hardly holds in general.  For instance, it’s unclear whether this would still apply even when the Gaussian has a non-diagonal covariance.



Minor comments
- In eq (3), instead of $\theta_i\subset \theta_{tot}$, $(\theta_1,\dots, \theta_K) = \theta_{tot}$ would be better, as the order of coordinate matters.
- $\epsilon^{\theta_1}_1$ in eq (5) is not clearly defined.

**Questions:**

Can the authors provide formal statements and proofs showing that IGD generalizes IGM, individual global optimal principle or risk-sensitive IGM principles?

Are there any nontrivial examples that satisfy IGD beyond the Gaussian case (or a component-wise independent Markov chain with a stationary initial distribution)?

---
After discussing with the author, I remain deeply concerned about the paper’s mathematical writing, which requires substantial improvement. First, the claim that IGD generalizes IGM is neither formally stated nor proved, even after multiple responses. This is not merely an issue of correctness; the statement is fundamentally unfalsifiable and lacks rigor. Second, the definition of IGD is unnecessarily convoluted and appears redundant. A more straightforward would be that a diffusion model exhibits IGD if each component of the initial state is mutually independent. While the empirical results may be interesting, the mathematical statement is seriously deficient and demands thorough and precise revision to meet an acceptable standard.

---

> ### Author Response · Authors · 2024-11-18
>
> **Thanks for your honesty about the unfamiliarity with the diffusion and the diffusion-based RL domain.**  Thanks for your time and effort to read our paper. It is a pity that you cannot read our work thoroughly due to the lack of domain-specific expertise.  We are eager to discuss with you to obtain a quality review and fair judgment. We will try our best to make you familiar with the diffusion methods, diffusion-based RL, and diffusion-based MARL domain to gain a better understanding of our work. We hope that after reading our rebuttal and related literature, you can gain the expertise to give a quality and in-depth review.
>
>
> In this rebuttal, we will give the necessary background information, address the questions, and show that the IGD is an extension of the IGM and RIGM principles.
>
>
> > I am unfamiliar with this domain of literature, and my comment would mostly focus on mathematical writing. Overall, I have strong concerns about the claim that IGD generalizes IGM, and I feel the theory about diffusion factorization framework under Gaussian assumption satisfying IGD uninteresting. Thus, I suggest rejection.
>
> An in-depth understanding of our work requires the literature such as diffusion models, diffusion-based RL, and diffusion-based MARL. Although we have discussed background knowledge in the submission (and in the appendix), here we present more details of this background knowledge and point out related literature.
>
>
> **It seems that one of the major concerns of the reviewer is about Gaussian distribution. We use the Gaussian Distribution the same as the seminar DDPM work [1]. The Gaussian distribution is used both in the forward process and the backward process. It is used to model the noise used in diffusion and it is used for modeling the transition probability of adding noise into data and for modeling the backward transition probability of removing noise from the data. Using the Gaussion distribution to model the noise and the transitions are not invented by us, we take such an approach from DDPM**
>
> **1. Background Information about Diffusion Models**
>
> The seminal work DDPM [1] uses diffusion to generate images. It consists of the forward process and the backward (reverse) process. The backward processes of DDPM and ours are Markov chains. The forward step and the backward step of the diffusion process are both defined as Gaussian Distribution. Reference [2] is a very nice tutorial about diffusion models; it shows the math after diffusion models in excruciating detail. Please have a look if you have any doubts.
>
> In the forward diffusion process, data $x^k$ is polluted by adding noise to $x^{k-1}$ according following formula (Gaussian Distribution).
>
> $p(x^k|x^{k-1})  = \mathcal{N}(\sqrt{\alpha^k} x^{k-1}, \sqrt{1 - \alpha^k}\boldsymbol{\epsilon^{k-1}})$
>
> where $\epsilon^{k-1} \sim \mathcal{N}(0, \textbf{I})$ is a Gaussian Noise, $k$ is the diffusion time step, $\alpha^k$ is a pre-specified hyper-parameter. $\mathcal{N}(\sqrt{\alpha^k} x^{k-1}, \sqrt{1 - \alpha^k}\boldsymbol{\epsilon^{k-1}})$ is a Gaussian Distribution with mean $\sqrt{\alpha^k} x^{k-1}$, and variance $\sqrt{1 - \alpha^k}\boldsymbol{\epsilon^{k-1}}$. In essense, $p(x^k|x^{k-1})$ is a probability, that the probability of sampling $x^k$ condition on $x^{k-1}$. Thanks to the property of Gaussian distribution, it can be written as $x^k  = \sqrt{\alpha^k} x^{k-1} + \sqrt{1 - \alpha^k}\boldsymbol{\epsilon}^{k-1}$.
>
>
> The backward diffusion process （denoising process）is a Markov chain. It is used to sample data based on Gaussian Distribution. The probability $p_{\boldsymbol{\theta}}(\boldsymbol{x}^{k-1}|\boldsymbol{x}^k)$ of generating $\boldsymbol{x}^{k-1}$ based on $\boldsymbol{x}^{k}$ is defined as a Gaussian distribution, which is defined in the following formula.
>
> $p _ {\boldsymbol{\theta _ i}}(\boldsymbol{x}^{k-1}|\boldsymbol{x}^k) = \mathcal{N}(\boldsymbol{x}^{k-1}; \boldsymbol{\mu} _ {\boldsymbol{\theta}}(\boldsymbol{x}^k, k), \boldsymbol{\Sigma}(k))$, where $\boldsymbol{\mu} _ {\boldsymbol{\theta}}(\boldsymbol{x}^k, k)$ and $\boldsymbol{\Sigma}(k)$ are the mean and variance of the Gaussian distribution. They are trained to maximize the true data distribution.
>
> **We would like to point out that defining the transition probability of the Markov Chain (backward process) as Gaussion Distribution and modeling the noise as Gaussian Distribution is not invented or proposed by us. We just follow the approach of typical diffusion models such as DDPM.**

---

> ### Author Response · Authors · 2024-11-18
>
> **2. Background Information about Diffusion-based RL**
>
> Besides using diffusion models to generate images or videos, researchers have applied diffusion models to generate decisions for offline RL. There are two representative ways for diffusion-RL: trajectory-based and policy-based methods.
>
>
>
> Trajectory-based methods use diffusion to generate trajectories and plan based on the generated trajectory. The Diffuser [3] and Decision Diffuser [4] are the seminar work utilizing the diffuion model to generate trajectories. After the trajectories are generated, an inverse dynamic model is used to plan action based on the trajectories. This allows for more accurate long-horizon decision-making, especially in offline settings with limited data. They circumstance the need to estimate value functions, which are difficult to approximate in an offline-RL setting.
>
> Policy-based methods use diffusion to generate policies (actions). DiffusionQL [5] is the first diffusion-based RL method that generates actions through diffusion. It makes use of the powerful modeling ability of diffusion to model policy/action distribution.
>
>
>
>
> **3. Background Information about Diffusion-based MADM (MARL)**
>
> Due to the promising performance and flexible modeling ability of diffusion-based RL, researchers have applied diffusion-based methods to offline multi-agent decision-making (MADM). These approaches can be categories into trajectory-based and policy-based methods. As far as we know, as discussed in the introduction, the related work, the experiments, and the appendix, the trajectory-based diffusion MADM method [6] suffers from scalability issues, and the policy-based diffusion MADM method [7] suffers from poor cooperation issues.
>
>
> For the trajectory-based method, the first approach that uses a diffusion-based method for MADM is MADIFF [6]. It is built based on a decision diffuser, with the introduction of an attention-based mechanism to model the relationship among agents. Thus, it achieves promising performance. As we had discussed in the introduction, the related work, the experiments (Table 5), and the appendix (A.2), suffers from scalability issues. Our work enjoys better scalability than it (Table 5). Its performance can be further improved (Table 2, 3, 4). Moreover, we show that our work can be used to generate data beyond trajectories. Our work can be used to generate actions (DoF-policy), generate Q values (Table 1), and generate novel behaviors through flexible conditioning (Appendix, page 34, Figure 10).
>
> The policy-based diffusion MADM methods, such as [7], ignore the relationship among agents. It uses either DiffusionQL to model each agent without considering their cooperation. We call these diffusion-based approaches, which do not consider cooperation, as independent diffusion. They lead to poor performances, as it is shown in Figure 1, and Figure 3.
>
>
> In this work, we show that DoF can enjoy better scalability with better performance than existing diffusion-based MADM methods. Besides trajectories and actions, it can generate Q values. Moreover, it can generate novel behaviors through flexible conditioning.
>
>
>
> **Reference**
>
> [1] Ho, Jonathan, Ajay Jain, and Pieter Abbeel. "Denoising diffusion probabilistic models." Advances in neural information processing systems 33 (2020): 6840-6851.
>
> [2] Calvin Luo. Understanding diffusion models: A unified perspective. CoRR, abs/2208.11970, 2022
>
> [3] Michael Janner, Yilun Du, Joshua B. Tenenbaum, and Sergey Levine. Planning with diffusion for flexible behavior synthesis. In ICML, volume 162, pp. 9902–9915, 2022
>
> [4] Anurag Ajay, Yilun Du, Abhi Gupta, Joshua B. Tenenbaum, Tommi S. Jaakkola, and Pulkit Agrawal. Is conditional generative modeling all you need for decision making? In ICLR, 2023
>
> [5] Zhendong Wang, Jonathan J. Hunt, and Mingyuan Zhou. Diffusion policies as an expressive policy class for offline reinforcement learning. In ICLR 2023.
>
> [6] Zhengbang Zhu, Minghuan Liu, Liyuan Mao, Bingyi Kang, Minkai Xu, Yong Yu, Stefano Ermon, and Weinan Zhang. MADIFF: Offline multi-agent learning with diffusion models. NeurIPS 2024
>
> [7] Zhuoran Li, Ling Pan, and Longbo Huang. Beyond conservatism: Diffusion policies in offline multi-agent reinforcement learning. arXiv preprint arXiv:2307.01472, 2023.

---

> ### Author Response · Authors · 2024-11-18
>
> >The empirical results show their method perform slightly better than other methods.
>
>
> The empirical results show that our method advance the state-of-the-art methods not just in terms of rewards, but also in terms of data generation ability, in terms of scalability.
>
> **Data Generation Ability**
>
> DoF demonstrates strong modeling capabilities by generating data-consistent distributions, as shown in Section 5.1. This capability is beyond the reach of traditional offline multi-agent reinforcement decision making approaches. Moreover, it generates data much better than other diffusion-based MADM methods.
>
> DoF can be used to generate novel behaviors by flexible conditions. We conducted experiments in Appendix D.6. DoF successfully generated a complete circle, whereas its training data consisted of only half circles. Through using flexble conditions, DoF agents can collaborate to meet complex constraints and achieve novel behaviors.
>
> **Scalability for Diffusion-based MADM**
>
> As discussed in Appendix A.2, the input complexity of DoF is $O(1)$ with respect to the number of agents, whereas the input complexity of MADIFF [NeurIPS 24] is $O(N)$, where $N$ is the number of agents. We have demonstrated that the inference time of MADIFF can be 1.7 times than DoF in Table 5.
>
> **Rewards**
>
> SMAC and SMACv2 Experiments (Section 5.2.1, Tables 2 and 3): Our method, DoF, has achieved a 16% improvement over another diffusion-based method, MADIFF. DoF achieves a reward improvement of 0.8 over the second-best algorithm, which does not have flexible generation ability. MPE Experiments (Table 4): DoF outperforms the second-best algorithm by an average reward improvement of 6 and shows a 19% improvement compared to MADIFF.
>
> These results collectively demonstrate that the improvement of DoF over existing approaches is substantial. Reviewer y4iS comments DoF outperforms competing methods and performs well with an increasing number of agents. Reviewer YHzy regards the performance of DoF as strong. We would like to discuss this point with you.
>
>
> > Does IGD property apply to the initial distribution $p _ {\theta _ {tot}}(x^{0} _ {tot})$ or the whole series $p _ {\theta _ {tot}}(x^{0:K} _ {tot})$? Does $x^K _ {tot}$ need to be sampled from Gaussian? Does the reverse process need to be a Markov chain?
>
> 1. The IGD property applies to **neither** $p _ {\theta _ {tot}}(x^{0} _ {tot})$ **nor** the whole series $p _ {\theta _ {tot}}(x^{0:K} _ {tot})$, it apply to both the generated data $[p _ {\theta _ {i}}(x^{0} _ {i})] _ {i=1}^N$ and $p _ {\theta _ {tot}}(x^{0} _ {tot})$ as it is written in Line 200 and 201 of the initial submission.
>
> 2. $x^K_{tot}$ must be sampled from Gaussian Noise. We had already written in Line 194  of the initial submission *"Gaussian distribution starting as $p(x_{tot}^K) = \mathcal{N}(0, \textbf{I}) \in \mathcal{R}^{N \times d}$"*
>
> 3. The reverse process is a Markov chain. We had already written in Line 193 of the initial submission *"the reverse process, defined as a Markov chain ..."*.

---

> > ### Comment · Reviewer_zeea · 2024-11-22
> >
> > Thank you for this response.  I still feel definition 2 is misleading.  If I understand it correctly, it says a joint distribution $p_{\theta_{tot}}(x^0_{tot})$ can be decomposed into $[p_{\theta_i}(x^0_{i})]^N_{i = 1}$.  Note that without loss of generality, the IGD property only applies $p_{\theta_{tot}}(x^0_{tot})$ as $[p_{\theta_i}(x^0_{i})]^N_{i = 1}$ can be derived from $p_{\theta_{tot}}(x^0_{tot})$.
> >
> >
> > If that is the case,
> > - Is IGD just saying that each coordinate of $x^0_{tot} = (x^0_1, ... , x^0_N)$ is mutually independent?
> > - What is the point of introducing the whole series $x_{tot}^{0:K}$?  Those Markovian and Gaussian assumptions seem to be related to diffusion model, while IGD is still defined under general random variable $x^0_{tot} = (x^0_1,..., x^0_N)$.
> > - Finally, if you really want to focus on diffusion models, can you just say a diffusion model has IGD if each coordinate of initial state is mutually independent.

---

> ### Author Response · Authors · 2024-11-18
> **The IGD principle is a generalization of the IGM principle**
>
> > The claim that IGD generalizes IGM is not well-defined. IGM needs a value function but does not necessarily have a distribution over action or even under a diffusion model. Please give an explicit statement on the generalization: For instance, given a value function $Q_{jt}$ satisfies IGM what is the corresponding $p_{\theta_{tot}}$ such that Eqn.(1) holds for any history $\tau_{tot}$.
>
> > At the end of section 4.2, there should be formal statements of IGD vs individual global optimal principle or risk-sensitive IGM principles.
>
>
>
> We had shown the ability of DoF to generate value functions in Table 1 of the initial submission. DoF can be used to recover the optimal policy.
>
> To show that the principle is a generalization of the IGM principle, **we first show that the IGD principle is a generalization of the IGO principle.** We have added these proofs into the Appendix E.1 .
>
> **The definition of the IGO Pinrciple [1]**: For an optimal joint policy $\pi_{tot}^*(\mathbf{u_{tot}} \mid \mathbf{\tau_{tot}})$ : $\boldsymbol{\mathcal { T }} \times \boldsymbol{\mathcal { U }} \rightarrow[0,1]$, where $\mathbf{\tau_{tot}} \in \boldsymbol{\mathcal { T }}$ is a joint trajectory,
> if there exist individual optimal policies $\left[\pi_i^*\left(u_i \mid \tau_i\right): \mathcal{T} \times \mathcal{U} \rightarrow [0,1]\right]_{i=1}^N$, such that the following holds:
>
> $$
> \pi_{tot}^*(\mathbf{u_{tot}} \mid \mathbf{\tau_{tot}})=\prod_{i=1}^N \pi_i^*\left(u_i \mid \tau_i\right),
> $$
>
> then, we say that $\left[\pi _ i\right] _ {i=1}^N$ satisfy $\mathbf{I G O}$ for $\pi _ {tot}$ under $\boldsymbol{\tau _ {tot}}$.
>
> The IGD principle requires that
>
> $$
> \prod_{i=1}^N p_{\theta_i}(x_i^0) = p_{\theta_{tot}}(x_{tot}^{0})
> $$
>
> where $x_i^0$ is the generated data of agent $i$, and $x_{tot}^0$ is the generated data of the whole multi-agent sytstem, and $p_{\theta_i}(x_i^0)$ is the probaility of $x_i^0$, and $p_{\theta_{tot}}(x_{tot}^{0})$ is the probability of $x_{tot}^0$.
>
> We can use the IGD principle to generate multi-agent action. Let's treat the generated data $x_i^0$ as $u_i$, where $u_i$ is the action taken by agent $i$. Then, the probability $p_{\theta_i}(u_i)$ becomes $\pi_i(u_i\mid \tau_i)$, where $\pi_i$ is the policy of agent $i$ and $\tau_i$ is the local observation. Further, let's treat the generated total data $\mathbf{x_{tot}^0}$ as $\mathbf{u_{tot}}$. The probability $p_{\mathbf{\theta_{tot}}}(\mathbf{u_{tot}})$ becomes $\pi_{tot}(\mathbf{u_{tot}}\mid \mathbf{\tau_{tot}})$, where $\pi_{tot}$ is the policy of the multi-agent system, and $\tau_{tot}$ is the aggregated observations. Then the IGD principle becomes the following formulas.
>
> $$\prod_{i=1}^N p_{\theta_i}(u_i) = p_{\mathbf{\theta_{tot}}}(\mathbf{u_{tot}})
> $$
>
> $$
> \prod_{i=1}^N \pi_i(u_i\mid \tau_i) = \pi_{tot}(\mathbf{u_{tot}}\mid \mathbf{\tau_{tot}})
> $$
>
> The above formula requires that for any policy $\pi$, the total policy $\pi_{tot}$ is equal to the product of its per-agent policy. By substituting the optimal policy $\pi_i^*$ for the policy $\pi_i$ and the optimal total policy $\pi_{tot}^*$ as the policy $\pi_{tot}$, we can obtain the following formula.
>
> $$\prod_{i=1}^N \pi_i^*(u_i\mid \tau_i) = \pi_{tot}^*(\mathbf{u_{tot}}\mid \mathbf{\tau_{tot}})
> $$
>
> $$\pi_{tot}^*(\mathbf{u_{tot}}\mid \mathbf{\tau_{tot}}) = \prod_{i=1}^N \pi_i^*(u_i\mid \tau_i)
> $$
>
> The above formula is exactly the requirement of the IGO principle. Thus, we have shown that the IGD principle is a generalization of the IGO principle.
>
> **The IGD principle is a generalization of the IGM principle**
>
> We have shown that the IGD principle is a generalization of the IGO principle. And the IGO paper [1] shows that the IGO principle is a generalizatioin of the IGM principle, thus the IGD principle is a generalization of the IGM principle.

---

> > ### Comment · Reviewer_zeea · 2024-11-22
> >
> > Again, I am asking for formal statement showing that IGD generalizes IGM.  Table 1 seems to be special cases.

---

> ### Author Response · Authors · 2024-11-18
>
> **The IGD principle is a generalization of the RIGM principle**
>
> We first describe the RIGM principle [2], and we introduce a generation of the RIGM principle (GRIGM), in the end we show that the IGD principle is the generation of the GRIGM principle. Thus, the IGD principle is a generalization of the RIGM principle.
>
> **The definition of the RIGM principle**: Given a risk metric $\psi_\alpha$, a set of individual return distribution utilities $\left[Z _ i\left(\tau _ i, u _ i\right)\right] _ {i=1}^N$, and a joint state-action return distribution $Z _ {tot}(\boldsymbol{\tau _ {tot}}, \boldsymbol{u _ {tot}})$, if the following conditions are satisfied:
>
> $$
> \arg \max _ {\mathbf{u}} \psi _ \alpha\left(Z _ {tot}(\boldsymbol{\tau _ {tot}}, \mathbf{u _ {tot}})\right) =
> \left[ \arg \max _ {u _ 1} \psi _ \alpha\left(Z _ 1\left(\tau _1, u _1\right)\right), \ldots,
> \arg \max _ {u _ N} \psi _ \alpha\left(Z _ N\left(\tau _N, u _ N\right)\right)
> \right]
> $$
>
> where $\psi _ \alpha: Z \times R \rightarrow R$ is a risk metric such as the VaR or a distorted risk measure, $\alpha$ is its risk level. Then, $\left[Z _ i\left(\tau _ i, u _ i\right)\right] _ {i=1}^N$ satisfy the RIGM principle with risk metric $\psi _ \alpha$ for $Z _ {j t}$ under under $\tau$. We can state that $Z _ {tot}(\boldsymbol{\tau _ {tot}}, \boldsymbol{u _ {tot}})$ can be distributionally factorized by $\left[Z _ i\left(\tau _ i, u _ i\right)\right] _ {i=1}^N$ with risk metric $\psi _ \alpha$.
>
>
> **Generalization of the RIGM principle**
>
> Let's define probablities function $\pi _ {tot}(\mathbf{u _ {tot}} \mid \mathbf{\tau _ {tot}})$ and $\pi _ i(u _ i \mid \tau_i)$. $\pi_{tot}(\mathbf{u_{tot}} \mid \mathbf{\tau_{tot}}) = 1$, when $u_{tot} = \arg \max _{\mathbf{u}} \psi _ \alpha\left(Z _ {tot}(\mathbf{\tau _ {tot}}, \mathbf{u _ {tot}})\right)$, and it is $0$, otherwise. For $\pi _ i(u _ i \mid \tau _ i)$, $\pi _ i(u _ i \mid \tau _ i)=1$, when $u _ i = \arg \max _ {u _1} \psi _ \alpha\left(Z_i\left(\tau_i, u_i\right)\right)$, otherwise 0. The RIGM principle becomes the following formula.
>
> $$
> \pi_{tot}(\mathbf{u_{tot}}\mid \mathbf{\tau_{tot}})=\prod_{i=1}^N \pi_i(u_i\mid \tau_i)
> $$
>
>
> The same as the IGO case, we can use a diffusion model to generate risk-sensitive multi-agent action. Let's treat the generated data $x_i^0$ as risk-sensitive action $u_i$ of agent $i$. Then, the probability $p_{\theta_i}(x_i^0) = p_{\theta_i}(u_i)$ becomes $\pi_i(u_i\mid \tau_i)$, where $\pi_i$ is the risk-sensitive policy of agent $i$ and $\tau_i$ is the local observation. Further, let's treat the generated total data $\mathbf{x_{tot}^0}$ as $\mathbf{u_{tot}}$. The probability $p_{\theta_{tot}}(\mathbf{x_{tot}^0}) = p_{\mathbf{\theta_{tot}}}(\mathbf{u_{tot}})$ becomes $\pi_{tot}(\mathbf{u_{tot}}\mid \mathbf{\tau_{tot}})$, where $\pi_{tot}$ is the risk-sensitive policy of the multi-agent system, and $\tau_{tot}$ is the aggregated observations.
>
> $$\prod_{i=1}^N p_{\theta_i}(x_i^0) = p_{\theta_{tot}}(x_{tot}^{0})$$$$\prod_{i=1}^N p_{\theta_i}(u_i) = p_{\mathbf{\theta_{tot}}}(\mathbf{u_{tot}})$$$$\prod_{i=1}^N \pi_i(u_i\mid \tau_i) = \pi_{tot}(\mathbf{u_{tot}}\mid \mathbf{\tau_{tot}})$$$$\pi_{tot}(\mathbf{u_{tot}}\mid \mathbf{\tau_{tot}}) = \prod_{i=1}^N \pi_i(u_i\mid \tau_i)$$
>
> We have shown that the IGD principle is a generalization of the general RIGM principle. Thus, the IGD principle is a generalization of the RIGM principle.
>
>
> **Reference**
>
> [1] Tianhao Zhang, Yueheng Li, Chen Wang, Guangming Xie, and Zongqing Lu. FOP: factorizing optimal joint policy of maximum-entropy multi-agent reinforcement learning. In ICML, 2021
>
> [2] Siqi Shen, Chennan Ma, Chao Li, Weiquan Liu, Yongquan Fu, Songzhu Mei, Xinwang Liu, and Cheng Wang. Riskq: Risk-sensitive multi-agent reinforcement learning value factorization. In NeurIPS, 2023.

---

> ### Author Response · Authors · 2024-11-18
>
> > If I understand correctly, Theorem 1 basically states that each component of isotropic Gaussian is independent. I feel this result is very limited and hardly holds in general. For instance, it’s unclear whether this would still apply even when the Gaussian has a non-diagonal covariance.
>
> The understanding is not correct. We do not assume the data follow a Gaussian distribution. Theorem 1 does not state that each component of the isotropic Gaussian noise is independent. Instead, it establishes that a multi-agent diffusion model $p _ {\boldsymbol{\theta} _ {tot}}$ represents the data ($x _ {tot}^0$, **rather than noise**) following distribution (e.g., trajectories or actions), and can be factorized into individual data $x _ i$. Theorem 1 can be viewed as after training, it requires that each component $x _ i^0$ of a generated **data** $x _ {tot}^0$ is independent. The theorem focuses on modeling the global data distribution $p _ {\boldsymbol{\theta} _ {tot}}$ as a combination of locally optimal models $p _ {\boldsymbol{\theta} _ i}$, rather than making assumptions about the independence of Gaussian noise.
>
> For diagonal covariance Gaussian noise, the DDPM and the majority of the diffusion model use diagonal covariance Gaussian distribution for both the noise and the transition probability. Our work is built based on these Diffusion models. The use of the Gaussian distribution with diagonal covariance is inherited from the DDPM framework [1]. Researchers have demonstrated that through using the diagonal Gaussian distribution, the diffusion model can generate very realistic images, and we have shown that through using the default Gaussian noise, DoF can generate very good decision.
>
> If a diffusion model uses other forms of distributions instead of Gaussian distributions, it is unclear whether Theorem 1 still holds. We have added the following sentence in Theorem 1  "The noise $\epsilon_{tot}$ and $\epsilon_i$ and the transition probability ($p _ {\theta _ {tot}}(\mathbf{x^{k-1} _ {tot}}|\mathbf{x^k _ {tot}})$ and $p _ {\theta _ {i}}(x^{k-1} _ {i}|x^k_i)$) follow diagonal Gaussian distributions".
>
>
> > Minor comments:
> > - In eq (3), instead of $θ_i ⊂ θ_{tot}, (θ_1,...,θ_K) = θ_{tot}$ would be better, as the order of coordinate matters.
> > - $\epsilon_{1}^{\theta_1}$ in eq (5) is not clearly defined.
>
> We do not agree with the reviewer. Although the Concat and WConcat functions implement $(θ_1,...,θ_K) = θ_{tot}$, in the definition of IGD, we expect there could be other functions satisfy $\theta_i \subset \theta_{tot}$.
>
> $\epsilon_{1}^{\theta_1}$ in eq (5) was defined in lines 250-255 (around Equation 5) in the initial submission. We copy the text for the reviewer's reference *"multiple small noise models $\epsilon_{i}^{\theta_i}(x_i^{k}, k)$, each parameterized by $\theta_i$. $\theta_i \subset \theta_{tot} \quad \forall i,\;\theta_i \cap \theta_j = \emptyset\;\; i\neq j$. During execution, agent $i$ uses the noise model $\epsilon_i^{\theta_i}$ to generate data"*. We will make it more clear.
>
> > Can the authors provide formal statements and proofs showing that IGD generalizes IGM, individual global optimal principle or risk-sensitive IGM principles?
>
> Please refer to the above proofs.
>
> > Are there any nontrivial examples that satisfy IGD beyond the Gaussian case ?
>
> We would like to clarify that our method, DoF, is built on the diffusion modeling approach introduced in the DDPM[1] framework, which uses Gaussian noise. We do not assume that the data follows the Gaussian distribution. Similarly, well-known works in reinforcement learning, like Diffuser[2], Diffusion-QL[3], and Decision Diffuser[4], also use Gaussian noise. Other types of noise distribution is possible, but it is out of the scope of our work.

---

> > ### Comment · Reviewer_zeea · 2024-11-22
> >
> > > If I understand correctly, Theorem 1 basically states that each component of isotropic Gaussian is independent. I feel this result is very limited and hardly holds in general. For instance, it’s unclear whether this would still apply even when the Gaussian has a non-diagonal covariance.
> >
> > I should clarify my point. In my view, the proof technique relies heavily on the assumption that the process is Gaussian. Since I am not particularly interested in diffusion processes, I find the approach lacks broader insight.

---

> ### Author Response · Authors · 2024-11-18
>
> >or a component-wise independent Markov chain with a stationary initial distribution
>
>
> Regarding independent assumption, the IGM principle has a hidden independent assumption: It assumes that each agent can independently make decisions without communicating with others.  Below we give a detailed description:
>
> The IGM principle requires that $(\bar{u_1}, ..., \bar{u_N}) = \bar{u_{tot}}$, where $\bar{u_i} = argmax Q_u(\tau_i, u)$, $\bar{u_{tot}}= argmaxQ_{tot}(\tau_{tot}, u)$. There is a hidden independent assumption of the IGM principle. It implicitly assumes that each agent can make the optimal decision independently based on individual utility function $Q_i(\tau_i, u_i)$, where $\tau_i$ is the local observation-action history for agent $i$. However, the individual utility function does not guarantee to optimal decision. To obtain the true optimal action, the utility function should be $Q_i(\tau_{tot}, u_i)$, where $\tau_{tot}$ is the observation-action history for all the agents. Such an independent assumption for the IGM principle may not hold for multi-agent tasks that require communication among agents. The IGD principle has the independent assumption (limitation) as the IGM principle.
>
> **Reference**
>
>  [1] Ho, Jonathan, Ajay Jain, and Pieter Abbeel. "Denoising diffusion probabilistic models." Advances in neural information processing systems 33 (2020): 6840-6851.
>
> [2] Michael Janner, Yilun Du, Joshua B. Tenenbaum, and Sergey Levine. Planning with diffusion for flexible behavior synthesis. In ICML, volume 162, pp. 9902–9915, 2022
>
> [3] Zhendong Wang, Jonathan J. Hunt, and Mingyuan Zhou. Diffusion policies as an expressive policy class for offline reinforcement learning. In ICLR 2023.
>
> [4] Anurag Ajay, Yilun Du, Abhi Gupta, Joshua B. Tenenbaum, Tommi S. Jaakkola, and Pulkit Agrawal. Is conditional generative modeling all you need for decision making? In ICLR, 2023

---

> ### Author Response · Authors · 2024-11-24
>
> > If I understand it correctly, it says a joint distribution $p_{\theta_{tot}}(x^0_{tot})$ can be decomposed into $[p_{\theta_i}(x^0_{i})]^N_{i = 1}$. Note that without loss of generality, the IGD property only applies $p_{\theta_{tot}}(x^0_{tot})$ as
>
> Thanks for reading our response. **This understanding is close to the meaning of the IGD principle, but the generative diffusion model part of the IGD is missing.**
>
> The diffusion model is a type of generative model. It aims to model the true probability of data $q(x_{tot})$ through using $p_{\theta}(x_{tot}^0)$.  $p_{\theta}(x_{tot}^0)$ is parameterized by $\theta$. The goal of the diffusion model is to maximize the likelihood of $p_{\theta}(x_{tot}^0)$.  Once  a model satisfies the IGD is trained, the data $x_{tot}^0$ can be sampled through the Markov Chain. **IGD requires that sampled data $x_{tot}^0$ should match the true data distribution $q(x_{tot})$ well.** It is described in Line 195-196 *"$p_{\theta}(x_{tot}^0)$ learned to model ground truth distribution"*. This is achieved by requiring the generated data (or sampled data) $x_{tot}^0$ be generated by the diffusion model. **Note that  $x_{tot}^0$ is the generated data, and $x_{tot}$ is the true data we tried to model. They are not the same.**
>
> Besides requiring the generated data to match true data distribution, it requires factorization.
>
> >Is IGD just saying that each coordinate of $x^0_{tot} = (x^0_1, ... , x^0_N)$ is mutually independent?
>
> Not, in line 197 of the initial submission, $x^0_i \in \mathcal{R}^d$ is $d$-dimensional instead of just one-dimensional. IGD requires that $x^0_{tot}$ should close to true data distribution by using the diffusion models, as described in lines 195-196 of the initial submission.
>
> >What is the point of introducing the whole series $x_{tot}^{0:K}$? Those Markovian and Gaussian assumptions seem to be related to diffusion model, while IGD is still defined under general random variable $x^0_{tot} = (x^0_1,..., x^0_N)$.
>
> This diffusion-related concept is introduced because IGD is developed based on diffusion models. In line 204 of the initial submission, we had stated clearly that $p_{\theta}(x_{tot}^0)$ and $[p_{\theta_i}(x_i^0)]_{i=1}^N$ are diffusion models.
>
> $x^0_{tot} = (x^0_1,..., x^0_N)$ are **not general random variables**; they are random variables that match closely with the true data distribution thanks to diffusion models.
>
> >Finally, if you really want to focus on diffusion models, can you just say a diffusion model has IGD if each coordinate of initial state is mutually independent.
>
> Not, the IGD does not require that each coordinate (or each compenent) of the original data (state) is mutually independent. Instead, it requires that the **generated data (rather than the true data)** should match the true data distribution, and each component $x_i^0$ of the generated data $x_{tot}^0$ is mutually independent.
>
> For diffusion models that *each coordinate (component) of the initial state is mutually independent*, described by the reviewer, is called independent diffusion. It holds the oversimplified assumption that each component of the true data (rather than the generated data) is mutually independent. In independent diffusion, for each component of the jointed data, a distinct independent diffusion model is trained. It performs poorly, as  demonstrated by experiments and discussed in our submission. Please refer to Appendix A.2 for further discussion.
>
> We agree that the data generated through satisfying the IGD principle may not fail to model the complex relationship among each component of the data. Thus, after data are generated, we use the data factorization function (proposed by us) to model the complex relationship among data.
>
> **We will make the introduction of the diffusion model and the definitions better by introducing all necessary math backgrounds.**
>
>
>
> >Again, I am asking for formal statement showing that IGD generalizes IGM. Table 1 seems to be special cases.
>
> We have presented the formal statement proofs in the initial response regarding Table 1 (we had also emphasized this point in our revised paper in Appendix E.1). Could you please read it? We are eager to hear your opinion.
>
>
>
>
> >I should clarify my point. In my view, the proof technique relies heavily on the assumption that the process is Gaussian. Since I am not particularly interested in diffusion processes, I find the approach lacks broader insight.
>
> Thanks for reading the proofs. **We understand the research flavor of the reviewer towards the diffusion processes is different from ours.** However, we would like to point out the super powerful modeling ability of the diffusion models have already changed the world, especially in image and video generation. Our work wants to leverage such modeling ability into multi-agent decision making.

---

> > ### Author Response · Authors · 2024-11-24
> >
> > We appreciate your feedback and the opportunity to clarify these points. We hope our responses have clarified the points you've raised. If there are any further questions or aspects that remain unclear, please let us know.
> >
> > Sincerely,
> >
> > The Authors

---

> > ### Comment · Reviewer_zeea · 2024-11-26
> >
> > Thank you for your response. I assume you are referring to the IGO as the individual global max principle, which was referred to as IGM in your paper. However, your explanation about how IGD generalizes IGM (or IGO) still confuses me. IGM is a property of the state-action value function and does not depend on a diffusion model. How can IGD, defined under the diffusion model, be considered a generalization of IGM?

---

> ### Author Response · Authors · 2024-11-26
>
> Dear Reviewer zeea,
>
> Thanks for your response.
>
> >I assume you are referring to the IGO as the individual global max principle, which was referred to as IGM in your paper.
>
> No, in lines 205 to 212 of the initial submission, we show that IGD is a generalization of the IGM principle by using the property that the probability of the argmax operator $p(u_i)=1$ when $u_i=argmax(\tau_i, u_i)$, it is 0, otherwise.
>
> By the way, the IGO is shorted for the individual-global-optimal principle, which is a generalization of the individual-global-max (IGM) principle. The IGO principle requires that the product of the probability $\prod[p(u_i)]_{i=1}^N$ of each agent's optimal action is equal to the joint probability $p(u_1, u_2, ..., u_N)$ of the multi-agent system's optimal action.
>
> >IGM is a property of the state-action value function and does not depend on a diffusion model. How can IGD, defined under the diffusion model, be considered a generalization of IGM?
>
> **The diffusion model can be used to generate images, videos, actions, and state-action value functions. The IGM does not restrict the way how the state-action value function can be implemented or generated.**
> We have shown that the Diffusion model can generate actions and state-action value functions in this work. We have shown that the diffusion models can  generate four value functions in Table 1 and Table 11. Moreover, we have shown that the diffusion models can generate actions through multiple experiments. Other diffusion RL research , such as DiffusionQL [1] and LatentDiffusion [2], also confirmed this.
>
> We have provided formal proof of how the IGD generalizes the IGM principle in the initial response. It is located in the same page as this reply, its title is *"The IGD principle is a generalization of the IGM principle"*.
>
> We understand that the reviewer is still confused about how IGD generalizes IGM.
>
> **May I ask which part of the proofs confuse you?**
>
> **Are there any steps you think are not correct?**
>
> **Are you unclear on how the diffusion model generates actions or state-action value functions?**
>
> We are eager to discuss these questions with you.
>
>
>
> Best Regards,
>
> Authors
>
>
> **Reference**:
>
> [1] Zhendong Wang, Jonathan J. Hunt, and Mingyuan Zhou. Diffusion policies as an expressive policy class for offline reinforcement learning. In ICLR 2023.
>
> [2] Siddarth Venkatraman, Shivesh Khaitan, Ravi Tej Akella, John Dolan, Jeff Schneider, Glen Berseth. Reasoning with Latent Diffusion in Offline Reinforcement Learning. In ICLR 2024.

---

> > ### Comment · Reviewer_zeea · 2024-11-26
> >
> > I checked it again. It seems that your response does not address my review.
> > > For instance, given a value function satisfies IGM what is the corresponding diffusion process?
> >
> > Please specify the whole diffusion process.  Given $Q_{tot}$, what is the transition matrix for $p_{\theta_{tot}}(x_{tot}^{k-1}|x_{tot}^k)$ for each $k$? Do you use different diffusion process for different history $\tau_{tot}$?
> >
> > I thank the authors for their detailed responses to my questions.

---

> ### Author Response · Authors · 2024-11-28
>
> Dear Reviewer zeea,
>
> Thanks for reading our responses. **We are happy that the reviewer does not think there are any wrong steps in our proofs about how IGD generalizes IGM. We are happy that there is only one remaining concern.** The reviewer is concerned about the implementation of the diffusion model regarding how to generate actions or value functions given an observation or history $\tau$. As we mentioned in the very first response, we use **the standard diffusion approach**. We describe them first in an intuitive way and then describe the math details. After introducing this background knowledge, we answer the reviewer's questions in the end.
>
> Assuming that we have trained a diffusion model to generate images, we can sample a noise $x_{tot}^K$ from a Gaussian distribution. And then iteratively sample a new data $x_{tot}^{k-1}$ from $x_{tot}^{k}$ following a Gaussian Distribution. For this procedure, when sampling (generating) an image, we do not have control over which content to generate. If we want to control the content of an image through the class description (prompt) $y$ (e.g., dog, cat), there could be three ways to impose control:
>
> 1. Train a diffusion model for each possible $y$; this is very expensive.
> 2. Classifier-guidance diffusion: once a diffusion model is trained. Additionally, train a classifier $p(y|x_{tot})$, given an input image $x_{tot}$, the classifier outputs the class probability. The classifier will control the content it generates.
> 3. Classifier-free diffusion: It learns an unconditional diffusion model $p(x)$ and controls the content using condition score (a function of $y$).
>
> Approach 1 is the approach mentioned by the reviewer that trains a separate model for each $\tau_{tot}$. This is expensive; we do not adopt such an approach. Researchers have found that using the classifier-free approach [1] can lead to better performance than the classifier-guidance approach. Thus, we adopt the classifier-free diffusion approach. We have described them in lines 315--319 and 1816--1817 of the initial submission.
>
> Similarly, for the Individual-Global-Optimal (IGO) principle, we can use classifier-free diffusion to generate action following $\pi_{\theta_{tot}}(u_{tot} \mid \tau_{tot})$. Once a diffusion model is trained to generate actions. We can use a condition $\tau_{tot}$ to guide the diffusion process to generate the required action $u_{tot} \sim \pi_{\theta_{tot}}(u_{tot} \mid \tau_{tot})$.
>
> Specifically, to generate actions using a diffusion model, we follow a similar process as we would for image generation, but instead of generating images, we generate actions $u_{tot}$ for multi-agent.
>
>
> 1. **Forward Diffusion Process**:
>    We begin with an initial variable $x_{tot}^0 \sim q(u_{tot})$, where $q(u_{tot})$ is the true distribution of actions. data $x^k_{tot}$ is polluted by adding noise to $x^{k-1}_{tot}$ according following formula (Gaussian Distribution).
>
> $p(x^k_{tot} \mid x_{tot}^{k-1})  = \mathcal{N}(\sqrt{\alpha^k} x^{k-1}_{tot}, \sqrt{1 - \alpha^k} \boldsymbol{\epsilon^{k-1}})$
>
> where $\epsilon^{k-1} \sim \mathcal{N}(0, \textbf{I})$ is a Gaussian Noise, $k$ is the diffusion time step, $\alpha^k$ is a pre-specified hyper-parameter. $\mathcal{N} (\sqrt{\alpha^k} x^{k-1} _ {tot}, \sqrt{1 - \alpha^k} \boldsymbol{\epsilon^{k-1}})$ is a Gaussian Distribution with mean $\sqrt{\alpha^k} x^{k-1}$, and variance $\sqrt{1 - \alpha^k} \boldsymbol{\epsilon^{k-1}}$. In essense, $ p(x^k_{tot} \mid x^{k-1}_ {tot})$ is a probability, that the probability of sampling $x_{tot}^k$ condition on $x^{k-1}_{tot}$.
>
>
> 2. **Reverse Diffusion Process (Recovering Action following true action distribution)**:
>    The goal of the reverse diffusion process is to learn how to gradually denoise the latent variable $x_{\text{tot}}^k$ to recover the final action $u_{tot} = x_{\text{tot}}^0$. The probability $p _ { \boldsymbol {\theta}}$$( \boldsymbol{x} _ {tot}^ {k-1} \mid \boldsymbol{x} _ {tot} ^ k)$ of generating $\boldsymbol{x}_ {tot}^{k-1}$ based on $\boldsymbol{x} _ {tot}^{k}$ is defined as a Gaussian distribution, which is defined in the following formula.
>
> $p _ {\boldsymbol{\theta_ {tot}}}(\boldsymbol{x} _ {tot}^{k-1}$$ \mid \boldsymbol{x} _ {tot}^k)$$ = \mathcal{N} (\boldsymbol{x}_ {tot}^{k-1}; \boldsymbol{ \mu}$$ _ {\boldsymbol{\theta _ {tot}}}$$( \boldsymbol{x}^k_{tot}, k), \boldsymbol{ \Sigma}(k))$.
>
> $\boldsymbol{\mu}  _ {\theta  _{tot}}(\boldsymbol{x} _{tot}^k, k)$ and $\boldsymbol{\Sigma}(k)$ are the mean and variance of the Gaussian distribution. They are trained to maximize the true data distribution.

---

> ### Author Response · Authors · 2024-11-28
>
> 3. **Reverse Diffusion Process (Generating the Action for a specific $\tau_{tot}$)**: The reverse diffusion can be guided by the agent’s history $\tau_{\text{tot}}$ through classifier-free guidance to generate $u_{tot}$. In this way, the action $u_{tot} = \pi(\tau_{tot})$. The process follows the standard classifier-free approach [1]; to see the exact details of such a process, please refer to [1].
>
>
> To generate the individual action $u_i$ for agent $i$, the IGD principle requires that the parameters $\theta_{tot}$ can be factored into $\theta_i$, which can be used to sample individual action $u_i$ by using the diffusion process parameterized by $\theta_i$.
>
>
> 1. **Reverse Diffusion Process (Recovering Action following true action distribution)**:
>    The goal of the reverse diffusion process is to learn how to gradually denoise the latent variable $x_{i}^k$ to recover the final action $u_{i} = x_{\text{i}}^0$. The probability $p _ { \boldsymbol {\theta}}$$( \boldsymbol{x} _ {i}^ {k-1} \mid \boldsymbol{x} _{i}^k)$ of generating $\boldsymbol{x} _ {i}^{k-1}$ based on $\boldsymbol{x} _ {i}^{k}$ is defined as a Gaussian distribution, which is defined in the following formula.
>
> $p _ {\boldsymbol{\theta_ {i}}}(\boldsymbol{x} _ {i}^{k-1}$$ \mid \boldsymbol{x} _ {i}^k)$$ = \mathcal{N} (\boldsymbol{x}_ {i}^{k-1}; \boldsymbol{ \mu}$$ _ {\boldsymbol{\theta _ {i}}}$$( \boldsymbol{x}^k_{i}, k), \boldsymbol{ \Sigma}(k))$.
>
> $\boldsymbol{\mu}  _ {\theta  _{i}}(\boldsymbol{x} _{i}^k, k)$ and $\boldsymbol{\Sigma}(k)$ are the mean and variance of the Gaussian distribution. They are trained to maximize the true data distribution.
>
>
> 2. **Reverse Diffusion Process (Generating the Action for a specific $\tau_{i}$ or $\tau_{tot}$)**: The reverse diffusion can be guided by the agent’s history $\tau_{\text{i}}$ or aggregated history $\tau_{tot}$ through classifier-free guidance to generate $u_{i}$. The process follows the standard classifier-free approach [1]; to see the exact details of such a process, please refer to [1]. In this work, we follow the Centralized Training with Decentralized Execution (CTDE) paradigm. Thus, we choose to generate action $u_i$ conditions on local observation $\tau_i$ rather than $\tau_{tot}$.
>
>
> **We hope that the reviewer can have a better understanding of how we can use diffusion models following the IGD principle to generate actions.**
>
>
> >For instance, given a value function satisfies IGM what is the corresponding diffusion process?
>
> The diffusion process is the same as the above process to generate actions with only one difference.
>
> The IGD principle requires that $x_{tot} \in \mathcal{R}^{N \times d}$ and $x_i \in \mathcal{R}^d$. The dimension of $x_{tot}$ must be bigger than that of $x_i$. However, the value function $Q_{tot}$ and $Q_i$ are one-dimensional. For a $Q_{tot}(\tau_{tot})$ and $Q_i(\tau_i)$ that satisfy the IGM, a diffusion process $i$, parameterized by $\theta_i$, can be used to generated $Q_i(\tau_i)$ through condition on $\tau_i$. And then the aggregated results $[Q_i(\tau_i)] _ {i=0}^N$ is passed to the data factorization function $h$ (described in Sec.4.3.2) to obtained $Q_{tot}(\tau_{tot})$. The data factorization function $h$ can be any factorization that satisfies the IGM principle.
>
>
>
> > Given $Q_{tot}$, what is the transition matrix for $p_{\theta_{tot}}(x_{tot}^{k-1}|x_{tot}^k)$ for each $k$?
>
> These transition matrices are Gaussian Distributions, which are described in the above text and in the very first response to the reviewer's review. We follow the DDPM approach. Regardless of the type of data to be generated, the transition matrixes are the same.
>
> > Do you use different diffusion process for different history $\tau_{tot}$?
>
> No, we have described the reason in the above text.
>
>
> **REFERENCES**
>
> [1] Jonathan Ho and Tim Salimans. Classifier-free diffusion guidance. In NeurIPS, 2021

---

> ### Comment · Reviewer_zeea · 2024-11-29
>
> Thank you for the response.  The above argument is not rigors.  The existence of $\mu_{\theta_{tot}}$ in the reverse process is not proved, and $K$ is not specified.  For instance, if the optimal action is deterministic so that $|argmax_u Q_{jt}(\tau_{tot}, u)| = 1$, it seems highly unlikely to perfectly denoise it from a Markov chain with $\mu_{\theta_{tot}}$ and initial state Gaussian $x_{tot}^K$.  While it may be possible to approximate it, achieving exact recovery seems implausible.
>
> This may be due to the limitations of the diffusion model itself, but based on this discussion, I am not convinced the statement is true.

---

> > ### Author Response · Authors · 2024-11-29
> >
> > Thanks for your responses. **We appreciate your honesty about your stance on diffusion models and recognition of diffusion-based RL's complexities.** We agree that due to the random nature of the diffusion model, it is possible that true optimal action may not be recovered. **However, we would like to point out that the IGD principle is a requirement of the diffusion process. For the IGM case, the IGD requires such an ``exact recovery'' property; if a diffusion process satisfies such property, it satisfies the IGD principle.** If a diffusion model fails to satisfy such a property,  it fails to satisfy the IGD principle. The failure of a diffusion model to recover the optimal action exactly does not lead to the failure of the IGD principle. The IGD principle is a requirement of the diffusion process.
> >
> > >While it may be possible to approximate it, achieving exact recovery seems implausible.
> >
> > **Achieving exact recovery is indeed possible.**. In the initial submission, we have shown in Table 1 and Table 11 that DoF has exactly recovered the optimal action.** We have shown that our method performs better than other methods (e.g., QMIX satisfying the IGM) in the initial submission across multiple datasets, which indicates that our method can ``exactly recover'' the optimal actions better than other methods.
> >
> > > The existence of $\mu_{\theta_{tot}}$ in the reverse process is not proved.
> >
> > Such a value is defined as the mean of a Gaussian distribution for each backward step $k$. We have written its definition in Page 17, (B. 11). **Of course, it exists.** We have written them here for your convience. $\mu_ {\theta_{tot}}(x^k_{tot}, k) = \frac{1}{\sqrt{\alpha^k}}x^k_{tot} - \frac{1 - \alpha^k}{\sqrt{1 - \bar\alpha^k}\sqrt{\alpha^k}}\epsilon_{tot}^{\theta_{tot}}(x^k_{tot}, k)$, where $\alpha^k$ and $\bar{\alpha^k}$ are hyper-parameters. Please read the paper [1] about the mathematics of Diffusion models.
> >
> > > $K$ is not specified
> >
> > It is the number of diffusion steps, it is defined in the definition of the IGD principle.
> >
> > As we have demonstrated, DoF can exactly recover optimal actions in Table 1 and 11. **May I kindly ask the reviewer for some clarification on the following points, so that we can understand which aspects you are still uncertain about?**
> >
> > **Do you agree that the IGD principle is a requirement of the property of the diffusion process?** If not, could you please explain your reasoning?
> >
> >
> > **Do you agree that the IGD principle can be viewed as a generalization of the IGO principle?**  If not, could you please explain your reasoning?
> >
> > **Do you agree that the IGO principle can be viewed as a generalization of the IGM principle?** If not, could you explain your reasoning?
> >
> >
> > **Do you agree that similar to the randomness of the diffusion model, there exists randomness in RL, especially MARL?** If not, could you explain your reasoning?
> >
> > **Do you agree that due to the randomness of MARL and the approximation errors of neural networks, all existing neural network-based methods can fail to *exact recover* the optimal actions, which is required by the reviewer?**  If not, could you explain your reasoning?
> >
> >
> > We would like to point out that even with the ground-break work, DQN cannot guarantee the exact recovery of the optimal action too.
> >
> >
> > **Reference**:
> >
> > [1] Calvin Luo. Understanding diffusion models: A unified perspective. CoRR, abs/2208.11970, 2022

---

> > > ### Comment · Reviewer_zeea · 2024-11-29
> > >
> > > There should be a formal statement about IGD generalizing the IGM principle.
> > >
> > > > For the IGM case, the IGD requires such an ``exact recovery'' property; if a diffusion process satisfies such property, it satisfies the IGD principle.
> > >
> > > It seems that you only prove that IGD generalizes state-action functions that 1) satisfy IGM and 2) can be recovered by diffusion process.  Not all IGM state-action functions can be recovered by a diffusion process.  Note that a similar recovery assumption is also required for the diffusion process (e.g., page 6 of  Luo [1]).  For instance, for deterministic optimal action, the product $\bar{\alpha}^K<1$ (otherwise $x^K_{tot} = x^{0}_{tot}$).  Thus, the reverse diffusion process has a nonzero variance (B.12), which fails to recover the deterministic action.

---

> > > > ### Author Response · Authors · 2024-11-30
> > > >
> > > > Dear Reviewer zeea,
> > > >
> > > > Thanks for your reply. **We are happy that you agree that IGD is a generalization of the IGM principle, if the state-action functions can be recovered by diffusion process.** It seems that the remaining concern is whether the diffusion process can generate deterministic action which probability is exactly equal to 1.
> > > >
> > > > We understand the point of the reviewers that even when the DDPM process can generate actions with probability such as $(0.99999999999, 0.00000000001)$, it is not the same as the one-hot action probability $(1, 0)$. Although they are very close, they are different.
> > > >
> > > > We build the IGD principle based on the Gaussian-based diffusion processes which operate in **continuous spaces**, **rather than discrete spaces**. Thus, it may not exactly recover a deterministic action. Although it is possible to use a rounding function to change a probability close to 1 (e.g., 0.99999) to exactly become 1, we agree with the reviewer they are different.
> > > >
> > > >
> > > > Regarding deterministic actions, we would like to point out the existence of diffusion processes in discrete space [1][2]. For example, D3P [2] is a diffusion model that can generate discrete data $x \in \{0,1\}^d$. A deterministic action can be represented by a one-hot vector with $d$ dimension. If the $i$-th element of the one-hot vector is equal to 1, it means action $i$ will be selected deterministicly. **D3P can be used to generate the deterministic action required by the reviewer**. In the following we briefly describe the forward and the backward process of the discrete diffusion model D3P. Please refer to [2] for more math details.
> > > >
> > > > 1. Forward Diffusion Process
> > > >
> > > > In the D3P framework, the forward diffusion process corrupt the discrete data through a Markov chain with categorical distributions. The forward process is defined as:
> > > > $$
> > > > q(x_t|x_{t-1}) = \text{Categorical}(x_t; x_{t-1}Q_t)
> > > > $$
> > > >
> > > > where $x_t$ is the discrete state at time $t$. $Q_t$ is predefined transition matrix.
> > > >
> > > > The transition matrix  $Q$  is designed to gradually corrupt the data by increasing uncertainty, moving towards a uniform distribution as  $t \to T$ . The cumulative forward process from  $x_0$  to  $x_t$  is:
> > > >
> > > > $$
> > > > q(x_t|x_0) = \text{Categorical}(x_t;x_0 \bar{Q}_t), \text{with}\ \ \bar{Q}_t = Q1Q1\cdots Q_t.
> > > > $$
> > > >
> > > >
> > > > 2. Backward Diffusion Process
> > > >
> > > > The reverse diffusion process aims to reconstruct the original data from the noisy discrete data $x_T$ by learning the reverse transitions $p_\theta(x_{t-1}|x_t)$:
> > > >
> > > > $p_\theta(x_{t-1} | x_t) = \text{Categorical}(x_{t-1}; \mathbf{P}_\theta(x_t))$
> > > >
> > > > where $\mathbf{P}_\theta(x_t)$ represents the learned reverse transition probabilities, parameterized by $\theta$.
> > > >
> > > > Please refer to [2] about the details of ${Q} _ t$ and $\mathbf{P} _ \theta(x_t))$.
> > > >
> > > >
> > > > We agree with the reviewer that we assume that the diffusion process can generate actions with discrete probability exactly. **We will soften our claim as *the IGD principle can be viewed as a generalization of the IGM principle, if the diffusion process can generate deterministic actions exactly***
> > > >
> > > > We will discuss this limitation in our manuscript and specify that our generalization holds under the assumption of exact recoverability provided by diffusion models. This adjustment acknowledges the assumption that the diffusion process must be capable of exact probability generation for the generalization to hold.
> > > >
> > > >
> > > > **May I kindly ask the reviewer do you agree with the soften claim?** We are eager to discuss this with you.
> > > >
> > > > Best Regards
> > > >
> > > > Authors
> > > >
> > > >
> > > >
> > > > **Reference**:
> > > >
> > > > [1] Emiel Hoogeboom, Didrik Nielsen, Priyank Jaini, Patrick Forré, Max Welling, "Argmax Flows and Multinomial Diffusion: Learning Categorical Distributions", In NeurIPS 2021.
> > > >
> > > > [2] Jacob Austin, Daniel D. Johnson, Jonathan Ho, Daniel Tarlow, Rianne van den Berg, "Structured Denoising Diffusion Models in Discrete State-Spaces", In NeurIPS 2021.

---

> > > > > ### Comment · Reviewer_zeea · 2024-12-01
> > > > >
> > > > > I appreciate the author's response and the detailed discussion addressing my concerns. I have no further questions.

---

> ### Author Response · Authors · 2024-12-01
>
> We are glad to see that the reviewer has no further questions and appreciates the feedback provided. Given our detailed responses, we believe that some of the initial concerns may have stemmed from misunderstandings, particularly regarding the nature of the diffusion process, the conditions under which exact recovery is possible, and the implementation of diffusion processes for deterministic actions. We notice the reviewer rated our work as "Reject" and would like to respectfully ask if this rating was influenced by such misunderstandings. If clarity was a major issue, we have already addressed this in the responses. We kindly ask that you reconsider your evaluation in light of these improvements.
>
> At the same time, we hope that through this process, the authors and the reviewer can contribute to the ICLR community by providing an example of how to offer thorough, constructive feedback and how to effectively respond to reviewers' concerns. We deeply value every piece of feedback in the review process and aim to further improve and advance the field.

---

### Official Review · Reviewer_5VM5 · 2024-11-23

**Soundness:** 3
**Presentation:** 2
**Contribution:** 4
**Rating:** 8
**Confidence:** 3

**Summary:**

The paper proposes to use Individual-Global-Identically-Distributed (IGD) principle for the training of diffusion models in multi-agent decision making (MADM). This IGD, requiring the outcome of multi-agent diffusion model should be identically distributed as the collective outcomes from multiple individual agent diffusion models, is a generalization of the individual global max (IGM) principle. The authors proposes DoF a way to do factorization in this setting and validated their proposed approaches via experiments.

**Strengths:**

+ IGD is a very interesting and promising idea. I can see its potential in multi-agent diffusion.
+ IGD captures recovers IGM or risk-sensitive IGM principle.

**Weaknesses:**

- While the ideas are interesting, presentation of the paper makes it difficult to fully appreciate the contribution. For example, the algorithms are missing from the main manuscript.
- Related to the previous point, it is unclear how to formulate a diffusion algorithm based on Theorem 1.
- Some of the notations are not well defined, such as $\epsilon$ in the theorem, and what role $\theta$ play in Eq (2).
- The POMDP setting should be described a bit more in the main text, for instance, there is one environment that all agents are interacting with and the transition probability depends on the product space of all agent's actions.
- Typos: I list a few here, but please go through the paper one more time to double check.  For example, in definition 1, should be "there exist ... functions", the sentence before section 3, "to for $x^k$".

**Questions:**

-  How long did the proposed methods take to train compared to other methods? The paper reported inference time, but not the training time.
- I was a bit confused by the overall setup, are the agents doing global training and then individual sampling of the noise after doing the factorization? Is there a way for the agents to train simultaneously?

I just got the invitation to review this paper last night, and I understand that the authors probably won't have time to address all of my questions/comments. I hope these comments will be helpful for improving the manuscript for publication (at ICLR or elsewhere).

---

> ### Author Response · Authors · 2024-11-25
>
> Thank you for your in-depth emergency review, especially for providing thoughtful and constructive feedback under such a limited time. We greatly appreciate the effort you invested in carefully evaluating our work. Below, we address your main points:
>
>
> >While the ideas are interesting, presentation of the paper makes it difficult to fully appreciate the contribution. For example, the algorithms are missing from the main manuscript.
>
> Thank you for pointing this out. We included the pseudocode for DoF-Trajectory and DoF-Policy in Appendix C.3 and C.4, respectively. In the revised version, we have revised Section 4.4 to enhance clarity by adding the sentence: "Please refer to the details of the agents in Appendix." The organization of this paper will be reorganized to describe the algorithms better.
>
>
> > Related to the previous point, it is unclear how to formulate a diffusion algorithm based on Theorem 1.
>
> >I was a bit confused by the overall setup, are the agents doing global training and then individual sampling of the noise after doing the factorization? Is there a way for the agents to train simultaneously?
>
>
> During the training phase, we globally sample an overall noise vector. Each agent learns a noise predictor (Denoiser), and their outputs are combined through an noise factorization function $f$ to approximate this overall noise, enabling efficient collaborative training.
>
>
> During the inference phase, the diffusion method supports each agent independently sampling and generating its own trajectory or action, ensuring the capability for decentralized execution while maintaining alignment with the design philosophy of the CTDE framework.
>
> We have provided pseudocode specifically for the training phase and execution phase of the DoF diffusion model, which elaborates on how the diffusion process is implemented. The pseudocode below offers a step-by-step description of centralized training and decentralized execution (generation). The full pseudocodes of DoF-trajectory and DoF-policy are presented in Appendix C.3 and C.4.
>
> **Centralized Training**
>
> 1: **repeat**:
> 2: &nbsp;&nbsp;&nbsp; $\mathbf{x}^0_{\text{tot}} \sim q(\mathbf{x}_{\text{tot}})$  (sample global data)
>
> 3: &nbsp;&nbsp;&nbsp; $k \sim \text{Uniform}(\{1, \dots, K\})$   ($k$ is the diffusion time step)
>
> 4: &nbsp;&nbsp;&nbsp; $\epsilon \sim \mathcal{N}(\mathbf{0}, \mathbf{I}) \in \mathcal{R}^{d \times N}$  (sample global noise)
>
> 5: &nbsp;&nbsp;&nbsp; $x^k_{tot}=\sqrt{\bar{\alpha}^k} x^{k-1}_{tot} + \sqrt{1 - \bar{\alpha}^k}\epsilon$
>
> 6: &nbsp;&nbsp;&nbsp; $x^k_i=x^k_{tot}[(i-1) \times d: i\times d]\; $ $i \in [1,..N]$
>
> 7: &nbsp;&nbsp;&nbsp; $\epsilon_{\text{tot}} = f(\epsilon_{\theta_1}^1(\mathbf{x}^k_1, k), \epsilon_{\theta_2}^2(\mathbf{x}^k_2, k), \dots, \epsilon_{\theta_N}^N(\mathbf{x}^k_N, k))$  ($f$ is the noise factorization function)
>
> 8: &nbsp;&nbsp;&nbsp; Take gradient descent step on:  &nbsp;&nbsp;&nbsp; $$\nabla_\theta \|\epsilon - \epsilon_{\text{tot}}\|^2$$
>
> 9: **until** convergence.
>
>
> $\epsilon_{\theta_i}^i(\mathbf{x}^k_i, k)$ is a noise predictor for agent $i$ that predicts the $(i-1) \times d$ to the $i \times d$ index of the noise $\epsilon$.
>
>
> **Decentralized Execution (generation)**
>
> 1: $\mathbf{x}_i^K \sim \mathcal{N}(\mathbf{0}, \mathbf{I})$  (for each agent $i$)
>
> 2: **for** $k = K, \dots, 1$ **do**
>
> 3: &nbsp;&nbsp; $\epsilon _ \theta^i(\mathbf{x}^k _ i, k)$  (noise prediction by each agent $i$)
>
> 4: &nbsp;&nbsp; Update state for each agent $i$:
>    &nbsp;&nbsp; $\mathbf{x}^{k-1} _ i = \frac{1}{\sqrt{\alpha_k}} \left(\mathbf{x}^k _ i - \frac{1 - \alpha_k}{\sqrt{1 - \bar{\alpha} _ k}} \epsilon _ \theta^i(\mathbf{x}^k _ i, k)\right) + \sigma _ k \mathbf{z},$   &nbsp;&nbsp; where $\mathbf{z} \sim \mathcal{N}(\mathbf{0}, \mathbf{I})$ if $k > 1$, else $\mathbf{z} = \mathbf{0}$
>
> 5: **end for**
>
> 6: **return** $\mathbf{x}^0 _ i$  (final trajectory or action for each agent $i$)
>
> We will add the above algorithms into the main text.
>
> > Some of the notations are not well defined, such as $\epsilon$
>  in the theorem, and what role $\theta$
>  play in Eq (2).
>
> $\epsilon$ is a random noise. $\epsilon^{\theta_i}_i(\mathbf{x}^k_i, k)$ is the noise predictor of agent $i$ to predict the noise that added at time $k$ for the data $x_i$. The noise predictor $\epsilon^{\theta_i}_i(\mathbf{x}^k_i, k)$ is parameterized by $\theta_i$. In this work, we assume that $\theta_i \subset \theta$. $\theta$ is the aggregated parameters of all the noise predictors. We will describe them more clearly in the paper.
>
>
> >The POMDP setting should be described a bit more in the main text, for instance, there is one environment that all agents are interacting with and the transition probability depends on the product space of all agent's actions.
>
> We appreciate this suggestion. We will expand the description of the POMDP setting in the main text, including details about shared environments, joint actions, and how agents interact with the system.

---

> > ### Author Response · Authors · 2024-11-25
> >
> > >Typos: I list a few here, but please go through the paper one more time to double check. For example, in definition 1, should be "there exist ... functions", the sentence before section 3, "to for x^k".
> >
> > Thank you for pointing out the typos in our manuscript. We have carefully addressed the issues you mentioned:
> > 1. In Definition 1, we have corrected **"there exists ... functions"** to **"there exist ... functions"** to ensure proper subject-verb agreement.
> > 2. In the sentence before Section 3, we have replaced **"to for $x^k$"** with **"for $x^k$"** for clarity and correctness.
> >
> > Additionally, we will thoroughly review the manuscript again to identify and correct any remaining typographical errors. We appreciate your attention to detail and your feedback to improve the quality of our work.
> >
> >
> > >How long did the proposed methods take to train compared to other methods? The paper reported inference time, but not the training time.
> >
> > Thank you for your question about the training time of our methods.
> >
> > We have conducted additional experiments comparing training and convergence times of DoF, MADIFF, and MABCQ on both the 3m and 8m maps. These results are now included in the appendix as follows:
> >
> > |Maps|Methods|Training Time (1M steps)|
> > |----|----|----|
> > |    |DoF  |  48h|
> > |3m  |MADIFF | 60h|
> > |    |MABCQ| 20h |
> > |  |DoF | 54h|
> > | 8m |MADIFF| 70h|
> > |  | MABCQ| 23h|
> >
> >
> >
> >
> > The training time of DoF is lower than the training time of MADIFF. DoF takes longer training time than MABCQ, a well-known offline reinforcement learning method. The training time can be further reduced by using a variational auto-encoder (VAE) to compress the data. In our prior experiments, detailed in Appendix D.5 of the initial submission, we evaluated the training time of DoF and DoF+VAE on the SMAC 3m map. The results showed that DoF+VAE reduced training time by approximately 27% (from 48h to 39h) with a 16.6% performance trade-off.

---

> > > ### Comment · Reviewer_5VM5 · 2024-11-26
> > >
> > > Thank you to the authors for providing very detailed and thorough responses to my and other reviewers' comments. You went above and beyond in what is usually expected out of a conference rebuttal process. Assuming the authors manage to squeeze all the suggested edits within the page limits, these responses will significantly improve the clarity of the paper, which was my main concern during the initial review. I still think this is a very interesting topic and the proposed approach seems quite appealing (not super fast during the training phase though). I have adjusted my score to 8.

---

> > > > ### Author Response · Authors · 2024-11-27
> > > >
> > > > Dear Reviewer 5VM5,
> > > >
> > > > Thanks very much for your support and kind words. To increase the readability of our work, we have added the pseudo code of the DoF algorithms into the main text. We have moved the experimental results about SMACv2 and MPE to the appendix to make the paper within page limits.
> > > >
> > > > Best regards,
> > > >
> > > > Authors

---

### Official Review · Reviewer_StRZ · 2024-11-25

**Soundness:** 3
**Presentation:** 2
**Contribution:** 2
**Rating:** 6
**Confidence:** 4

**Summary:**

Please see the strengths and weaknesses sections.

**Strengths:**

- This paper tries to solve an important problem in Diffusion-model-based offline MARL, that is, the use of the centralized training with decentralized execution (CTDE) paradigm. As noted in [1], this is a research gap in this field.

- The algorithm design is simple yet effective, which is demonstrated in a series of benchmarks through comparisons with various baselines.

### Reference:

[1] Chen, et al. "Deep Generative Models for Offline Policy Learning: Tutorial, Survey, and Perspectives on Future Directions." TMLR, 2024.

**Weaknesses:**

- The definition of IGD seems to assume the independence among agents' behaviors/rollout trajectories. This assumption is also shown in the selection of the noise factorization function, which is a simple concatenation operator.

- The first paragraph of Section 4.3 is a little bit of confusing. A refined version of the overview should be provided.

- The provided codebase does not involve a detailed instruction to reproduce **ALL** numerical results in the paper. The code files are also not complete. If the authors fix this, I will improve my rating.

**Questions:**

- Can you intuitively explain why the agents trained with MADiff still suffer from collisions and perform worse than the ones trained with DoF in Figure 1? I suppose their main difference lie in scalability which shouldn't be an issue for the simple illustrative example.

- Functionally, \(f\) and \(h\) appear to be similar. Why is it necessary to train an additional \(h\)?

- (This is only a suggestion for future works.) Policy distillation methods can be an important category of baselines to compare with. Specifically, you can apply CDG to learn a centralized policy and distill a set of decentralized policies for each agent from the centralized agent, which is also a form of factorization.

---

> ### Author Response · Authors · 2024-11-26
>
> Dear Reviewer,
>
> Thanks for your time and effort in this emergency review. We are happy that the reviewer considers the problem we are addressing important and considers our algorithm as *simple yet effective*. We address your concerns as follows.
>
> >This paper tries to solve an important problem in Diffusion-model-based offline MARL, that is, the use of the centralized training with decentralized execution (CTDE) paradigm. As noted in [1], this is a research gap in this field.
>
> Thank you for pointing out this survey paper[1]. We have read the paper, it provides a very comprehensive summary of diffusion-based methods, including the main comparison methods in our paper, MADIFF and DOM2. Additionally, we noticed its emphasis and elaboration on addressing an important issue in diffusion-model-based offline MARL, namely the adoption of the centralized training with decentralized execution (CTDE) paradigm, which aligns with the research objective of our work. It is written in [1] that *"Clearly, more extensions can be developed regarding DM-based MARL, such as MARL algorithms for fully competitive or mixed (partially cooperative/competitive) multi-agent tasks, and integration of DM with state-of-the-art centralized training & decentralized execution (CTDE) MARL methods."*
>
> To emphasize the motivation and enhance the comprehensiveness of related work, we have included this survey in Section 1, Line 47~48 of the revised version. Moreover, we plan to supplement and refine our writing about future work based on related studies mentioned in the survey to better position and highlight the contributions of our work. Thank you again for bringing this valuable survey paper to our attention!
>
>
> **Reference**:
>
> [1] Chen, et al. "Deep Generative Models for Offline Policy Learning: Tutorial, Survey, and Perspectives on Future Directions." TMLR, 2024.

---

> > ### Author Response · Authors · 2024-11-26
> >
> > >The definition of IGD seems to assume the independence among agents' behaviors/rollout trajectories. This assumption is also shown in the selection of the noise factorization function, which is a simple concatenation operator.
> >
> >
> > Albeit IGD requires that **the generated data (rather than true data)** of each agent should be independent, we would like to argue that an independent assumption is held in the famous individual-global-max (IGM) principle, which is widely used by multi-agent researchers.
> >
> > The IGM principle requires that $(\bar{u_1}, ..., \bar{u_N}) = \bar{u_{tot}}$, where $\bar{u_i} = argmax Q_u(\tau_i, u)$, $\bar{u_{tot}}= argmaxQ_{tot}(\tau_{tot}, u)$. $u_i$ is the action of agent $i$, $u_{tot}$ is the joint action, $\tau_i$ is the local observation history of agent $i$, and $\tau_{tot}$ is the aggregared observation. There is an independent assumption of the IGM principle. It implicitly assumes that each agent can make the optimal decision independently based on individual utility function $Q_i(\tau_i, u_i)$. However, the individual utility function does not guarantee optimal decisions. To obtain the true optimal action, the utility function should be $Q_i(\tau_{tot}, u_i)$, where $\tau_{tot}$ is the observation-action history for all the agents. Such an independent assumption for the IGM principle may not hold for multi-agent tasks that require communication among agents.
> >
> > The independent distribution requirement in DoF requires that each $x_i$ is independently distributed with each other. Let's take an optimal discrete action generation (OAG) case as an example. In this case, each agent must decide (generate) independently its optimal discrete action without communication. It means that each $x_i = \bar{u_i}$ should be decided (generated) independently of each other. In the context of IGD, "decide (generate) independently" means "independently distributed." That is, each agent $i$ should make their decision $x_i = \bar{u_i}$ based on its local observation history $\tau_i$ without relying on others. And their collective decision $(\bar{u_1}, ..., \bar{u_N})$ is equal to the optimal joint decision $x_{tot} = \bar{u_{tot}}=\arg max_{u}Q_{tot}(\tau_{tot}, u)$. **From this point of view, the independent distribution requirement in DoF is the generalization of the hidden independent assumption used in the IGM principle.**
> >
> >
> > The IGD generates data $x_{tot}^0$ to match the distribution of true data $x_{tot}$. **The approach that assumes the true data (e.g., trajectory) of each agent is independent is called independent diffusion in this work**. Independent diffusion does not try to learn the collective behaviors. It performs poorly, as it is demonstrated in the initial submission. We have discussed the difference between the independent diffusion method and ours in Appendix A.2 of the initial submission.
> >
> > The IGD principle requires that **the generated data** of each agent should be mutually independent. Moreover, it requires the outcome of the centralized diffusion model should be identically distributed as the collective outcomes from multiple individual agent diffusion models. The noise factorization function **learns collectively (rather than separately) agent behaviors**. Moreover, **it does not assume that true data (e.g., trajectory) of each agent are independent**. After the end of the sampling process, a data factorization function is used to model the relationship among the data generated by each agent.
> >
> > **We have tried more powerful noise factorization functions such as the QMIX and the Attention function in Table 6**. They can model more complex relationships among agents. However, QMIX performs poorly, as it cannot guarantee the generated noise is still Gaussian. The Attention function (Atten) performs better than the concat operator. However, it is not chosen, as it should be centrally executed. For the Atten function, each agent must read the noise from other agents. The concat and wconcat functions are choosen due to their simplicity and effectiveness.

---

> ### Author Response · Authors · 2024-11-26
>
> >The first paragraph of Section 4.3 is a little bit of confusing. A refined version of the overview should be provided.
>
> The training and the testing procedure of the DoF framework is described as follows.
>
>
> During the training phase, we globally sample an overall noise vector. Each agent learns a noise predictor (Denoiser), and their outputs are combined through a noise factorization function $f$ to approximate this overall noise, enabling efficient collaborative training. During the inference phase, the diffusion method supports each agent independently sampling and generating its own trajectory or action, ensuring the capability for decentralized execution while maintaining alignment with the design philosophy of the CTDE framework.
>
> Regarding the specific implementation of the training process, we have provided pseudo-code and step-by-step instructions, which you can refer to for more details.
>
> **Centralized Training**
>
> 1: **repeat**:
>
> 2: &nbsp;&nbsp;&nbsp; $\mathbf{x}^0 _ {\text{tot}} \sim q(\mathbf{x} _ {\text{tot}})$  (sample global data)
>
> 3: &nbsp;&nbsp;&nbsp; $k \sim \text{Uniform}(\{1, \dots, K\})$    ($k$ is the diffusion time step)
>
> 4: &nbsp;&nbsp;&nbsp; $\epsilon \sim \mathcal{N}(\mathbf{0}, \mathbf{I}) \in \mathcal{R}^{d \times N}$  (sample global noise)
>
> 5: &nbsp;&nbsp;&nbsp; $x^k _ {tot}=\sqrt{\bar{\alpha}^k} x^{k-1} _ {tot} + \sqrt{1 - \bar{\alpha}^k}\epsilon$
>
> 6: &nbsp;&nbsp;&nbsp; $x^k _ i = x^k _ {tot}[(i-1) \times d: i\times d]\; $ $i \in [1,..N]$
>
> 7: &nbsp;&nbsp;&nbsp; $\epsilon_{\text{tot}} = f(\epsilon _ {\theta _ 1}^1(\mathbf{x}^k _ 1, k), \epsilon _ {\theta _ 2}^2(\mathbf{x}^k _ 2, k), \dots, \epsilon _ {\theta _ N}^N(\mathbf{x}^k _ N, k))$  ($f$ is the noise factorization function)
>
> 8: &nbsp;&nbsp;&nbsp; Take gradient descent step on:
>
> &nbsp;&nbsp;&nbsp; $\nabla _ \theta \|\epsilon - \epsilon _ {\text{tot}}\|^2$
>
> 9: **until** convergence.
>
>
> $\epsilon _ {\theta_i}^i(\mathbf{x}^k _ i, k)$ is a noise predictor for agent $i$ that predicts the $(i-1) \times d$ to the $i \times d$ index of the noise $\epsilon$.
>
>
> **Decentralized Execution (generation)**
>
> 1: $\mathbf{x}^K_i \sim \mathcal{N}(\mathbf{0}, \mathbf{I})$  (for each agent $i$).
>
> 2: **for** $k = K, \dots, 1$ **do**:
>
> 3: &nbsp;&nbsp;&nbsp; $\epsilon _ {\theta}^i(\mathbf{x}^k_i, k)$  (noise prediction by each agent $i$).
>
> 4: &nbsp;&nbsp;&nbsp; Update state for each agent $i$:
> &nbsp;&nbsp;&nbsp; $\mathbf{x}^{k-1} _ i = \frac{1}{\sqrt{\alpha_k}} \left(\mathbf{x}^k_i - \frac{1 - \alpha_k}{\sqrt{1 - \bar{\alpha}_k}} \epsilon _ {\theta}^i(\mathbf{x}^k_i, k)\right) + \sigma _ k \mathbf{z},$
> &nbsp;&nbsp;&nbsp; where $\mathbf{z} \sim \mathcal{N}(\mathbf{0}, \mathbf{I})$ if $k > 1$, else $\mathbf{z} = \mathbf{0}$.
>
> 5: **end for**.
>
> 6: **return** $\mathbf{x}^0_i$  (final trajectory or action for each agent $i$).
>
> We have added the above procedures in Sec. 4 of the main text. Can you please tell us which part of the overview is confusing after reading our response? We are happy to revise them accordingly.
>
> >The provided codebase does not involve a detailed instruction to reproduce ALL numerical results in the paper. The code files are also not complete. If the authors fix this, I will improve my rating.
>
> We had discussed our results with an author of a diffusion-based MARL approach before the ICLR submission. We quote the author's reply as follows *"I understand that reproducing results from other methods can be challenging."* In DoF, we try our best to reproduce the results of other competing methods.
> To address your concerns, we have updated the codebase with a detailed `README.md` file that includes instructions for environment setup, data preparation, and step-by-step guidance to run experiments thus to reproduce the numerical results.
> Furthermore, **following our tradition, we will release the full code for all experiments presented in the paper on GitHub if the paper is published**, to ensure complete transparency and reproducibility of our work.

---

> ### Author Response · Authors · 2024-11-26
>
> >Can you intuitively explain why the agents trained with MADiff still suffer from collisions and perform worse than the ones trained with DoF in Figure 1? I suppose their main difference lie in scalability which shouldn't be an issue for the simple illustrative example.
>
>
> Due to the lack of factorization, each MADiff agent must condition **based on a noisy condition rather than a clean condition as ours**. Thus, it performs poorly and causes collisions due to noisy conditions. Let's think we have trained a diffusion process to generate agents' destinations using a text (the condition) as a prompt. The prompt (input to the agent network) is written as (station, city, Paris) during training. Due to lack of factorization, during execution, the MADiff process is given an input (prompt), written as (xxx, city, xxx), where xxx represents random text. The generated destination is expected to not match the prompt (station, city, Paris). As the generated destination is random, agents may cause collisions. We discussed the difference between our approach and MADiff in Appendix A.2 of the initial submission.
>
> Specifically, MADIFF trains $1$ (rather than $n$) diffusion process for all the agents. After the diffusion process is trained, each agent uses the same diffusion process to generate data. During training, the MADIFF agent is trained conditions on aggregated observation-history $\tau_{tot} = (\tau_1, ..., \tau_n)$ and a return value $R$ to generate data $x_{tot}$. Due to lack of factorization, during execution, MADIFF conditions on $\tau^i_{tot}$ and a return value $R$ to generate data $x_i$. $\tau^i_{tot} = (z_1, ..., \tau_i, ..., z_n)$, where $\tau_i$ is the local observation-history of $i$, and $z_1, ...z_n$ are random noises. As agent $i$ can only observe $\tau_i$, except for the $i$-th element, all the other elements in $\tau^i_{tot}$ are filled with random noise. **Its signal-to-noise ratio is low, only $1/n$**. Thanks to the modeling ability of the diffusion model, MADIFF can still generate data. However, it is unclear how the generated data matches the original data distribution. Different from MADIFF, DoF agent $i$ conditions on its local observation $\tau_i$ rather than noisy observations.
>
> >Functionally, (f) and (h) appear to be similar. Why is it necessary to train an additional (h)?
>
> Although the noise factorization function (f) and the data factorization function (h) can share the same math form, they are used for different purposes.
>
> The noise factorization function is used to learn factored diffusion processes. To satisfy the IGD principle, the noise factorization may not be able to sufficiently model complex relationships among agents. Because of this, we use the data factorization function (h) to model the complex relationship among the data generated by agents. We have demonstrated the importance of the data factorization function (h) in the appendix (Table 11 and Table 14) of the initial submission. Moreover, we have conducted new experiments about the data factorization function, the experimental results on heterogeneous SMAC environments are included as follows.
>
>
> | Maps | Dataset | $h = \text{Concat}$ | $h = \text{WConcat}$ | $h = \text{Atten}$ |
> |------|---------|-----------------------|-------------------------|-------------------------|
> |   | Good    | 11.3±0.9            | 12.8±0.8     |      15.2±0.7       |
> |  3s5z_vs_3s6z     | Medium  | 9.4±0.7            | 11.9±0.7    |     12.8±0.5          |
> |      | Poor    | 6.8±0.3    | 7.5±0.2|    8.2±0.3       |
> |   | Good    | 15.5±1.0              | 18.5±0.8                | 19.5±0.3               |
> |  2s3z     | Medium  | 14.8±0.8              | 18.1±0.9                | 18.5±0.3               |
> |      | Poor    | 9.6±1.1               | 10.0±1.1                 | 10.2±0.7               |
>
> As shown in both Table 14 and the heterogeneous 2s3z and 3s5z_vs_3s6z environment, **the choice of data factorization $h$ has a significant impact on performance.** The results are added in Table 24 of the revised submission.

---

> ### Author Response · Authors · 2024-11-26
>
> >(This is only a suggestion for future works.) Policy distillation methods can be an important category of baselines to compare with. Specifically, you can apply CDG to learn a centralized policy and distill a set of decentralized policies for each agent from the centralized agent, which is also a form of factorization.
>
> Thanks for the novel view about policy distillation is a form of factorization. Indeed, distilling a centralized policy into a set of individual policies is a form of factorization. They could be an important category of baseline to compare. We have performed a literature survey regarding policy distillation methods for offline MARL. The only one we found is MADT+Distillation[1]. We have compare DoF with MADT on 18 offline datasets. The experiment results show that DoF perform better than MADT.  The MADT+Distillation [1] is not open source yet, it is challenging to implement their approach and compare them during the rebuttal period. **As we have already compared [1] in the related work of the initial submission, we may miss some related offline MARL policy distillation work,  can you please tell us more related MARL policy distillation work?**
>
> **Reference**:
>
> [1] Tseng, et al. "Offline Multi-Agent Reinforcement Learning with Knowledge Distillation."  NeurIPS, 2022.

---

> > ### Author Response · Authors · 2024-11-28
> >
> > Dear Reviewer StRZ,
> >
> > Thanks for your emergency reviews. We have responded to your reviews 2 days ago. We have uploaded a readme file to describe how to reproduce the experiments. Are there any remaining concerns? We are eager to discuss this with you.
> >
> > Best Regards,
> >
> > Authors

---

> ### Author Response · Authors · 2024-11-29
>
> Dear Review StRZ,
>
>
> Thank you for your timely and thoughtful feedback. We greatly appreciate your recognition of our algorithm's strengths and the important research gap we address in Diffusion-model-based offline MARL. We also thank you for sharing the excellent survey on diffusion-based methods.
>
> We have addressed all of your concerns and have uploaded a README file with the necessary details for experiment reproducibility.
>
> We have received positive feedback from another reviewer, noting that our responses addressed the reviewer's concern and led to an improved rating. We hope our responses address your concerns and would appreciate it if you could reconsider your rating.
>
> Best Regards,
>
> Authors

---

> > ### Comment · Reviewer_StRZ · 2024-12-01
> >
> > Thank you for your efforts. This paper presents a novel contribution to diffusion-model-based multi-agent reinforcement learning (MARL) and provides a solid set of experimental results.
> >
> > However, I still have a concern regarding the numerical results presented (e.g., Table 2), as they differ from those reported in the baseline papers. For instance, the performance of MADT on 2s3z and 3s5z vs. 3s6z in its original paper appears better than the results reported here. Similarly, the performance of MADiff on 3m and 2s3z in its original paper is superior to the corresponding results in the submission. This discrepancy leaves me uncertain about which results to trust.
> >
> > That said, as promised, I will raise my rating to 6.

---

> ### Author Response · Authors · 2024-12-02
>
> Dear Reviewer,
>
> Thank you for your thoughtful feedback and for acknowledging the contributions of our work. We appreciate your concerns regarding the discrepancies in numerical results compared to the baseline papers, and we would like to clarify these differences.
>
> **1) Performance of MADT on 2s3z and 3s5z vs. 3s6z**:
>
> The variations in MADT’s performance arise primarily due to the use of different datasets. In our paper, we employ the OGMARL dataset [1], where data for the SMAC discrete environment is collected using the QMIX algorithm. In contrast, the original MADT paper utilizes datasets collected via MADDPG. This difference significantly impacts the reward distributions and the number of trajectories, leading to performance variations.
>
> For clarity, here are the dataset statistics:
>
> Dataset from MADT [Original Paper]:
>
> | |2s3z | 3s5z vs 3s6z |
> |--|--|--|
> |number of trajectories|4177846 |1542571|
> |return distribution|19.93±0.09 | 18.35 ±2.04|
>
>
>
> Dataset from OGMARL [Our Paper]:
> | |2s3z | 3s5z vs 3s6z |
> |--|--|--|
> |number of trajectories|107900 |101335|
> |return distribution|18.2±2.9 |17.0±3.3|
>
> These statistics are sourced from Appendix C of both the MADT and OGMARL papers. As evident, the datasets differ in size and reward distribution, which explains the discrepancies in performance.
>
>
> **2) Performance of MADiff on 3m and 2s3z**:
>
> The results we reported for MADiff are based on our experimental runs. Variations from the original paper can occur due to the stochastic nature of reinforcement learning and the challenges in reproducing exact results. Factors such as different random seeds, environmental setups, and implementation details can lead to performance differences.
>
> To promote transparency and facilitate replication, we have open-sourced our code and provided detailed commands for running the experiments. This allows others to reproduce our results and understand the experimental conditions fully.
>
> It’s also worth noting that such discrepancies are not uncommon in the field. For instance, in the seminal works of SAC [2] and TD3 [3], the reported performance metrics for their algorithms differ when reproduced, yet both are highly regarded contributions to reinforcement learning.
>
> We hope this clarifies the reasons behind the observed discrepancies and assures you of the validity of our results. Thank you again for your valuable feedback.
>
> **Reference**:
>
> [1] Claude Formanek, Asad Jeewa, Jonathan Shock, Arnu Pretorius, Off-the-Grid MARL: Datasets with Baselines for Offline Multi-Agent Reinforcement Learning, In AAMAS 2023.
>
> [2] Tuomas Haarnoja, Aurick Zhou, Pieter Abbeel, Sergey Levine, Soft Actor-Critic: Off-Policy Maximum Entropy Deep Reinforcement Learning with a Stochastic Actor, In ICML 2018.
>
> [3] Scott Fujimoto, Herke van Hoof, David Meger, Addressing Function Approximation Error in Actor-Critic Methods, IN ICML 2018.

---

### Meta-Review · Area_Chair_3xRE · 2024-12-08

**Metareview:**

This paper tries to solve an important problem in Diffusion-model-based offline MARL, that is, the use of the centralized training with decentralized execution (CTDE) paradigm. The algorithm design is simple yet effective, which is demonstrated in a series of benchmarks through comparisons with various baselines. The authors and reviewers had multiple interactions, and the responses seem to address the key questions by the reviewers.

**Additional Comments On Reviewer Discussion:**

NA

---

### Decision · Program_Chairs · 2025-01-22

Accept (Poster)